# Rbm24a dictates mRNA recruitment for germ granule assembly in zebrafish

Yizhuang Zhang [1,2], Jiasheng Wang[1,2], Hailing Fang [1,2], Shuqi Hu[1], Boya Yang[1,2], Jiayi Zhou[1,2], Raphaëlle Grifone[3], Panfeng Li[1], Tong Lu[1,2], Zhengyang Wang [4], Chong Zhang[5], Yubin Huang[6], Dalei Wu [6], Qianqian Gong [1,2], De-Li Shi [3,7✉], Ang Li [1✉] & Ming Shao [1,2,8✉]

## Abstract

The germ granules are ribonucleoprotein (RNP) biomolecular condensates that determine the fate of primordial germ cells (PGCs) and serve as a model for studying RNP granule assembly. Here, we show that the maternal RNA-binding protein Rbm24a is a key factor governing the specific sorting of mRNAs into germ granules. Mechanistically, Rbm24a interacts with the germ plasm component Buc to dictate the specific recruitment of germ plasm mRNAs into phase-separated condensates. Germ plasm particles lacking Rbm24a and mRNAs fail to undergo kinesin-dependent transport toward cleavage furrows where small granules fuse into large aggregates. Therefore, the loss of maternal Rbm24a causes a complete degradation of the germ plasm and the disappearance of PGCs. These findings demonstrate that the Rbm24a/Buc complex functions as a nucleating organizer of germ granules, highlighting an emerging mechanism for RNA-binding proteins in reading and recruiting RNA components into a phase-separated protein scaffold.

**Keywords** Rbm24; Ribonucleoprotein Granule; RNA Recruitment; Primordial Germ Cell; Degron
**Subject Categories** Development; RNA Biology

## Introduction

How germ cells differentiate in the background of somatic fate represents a long-standing interest in developmental and reproductive biology. A preformation strategy for germ cell development is adopted by a large group of animals such as *C. elegans*, *Drosophila*, zebrafish, *Xenopus* and Reptilia (Bertocchini and Chuva de Sousa Lopes, 2016; Eno et al, 2019; Kloc et al, 2001; Lehmann, 1992; Seydoux and Braun, 2006), in which maternal factors play a crucial role in the assembly of specialized cytoplasmic membrane-less structures called germ granules (i.e., phase-separated germ plasm particles or aggregates) and the specification of germ cells (Bontems et al, 2009; Marlow and Mullins, 2008). For example, during oogenesis in zebrafish, germ plasm components are initially localized in the Balbiani body (BB), a phase-separated condensate, and then distributed to different regions of the cortical layer in the oocyte as a result of BB dissociation. After fertilization, germ plasm components reassemble rapidly and are gathered at the edge of the blastodisc as small granules. During the first two cell cycles, germ granules are transported to and fuse at four distal points of the cruciform cleavage furrows in a manner that is dependent on kinesin-1, actin-myosin, and tight junctions (Campbell et al, 2015; Moravec and Pelegri, 2020; Rostam et al, 2022). During subsequent cell divisions, large germ plasm aggregates are asymmetrically inherited in a small group of blastomeres, which subsequently acquire the fate of primordial germ cells (PGCs) in a cell-autonomous manner (D'Orazio et al, 2021; Eno et al, 2019). After 70%-epiboly, PGCs begin to migrate as two clusters toward the mesoderm-derived gonadal ridge located on both sides of the anterior yolk extension, where they will further differentiate into sperms or oocytes (Grimaldi and Raz, 2020; Jamieson-Lucy and Mullins, 2019). The available evidence suggests that key molecular mechanisms underlying PGC determination are largely conserved, although the localization of germ plasm components may differ among species (Aguero et al, 2017; Escobar-Aguirre et al, 2017).

Germ granules, like other ribonucleoprotein (RNP) granules, such as chromatoid bodies, stress granules, P-bodies, nucleoli and paraspeckles, play essential roles in determining cell fate and regulating RNA metabolism (Banani et al, 2017; Gomes and Shorter, 2019). They are formed through multivalent interactions between RNAs, proteins as well as proteins and RNAs (Ripin and Parker, 2023; Tauber et al, 2020). However, mechanisms underlying the assembly of these RNP granules are largely unknown. In particular, it is still debated whether proteins or RNAs coordinate the aggregation of RNP components such as stress granules and

[1]Shandong Provincial Key Laboratory of Animal Cell and Developmental Biology, School of Life Sciences and Qilu Hospital (Qingdao), Cheeloo College of Medicine, Shandong University, 266237 Qingdao, China. [2]Key Laboratory for Experimental Teratology of the Ministry of Education, Shandong University, 266237 Qingdao, China. [3]Sorbonne Université, Institut de Biologie Paris-Seine (IBPS), UMR CNRS 8263, INSERM U1345, Development, Adaptation and Ageing, Paris, France. [4]Shandong University Taishan College, 266237 Qingdao, China. [5]Zhanjiang Institute of Clinical Medicine, Central People's Hospital of Zhanjiang, Guangdong Medical University Zhanjiang Central Hospital, 524045 Zhanjiang, China. [6]State Key Laboratory of Microbial Technology, Institute of Microbial Technology, 266237 Qingdao, China. [7]Fang Zongxi Center, Key Laboratory of Marine Genetics and Breeding, College of Marine Life Sciences, Ocean University of China, Qingdao, China. [8]Shandong University-Yuanchen Joint Biomedical Technology Laboratory, 266237 Qingdao, China. ✉E-mail: de-li.shi@upmc.fr; angli41@sdu.edu.cn; shaoming@sdu.edu.cn

P-bodies (Ripin and Parker, 2023; Tauber et al, 2020). In *Drosophila*, proteins play a key role in pole plasm assembly (Curnutte et al, 2023). It is well established that several proteins, such as Oskar, Vasa and Tudor, are critically involved in the biogenesis of the pole plasm by regulating mRNA localization and translation (Lehmann, 2016). In zebrafish, a large number of germ plasm components have been identified since the discovery of *vasa* (*ddx4*) as the first localized maternal germ plasm mRNA (Yoon et al, 1997). Bucky ball (Buc), Dead end1 (Dnd1), and Nanos3 are important for PGC formation, with Dnd1 and Nanos3 playing an essential role in PGC migration and maintenance (Bontems et al, 2009; Draper et al, 2007; Gross-Thebing et al, 2017; Koprunner et al, 2001; Marlow and Mullins, 2008; Weidinger et al, 2003). In addition, Dnd1 protein facilitates the localization of *nanos3* mRNAs to the periphery of germ plasm particles and regulates their accessibility to the translation machinery (Westerich et al, 2023). Besides the three essential factors, maternal Tdrd6a, Tdrd7a, Rgs14a, *mir202-5p* and Hook2 function in the organization of germ granules and/or migration of PGCs, but they are not essential for the differentiation of PGCs (Hartwig et al, 2014; Jin et al, 2020; Roovers et al, 2018; Strasser et al, 2008). Dazl overexpression inhibits miR430 and promotes the stability and polyadenylation of germ plasm mRNAs (Maegawa et al, 2002; Takeda et al, 2009). However, the precise function of Dazl and other germ plasm components, such as Piwil1 and Ddx4, in germ granule formation and PGC specification remains unclear due to obstacles in obtaining their maternal mutants (Bertho et al, 2021; Hartung et al, 2014; Houwing et al, 2007). Thus, there is still a gap in understanding how germ plasm RNAs are recruited explicitly into the condensates and why they are maintained in PGCs but degraded in somatic cells. Uncovering these riddles depends on identifying new germ plasm components essential for germ plasm particle assembly and PGC development.

Rbm24 is an evolutionarily conserved RNA-binding protein (RBP) with a single N-terminal RNA recognition motif (RRM) and functions as a multifaceted post-transcriptional regulator of cell differentiation and homeostasis (Grifone et al, 2020). It controls pre-mRNA alternative splicing and mRNA stability during myogenesis, cardiomyogenesis, and inner ear hair cell differentiation (Cheng et al, 2020; Jin et al, 2010; Liu et al, 2019; Wang et al, 2023; Yang et al, 2014; Zhang et al, 2020). Zebrafish has two *rbm24* genes, *rbm24a* and *rbm24b* (Maragh et al, 2014), of which *rbm24a* displays the same expression pattern and function as Rbm24 in other vertebrates. Our previous work has shown that zygotic mutation of *rbm24a* impairs lens differentiation and causes the formation of cataracts. In addition, Rbm24a interacts with Cpeb1b and Pabpc1l, critical members of the cytoplasmic polyadenylation mechanism, to maintain poly(A) tail length of target mRNAs, thereby promoting their stability and translation efficiency in differentiating lens fiber cells (Shao et al, 2020). Maternal *rbm24a* is also highly expressed in oocytes and cleavage-stage embryos, but its function is unclear. In this work, we demonstrate that maternal Rbm24a functions as an organizer to recruit mRNAs and assemble germ plasm aggregates. Importantly, the loss of maternal Rbm24a prevents large germ plasm aggregate formation and PGC specification, leading to complete sterility. These findings uncover a critical role of Rbm24a in germ cell fate determination during vertebrate development.

# Results

## Rbm24a is a novel protein component of the germ plasm

Published RNA-seq data suggest that *rbm24a* transcripts are maternally supplied with a drastic decrease after zygotic genome activation (White et al, 2017) (Fig. EV1A). In situ hybridization (ISH) analysis showed that *rbm24a* transcripts are ubiquitously distributed during early cleavage and blastula stages (Fig. EV1B). To assay the function of maternal Rbm24a protein, we first asked how it is localized subcellularly during early development. As no specific antibody against zebrafish Rbm24a is available, we created a *rbm24a-GFP* knock-in line (designated as *rbm24a-GFP* KI) with a C-terminus GFP tag sequence ligated to the endogenous *rbm24a* coding region following an intron-based genome editing strategy (Li et al, 2015) (Figs. 1A and EV1C). In the F0 embryos, zygotic Rbm24a-GFP protein was expressed explicitly in the lens, heart, skeletal muscle, and inner ear hair cells, consistent with previous observations by in situ hybridization and immunofluorescence staining (Fetka et al, 2000; Grifone et al, 2014; Poon et al, 2012) and highly matching the *rbm24a* mRNA expression pattern in different tissues (see also the rightmost panel in Fig. EV1B). This suggests a successful knock-in event occurring during the early cleavage stage, which we term early integration (inset in Fig. EV1C). The efficiency of this early integration event was 0.45% (*n* = 1100), and all three early integration founders exhibited germline transmission with high percentages of knock-in offspring (Fig. EV1D).

We next examined the expression of maternal Rbm24a-GFP in early embryos spawned by the female F1 KI fish and were pleasantly surprised by the localization of GFP signals to the germ plasm particles and early PGCs (Fig. 1B). At the 1-cell stage, Rbm24a-GFP was present as tiny dots in the entire cortical layer. Time-lapse analysis revealed that these dots were engaged in active directed transport to accumulate along the forming cleavage furrows during the first several cell divisions (Fig. 1B; Movie EV1). At the 1-somite stage, maternal Rbm24a-GFP protein specifically emerged in PGCs, which appeared as two clusters located on both sides of the notochord (Fig. 1B). At 24 h post fertilization (hpf), it is interesting to see that the maternal protein still persisted in PGCs already migrated to the gonadal ridge and was trapped into large perinuclear granules (arrows in Fig. 1B), whereas zygotically expressed Rbm24a-GFP protein was almost evenly distributed in myoblasts within the adjacent myotomes (Fig. 1B). Moreover, Rbm24a-GFP showed strong colocalization with *ddx4-*, *dazl-*, and Piwil1-positive aggregates in early cleavage embryos (Fig. 1C), and it also perfectly merged with Ddx4 protein in the perinuclear granules of PGCs at 24 hpf (Fig. 1D).

We also examined the localization of Rbm24a-GFP during early oogenesis and found that it was exclusively present in the Balbiani body (BB) of stage I oocytes, occupying a slightly larger region than *ddx4* and *dazl* mRNAs (Fig. EV2A–F). After BB dissociation, Rbm24a-GFP protein also spread to the entire cortical layer of stage III oocytes (Fig. EV2G–L). Interestingly, the distribution of Rbm24a in the cortical layer was not uniform or clustered as dots; instead, it exhibited an appearance of irregular patches (Fig. EV2M–O). Together, these results demonstrate that the Rbm24a protein is a novel germ plasm component in zebrafish.

The homozygous *rbm24a-GFP* KI line is viable and fertile (Appendix Fig. S1a–d). Thus, GFP tagging does not affect the

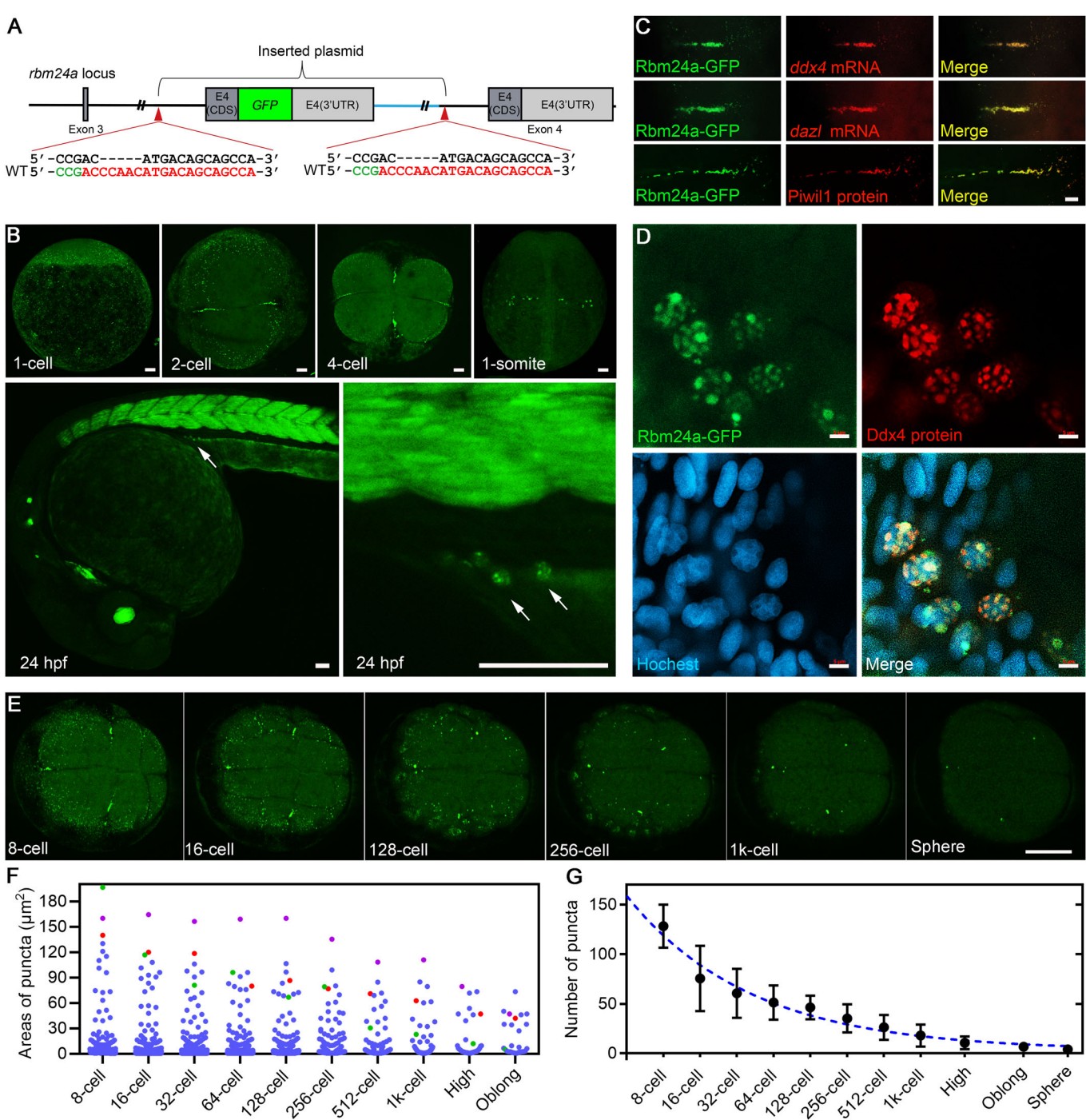

Figure 1. Rbm24a is a protein component of the germ granule.

(A) Schematic illustrating the strategy for generating the *rbm24a-GFP* KI line. A GFP-containing tagging cassette was introduced into the last intron of the *rbm24a* locus to replace the last exon. (B) Germ plasm- and germ cell-specific distribution of maternal Rbm24 protein. Scale bars, 50 μm. (C) Colocalization of Rbm24a protein with germ plasm mRNAs *ddx4* and *dazl* as well as with the germ plasm protein Piwil1 in the germ plasm aggregates. Scale bar, 20 μm. (D) Colocalization of Rbm24a and Ddx4 in perinuclear germ granules within PGCs at 24 hpf. Scale bars, 5 μm. (E) Live imaging of Rbm24a-GFP from 8-cell stage to sphere stage. Scale bar, 200 μm. (F) Size distribution of Rbm24a-GFP- positive germ granules in the early embryos. Dots are colored to trace individual germ granules. $n = 555$. (G) The numbers of germ plasm condensates are reduced in an exponential decay pattern during early development ($n = 3$ independent biological samples). Data are presented as mean ± SD. Source data are available online for this figure.

function of the endogenous Rbm24a protein. Moreover, the distribution pattern of Rbm24a-GFP is similar to the localization of endogenous *rbm24a* transcripts, and *rbm24a-GFP* transcripts in the homozygous knock-in line also showed an identical expression pattern to *rbm24a* mRNA in the wild-type embryo at 24 hpf (Appendix Fig. S1e). As the *rbm24a-GFP* KI line represents a convenient live marker for germ granules, it can be used to trace the dynamic changes of germ plasm aggregates during early development. By performing such an analysis, we found that germ granules were initially excessive (Fig. 1E). During the first three cleavages, germ granules became rapidly aggregated at the cleavage furrows and adhered to each other, thus contributing to the formation of larger particles. However, a considerable number of small granules were not engaged in this process. Instead, they were distributed around the nuclei at the edge of the blastodisc and moved rhythmically with the cell cycle (Movies EV1 and 2). The aggregation of germ granules is essential for stabilizing the germ plasm components because small particles, but not larger aggregates, quickly disappeared from the 128-cell stage onward (Fig. 1E; Movies EV2 and 3). The number of germ plasm condensates also decreased over time due to fusion and dissociation of small granules (Fig. 1F,G). We measured the size of 555 condensates at the 8-cell stage and tracked them in real-time until the sphere stage (Fig. 1F). At the 32-cell stage, the minimum projected area required for successful retention in PGCs was 43.3 μm² (projected area).

## Maternal *rbm24a* mutant is viable but sterile

What is the function of this novel germ plasm protein? As zygotic *rbm24a* mutants exhibit lethal phenotypes such as heart failure, cataracts, and hair cell defects, we adopted a conditional knockout strategy established in our previous work to generate maternal *rbm24a* mutants (Zhang et al, 2021). To facilitate screening, we first constructed a knock-in line, designated as *rbm24a-RFP* KI*zpc:cas9*, by fusing the *RFP* sequence inframe with the *rbm24a* coding region and simultaneously inserting a *zpc:cas9* cassette downstream of the *rbm24a* gene (Appendix Fig. S2a). Rbm24a-RFP signals were present in tissues where the endogenous *rbm24a* gene is expressed (Appendix Fig. S2b). The early integration efficiency (0.56%, $n = 530$) and germline transmission rate were similar to that observed in generating the *rbm24a-GFP* KI line (Appendix Fig. S2c). Like the *rbm24a-GFP* KI line, the homozygous *rbm24a-RFP* KI*zpc:cas9* line was healthy and fertile (Appendix Fig. S1a–d). Moreover, the *zpc:cas9* insert could support sufficient maternal expression of Cas9 protein, as injecting *bmp2b* sgRNA into embryos spawned by *rbm24a-RFP* KI*zpc:cas9* female fish led to a near 100% dorsalized phenotype (Appendix Fig. S2d,e). The high genome editing efficiency of *rbm24a-RFP* KI*zpc:cas9* remained stable when traced in three consecutive generations (Appendix Fig. S2f).

We also constructed a transgenic vector allowing the expression of a maternal BFP marker and four highly efficient sgRNAs against *rbm24a* coding sequence (Appendix Table S1). After introducing this vector into homozygous *rbm24a-RFP* KI*zpc:cas9* embryos by Tol2 transposition (Fig. 2A), it was easy and rapid to screen maternal *rbm24a* mutants (M*rbm24a*) among BFP-positive embryos in the next generation, because they lack Rbm24a-RFP protein due to disruption of the *rbm24a* gene (Fig. 2A; Appendix Fig. S3a–e). By measuring the remaining wild-type *rbm24a* transcripts at 1-cell

stage using qRT-PCR and capillary electrophoresis (Appendix Fig. S3f–i), we observed a genome editing rate ranging from 71% to 81% in BFP-positive embryos (Appendix Fig. S3j). M*rbm24a* embryos appeared normal and could survive to adulthood (Appendix Fig. S4a), while maternal and zygotic *rbm24a* mutants (MZ*rbm24a*) exhibited identical lethal phenotypes as zygotic mutants (Appendix Fig. S4a), suggesting that maternal Rbm24a protein does not contribute to organogenesis. Nevertheless, M*rbm24a* adult fish exhibited an all-male phenotype. They had sexual behaviors but failed to fertilize eggs when crossed with wild-type females (Fig. 2B,C). Anatomically, in sharp contrast to wild-type males displaying testes along both sides of the swim bladder, maternal *rbm24a* mutant fish deposited fatty tissue at corresponding locations (Fig. 2D,E). Analyses by sectioning, H&E staining and fluorescence in situ hybridization (FISH) showed the complete absence of male germ cells and spermatozoa in M*rbm24a* fish (Fig. 2F,G; Appendix Fig. S4b). These defects are highly similar to the phenotypes of *dnd1*-deficient fish (Siegfried and Nusslein-Volhard, 2008).

## Maternal *rbm24a* mutant is devoid of PGCs

The lack of germ cells in adult M*rbm24a* fish may be caused by a disrupted formation of PGCs in the early embryos. We thus examined a panel of PGC markers by ISH or immunofluorescence. In M*rbm24a* mutants at 24 hpf, there was a complete absence in the expression of PGC-specific mRNAs, including *nanos3*, *ddx4*, *ca15b*, *tdrd7a*, and *kop* (Fig. 2H), and PGC-specific proteins such as Piwil1 and Ddx4 were also undetectable (Fig. 2I–L). We also injected *YFP* fused to the 3'-UTR of the *nanos3* mRNA (Mishima et al, 2006) into wild-type and M*rbm24a* embryos but failed to detect YFP-positive PGCs in the mutants at 24 hpf (Fig. 2M,N). The absence of PGCs in M*rbm24a* mutants could be observed at 6 hpf, the initial formation of this cell type (Fig. 2O). These results collectively suggest a complete absence of PGCs following the loss of Rbm24a function.

Next, we performed single-cell RNA-seq (scRNA-seq) analysis to compare gene expression between sibling embryos and M*rbm24a* mutants. Another purpose of this omic analysis was to examine cell populations possibly affected by the loss of maternal Rbm24a. At the shield stage, 14,455 cells in two batches were successfully collected from siblings and M*rbm24a* embryos, which were then profiled by scRNA-seq. An unbiased clustering analysis has allowed us to identify distinct cell types (Fig. 2P). For cell-type annotation, differentially expressed genes (DEGs) in each cluster were analyzed for known cell-type specific marker genes. These clusters were categorized as ectoderm, lateroventral mesoderm, dorsal mesoderm (dorsal organizer), endoderm, enveloping layer (EVL), dorsal forerunner cells (DFCs), and PGCs. They also included three groups of cells without known marker genes and thus combined as "not determined" (Fig. 2P; Appendix Fig. S5a). Their top differentially expressed genes (DEGs) predominantly consisted of housekeeping genes with relatively mild up- or downregulation (Appendix Fig. S5b). We propose that these cells may represent pluripotent cells undergoing differentiation, characterized by unstable gene expression.

The top DEGs in annotated clusters were summarized by a dot plot and a heat map (Appendix Fig. S6a,b). Consistent with an early specification status, some DEGs in ectoderm and mesoderm showed expression across cell types, suggesting a multipotent status of these clusters (Appendix Fig. S6a,b). After splitting into sibling and

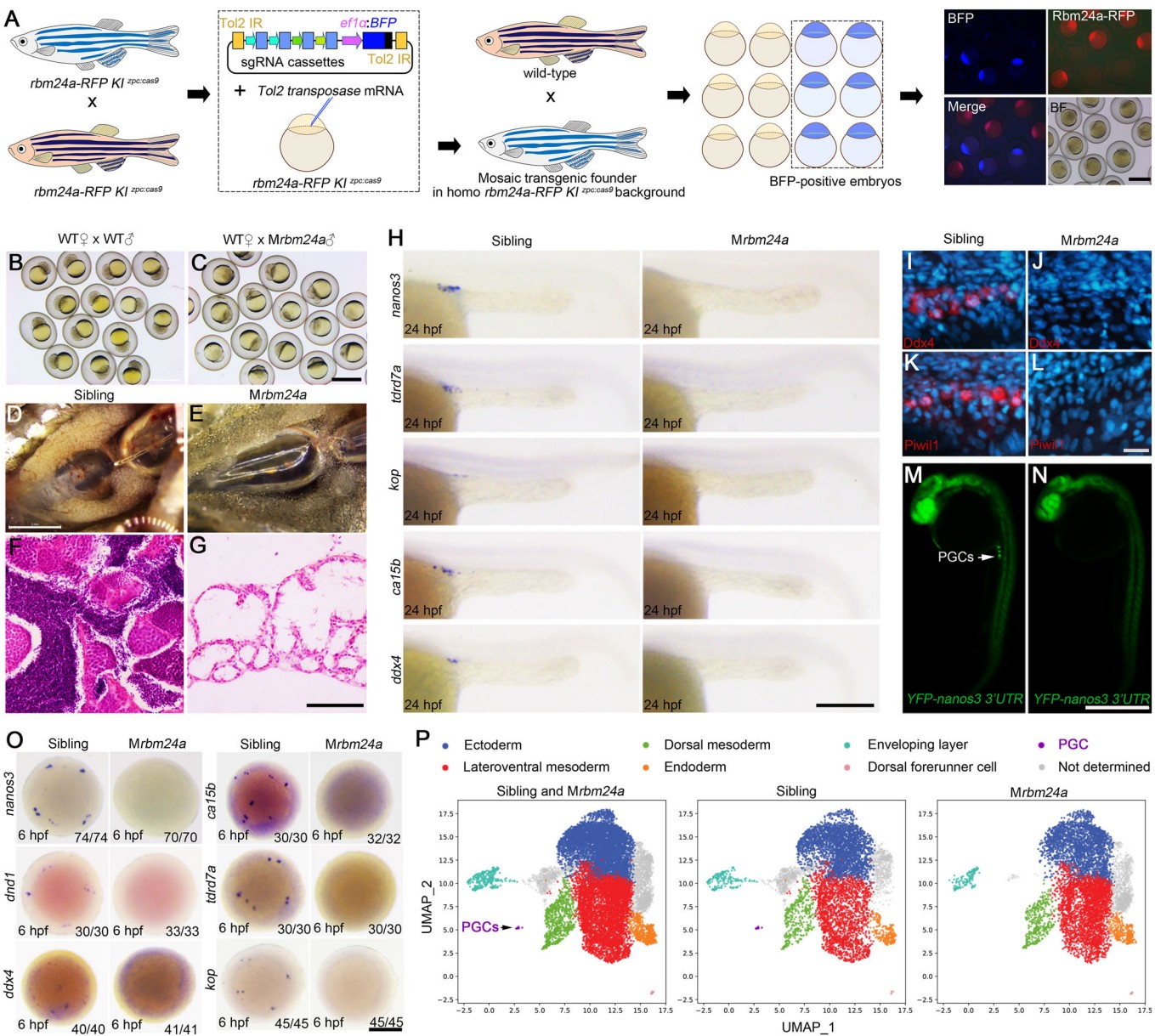

**Figure 2. Loss of maternal *rbm24a* results in the specific absence of PGCs.**

(A) Pipeline to obtain maternal *rbm24a* mutant embryos. Scale bar, 1 mm. (B, C) Embryos spawned by crossing wild-type female and M*rbm24a* male fish fail to undergo cleavage. Scale bar, 1 mm. (D, E) Absence of testis structure in an M*rbm24a* mutant adult. Scale bar, 2 mm. (F, G) Testicular tubules from an M*rbm24a* adult contain no germ cells and exhibit a vacuolar shape. Scale bar, 12.5 μm. (H) Absence of germ plasm mRNA markers in M*rbm24a* embryos at 24 hpf. Scale bar, 250 μm. (I–L) Absence of germ plasm protein markers Ddx4 and Piwil1 in M*rbm24a* embryos at 24 hpf. Scale bar, 500 μm. (M, N) Loss of YFP signals in PGCs of an M*rbm24a* embryo at 24 hpf expressing *YFP* fused with the 3'-UTR of *nanos3* mRNA. Green serves as a pseudocolor of YFP fluorescence. Scale bar, 500 μm. (O) Loss of PGCs in M*rbm24a* embryos at the shield stage, as examined by the expression of specific markers. Numbers in the bottom right corner indicate the ratio of analyzed embryos with the representative expression pattern shown in each panel. Scale bar, 250 μm. (P) Specific loss of PGCs in M*rbm24a* embryos at 6 hpf, as shown by the clustering map of scRNA-seq. Source data are available online for this figure.

M*rbm24a* subsets, the most significant difference was the absence of PGCs in M*rbm24a* mutants, further supporting the data obtained by directly examining the expression of PGC markers (Fig. 2P; Appendix Fig. S6c). By contrast, the cell number of other identified cell types was approximately similar between siblings and M*rbm24a* embryos (Appendix Fig. S6c). The uniform manifold approximation and projection (UMAP) cluster map also highlighted a specific absence of

PGC markers in M*rbm24a* embryos (Appendix Fig. S6d). One cluster of "not determined" (not determined-3), located near the EVL group, appeared to be markedly reduced in the mutants (Appendix Fig. S5a). In contrast, another cluster, "not determined-2", was increased in M*rbm24a* embryos (Appendix Fig. S5a). Since two of the top DEGs in "not determined-3" are *krt4* and *lye*, which are markers of the EVL (Appendix Fig. S5b,c), it is likely that "not determined-3" represents

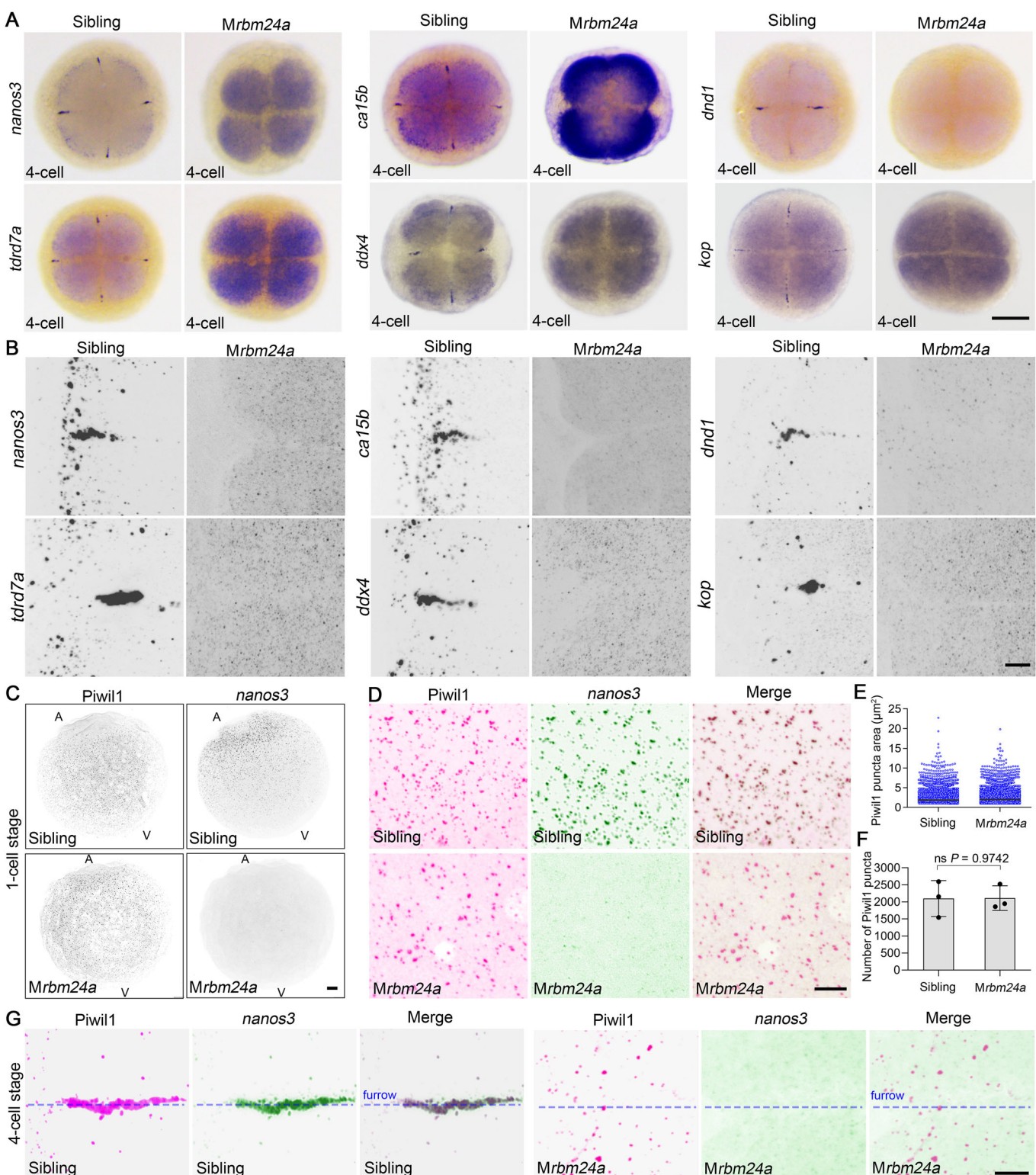

cells transitioning to the EVL, while "not determined-2" may correspond to an earlier stage of this transition. The subtle gene expression differences between these two somatic cell clusters could be related to the function of Rbm24a; however, they may also reflect artifacts due to slight developmental stage variations between siblings and mutants. Hence, the loss of maternal Rbm24a predominantly impairs germ cell differentiation, with a relatively limited influence on the development of somatic cells.

◄ **Figure 3.   Loss of maternal Rbm24a specifically causes the dispersion of germ plasm mRNAs from phase-separated structures.**

(A) Distribution of germ plasm mRNAs in wild-type and M*rbm24a* embryos at the 4-cell stage. Scale bar, 200 μm. (B) FISH combined with confocal imaging showing the dispersal of germ plasm mRNAs in M*rbm24a* mutants. Scale bar, 20 μm. (C, D) The absence of maternal Rbm24a disrupts the localization of germ plasm mRNAs but does not affect the phase separation of germ plasm proteins. A animal pole, V vegetal pole. Scale bars, 20 μm. (E) Similar distribution of germ granules in wild-type embryos and M*rbm24a* mutants. For wild-type n = 6294, For M*rbm24a* mutants n = 6332. (F) Loss of maternal Rbm24a does not affect the number of germ granules (n = 3 independent biological samples). Values are means ± SD. ns, not significant, unpaired Student's *t* test. (G) Rbm24a-deficient germ granules fail to form large aggregates along the cleavage furrows. Magenta and green are pseudocolors designating Piwil1 protein and *nanos3* mRNA, respectively. Scale bar, 20 μm. Source data are available online for this figure.

## Maternal Rbm24a is required for the recruitment of germ plasm mRNAs to the condensates

As an RBP, Rbm24a may influence the localization of germ plasm-specific mRNAs to regulate the formation of phase-separated condensates. By examining the status of *nanos3, ddx4, dnd1, ca15b, tdrd7a* and *kop* mRNAs at the 4-cell stage, we were astonished to find that they were all absent at the cleavage furrows in M*rbm24a* embryos (Fig. 3A). The enhanced cytoplasmic staining of these mRNAs in M*rbm24a* embryos raises a possibility that they may be released from highly compacted RNP granules. We then performed FISH to analyze the Rbm24a-dependent distribution of germ plasm mRNAs at high resolution. Germ plasm mRNAs were tightly packed into condensates in wild-type embryos and assembled into large aggregates along the cleavage furrows (Fig. 3B). By contrast, in M*rbm24a* embryos, these mRNAs displayed tiny spot patterns when visualized at higher laser intensity and gain, possibly representing single molecules that were distributed freely in the cytoplasm (Fig. 3B). These results strongly demonstrate that Rbm24a is required for maintaining germ plasm mRNAs in the phase-separated condensates.

Is Rbm24a involved in the formation of phase-separated germ plasm protein scaffold? To address this question, we checked the status of germ granules in M*rbm24a* embryos at the 1-cell stage by examining Piwil1 protein and *nanos3* mRNA. Unexpectedly, germ granules containing Piwil1 protein but not *nanos3* mRNA were normally formed (Fig. 3C,D). The size and density of germ granules were similar between M*rbm24a* mutants and wild-type embryos (Fig. 3E,F). At the 4-cell stage, however, there were no large Piwil1-positive aggregates at the cleavage furrows in M*rbm24a* embryos (Fig. 3G). When simultaneously visualizing Piwil1 and *nanos3* mRNA, they showed no colocalization and *nanos3* mRNA failed to localize in even small germ granules in M*rbm24a* embryos, either at 1-cell stage or 4-cell stage (Fig. 3D,G). These observations further suggest that Rbm24a controls the recruitment of germ plasm mRNAs rather than protein components.

To examine whether the granular distribution of other germ plasm protein components was affected by the absence of maternal Rbm24a, we injected *tdrd6-myc* and *celf1-myc* mRNAs into 1-cell stage embryos and analyzed the localization of the corresponding proteins at 32-cell stage by immunofluorescence staining. Both Tdrd6-myc and Celf1-myc colocalized with endogenous Piwil1 in the condensates either in wild-type embryos or M*rbm24a* mutants, even though large aggregates were absent in the mutants at 4-cell stage (Fig. EV3A). We also overexpressed Buc-GFP and examined its localization by live imaging. In wild-type embryos, Buc-GFP protein accumulated in both small granules and large aggregates. In M*rbm24a* embryos, however, it was exclusively localized in small germ granules (Movie EV4). Phosphorylated non-muscle myosin II (NMII-p) is another protein marker for germ plasm condensates (Nair et al, 2013). It perfectly colocalized with Rbm24a (Fig. EV3B). Upon removal of maternal Rbm24a, NMII-p also failed to form large aggregates along the cleavage furrows but was still present in small germ granules (Fig. EV3C). These findings strongly suggest that Rbm24a does not affect the initial phase separation of germ plasm protein scaffold but controls the aggregation and fusion of granules into large aggregates at the cleavage furrows. They were further confirmed by experiments separating germ granules from the cytoplasm of 4-cell stage embryos through centrifugation (Fig. EV3D,E). We found that in the supernatants, the levels of germ plasm mRNAs, including *nanos3, dnd1, dazl* and *ddx4*, were 1.6–6-folds higher in M*rbm24a* embryos than those in siblings (Fig. EV3F). As Rbm24a does not contribute to BB formation and dissociation during oogenesis (Fig. EV4), we can conclude that it is a crucial component in recruiting germ plasm mRNAs into phase-separated germ granules.

## Rbm24a is required for kinesin-dependent transport of the germ granules toward the cleavage furrows

To directly follow the process associated with the formation of large germ plasm aggregates, we established a *buc-gfp* transgenic line under wild-type or M*rbm24a* background. Live imaging indicated that Buc-GFP-labeled germ granules rapidly and directly moved towards the cleavage furrows at the moment of its formation (Fig. 4A,B; Movies EV5 and 6), possibly along parallel microtubule bundles assembled just before the onset of cleavage (Urven et al, 2006). In M*rbm24a* mutants, the initial condensation of Buc-GFP was normal (Appendix Fig. S7a,b), consistent with the observation that the absence of Rbm24a does not disrupt phase separation of protein components (Appendix Fig. S7a). However, the movements of Buc-positive granules were uncoupled from cleavage furrow formation (Fig. 4C,D; Movies EV5 and 6). Both the average and top velocity of particle movements were significantly reduced in M*rbm24a* mutants (Fig. 4E). The movement trajectories of mutant granules also became passive, highly coincident with cell division. However, they did not converge to the cleavage furrows (Fig. 4F). Interestingly, the animal pole-directed transport of germ granules was not affected in M*rbm24a* embryos (Movies EV5), suggesting that only the cleavage furrow-specific transport is controlled by Rbm24a.

Kif5ba, which belongs to the kinesin-1 family, is essential for the recruitment of germ granules to the cleavage furrow (Campbell et al, 2015). We hypothesized that the absence of germ granule transport toward the cleavage furrows in M*rbm24a* mutants may be due to a disruption of kinesin-1-mediated trafficking. This

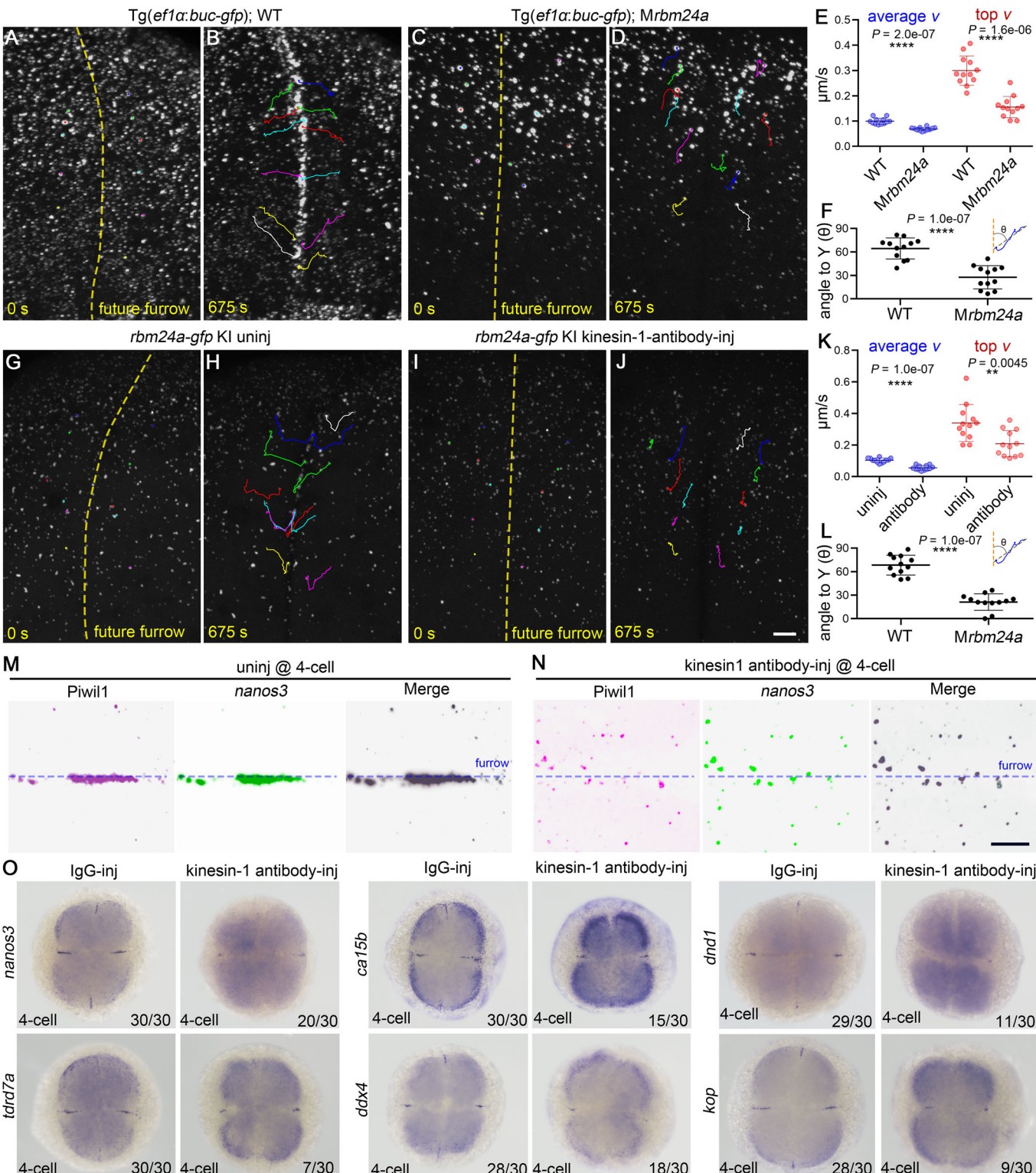

possibility was addressed by using a monoclonal antibody known to inhibit kinesin-1 function in squid (Brady et al, 1990). Blocking kinesin-1 activity in *rbm24a-GFP* KI or wild-type embryos inhibited the movements toward the cleavage furrows and significantly reduced the formation of large aggregates (Fig. 4G–J; Movies EV7 and 8). The slow and passive movements of germ granules in antibody-injected embryos resemble the defects observed in M*rbm24a* mutants (Fig. 4K,L). As expected, blocking kinesin-1-mediated transportation did not affect the recruitment of germ plasm mRNAs into granules (Fig. 4M,N). By assessing the

Figure 4.   Transport of germ granules to the cleavage furrows requires Rbm24a and kinesin-1.

(A, B) Movements of Buc-GFP-positive granules during cleavage. (C, D) Defective movements of Buc-GFP granules in M*rbm24a* mutants. (E, F) Velocity and directionality of marked granules in siblings and M*rbm24a* embryos. Data are presented as mean ± SD, n = 12. ****P < 0.0001, unpaired Student's t test. (G, H) Movement trajectories of marked granules in *rbm24a-GFP* KI embryos during cleavage. (I, J) Disruption of directed movements of Rbm24a-positive granules by injection of a function-blocking antibody against kinesin-1. Scale bar, 20 μm for (A–D, G–J). (K, L) Velocity and directionality of labeled granules in uninjected and kinesin-1 antibody-injected Tg(*ef1a:buc-GFP*) embryos. Data are presented as mean ± SD, n = 12. **P < 0.01 and ****P < 0.0001, unpaired Student's t test. (M, N) Distribution of Piwil1 protein and *nanos3* mRNA at the cleavage furrows of uninjected and kinesin-1 antibody-injected embryos. Scale bar, 20 μm. (O) Compromised localization of germ plasm mRNAs at the cleavage furrows in kinesin-1 antibody-injected embryos. Numbers in the bottom right corner indicate the ratio of analyzed embryos with the representative expression pattern shown in each panel. Scale bar, 200 μm. Source data are available online for this figure.

expression of germ plasm mRNAs via ISH, the reduced formation of germ plasm aggregates at the cleavage furrows was also evident following kinesin-1 inhibition (Fig. 4O).

## Rbm24a is a Buc-binding protein

As Rbm24 has been reported as a splicing regulator, it remains possible that the loss of maternal Rbm24a protein could affect the normal splicing of germ plasm mRNAs and other related genes, such as *buc* and *kif5Ba*, potentially leading to defective mRNA recruitment. To investigate this possibility, we performed a comprehensive transcriptome and qRT-PCR analysis comparing M*rbm24a* mutants with wild-type embryos, focusing on the splicing status and expression levels of germ plasm and related mRNAs. RNA-seq analysis, visualized by mapping the reads onto the genome, revealed no discernible alternative splicing changes between sibling and M*rbm24a* embryos at the 4-cell stage (Appendix Table S2, Appendix Fig. S8). Furthermore, the expression levels of various germ plasm mRNAs were comparable between siblings and M*rbm24a* embryos at the 4-cell stage, with the exception of the intronless *nanos3*, which showed a significant reduction in expression (Appendix Fig. S9). The expression level of *buc* and *kif5Ba* (Appendix Fig. S10a–d) and their splicing were also unaffected in M*rbm24a* mutants (Appendix Table S2; Appendix Fig. S10e,f). These results suggest that the loss of *rbm24a* function does not cause widespread splicing defects in germ plasm-related genes.

Hence, data obtained above suggest that Rbm24a likely functions with other proteins to promote the aggregation of germ plasm mRNAs and the transportation of germ granules. To probe into the protein interactome of maternal Rbm24a protein, we performed immunoprecipitation followed by mass spectrometry (IP/MS) using *rbm24a-GFP* KI embryos at 4 hpf. After confirming the specificity and efficiency of the IP procedure (Appendix Fig. S11a,b), the precipitants were subjected to SDS-PAGE and silver staining. Subsequently, each lane was cut into six pieces mainly based on the molecular masses (Appendix Fig. S11c). LC-MS/MS analysis identified 1066 proteins exclusively present in the sample from *rbm24a-GFP* KI embryos. Another 158 proteins showed at least a threefold enrichment in *rbm24a-GFP* KI embryos compared to wild-type controls (Fig. 5A). Notably, Pabpc1l, a previously identified Rbm24a-binding protein (Shao et al, 2020), was enriched more than 6.5-fold over the control (Fig. 5A). Furthermore, GO analysis indicated that putative Rbm24a-interacting proteins were related to the mRNA metabolic process, including translation, splicing and localization (Fig. 5B). Therefore, Rbm24a presents the potential to interact and/or function with these proteins in a common complex.

The IP/MS data indicated that Buc, Celf1 and Tdrd6, three germ plasm protein components, were associated with Rbm24a (Fig. 5A). However, the interaction of Rbm24a with Buc, but not with Celf1 or Tdrd6, was further validated experimentally by co-immunoprecipitation (Co-IP) using tagged proteins overexpressed in wild-type embryos (Fig. 5C,D; Appendix Fig. S12a,c–f) or those expressed at physiological levels in genetically modified lines (Appendix Fig. S12b). We next mapped the interaction domains between Rbm24a and Buc using different truncated versions of these proteins. Clearly, the disordered C-terminal region of Rbm24a (ΔRRM) mediates its binding to Buc (Fig. 5C,D), and Buc also uses a disordered region (amino acids 581–614) to interact with Rbm24a (Fig. 5E,F). Thus, it is likely that Rbm24a and Buc act together to assemble the germ granules. Interestingly, at optimal doses, Rbm24a-RFP showed perfect colocalization with Buc-GFP around the nucleus of HEK293 cells, closely mimicking the granular patterns of the germ plasm in zebrafish embryos (Fig. 5G). Similar to the requirement of Rbm24a for Buc localization at the cleavage furrows, Rbm24a-GFP failed to form germ granules and was uniformly distributed in the cytoplasm of M*buc* mutants (Fig. 5H). These results not only demonstrate the interaction between Rbm24a and Buc but also uncover a mutual regulation of their localization in forming the germ plasm condensates.

## Rbm24a-Buc complex interacts with germ plasm mRNAs and protects them from degradation

As Rbm24a is required for the recruitment of germ plasm mRNAs, we examined their interaction by RNA immunoprecipitation followed by RNA-seq (RIP-seq) analysis using *rbm24a-GFP* KI embryos at 4-cell stage. Wild-type embryos served as a control for two reasons: first, there is no GFP antigen in these embryos, so the mRNAs precipitated by the GFP antibody should represent background levels; second, the *rbm24a-GFP* KI fish exhibit normal development and fertility, indicating that the function of Rbm24a-GFP should be comparable to the wild-type protein.

In this RIP-seq analysis, a total of 5981 transcripts were identified, with 65.8% peaking at the coding regions and 3'-UTRs (Fig. 6A). There were 838 mRNAs showing significant enrichment in the *rbm24a-GFP* KI group (Fig. 6B). As expected, nearly all germ plasm mRNAs displayed a high affinity of binding with Rbm24a (Fig. 6B). The putative *rbm24a*-binding mRNAs were enriched in terms like "reproduction", "gamete generation", "pole plasm" and "ribonucleoprotein granule" (Fig. 6C). We also performed a parallel RIP-seq analysis on *rbm24a-GFP* KI embryos using a scrambled IgG as a negative control and obtained similar results, with most germ plasm mRNAs significantly enriched in the GFP-Trap group

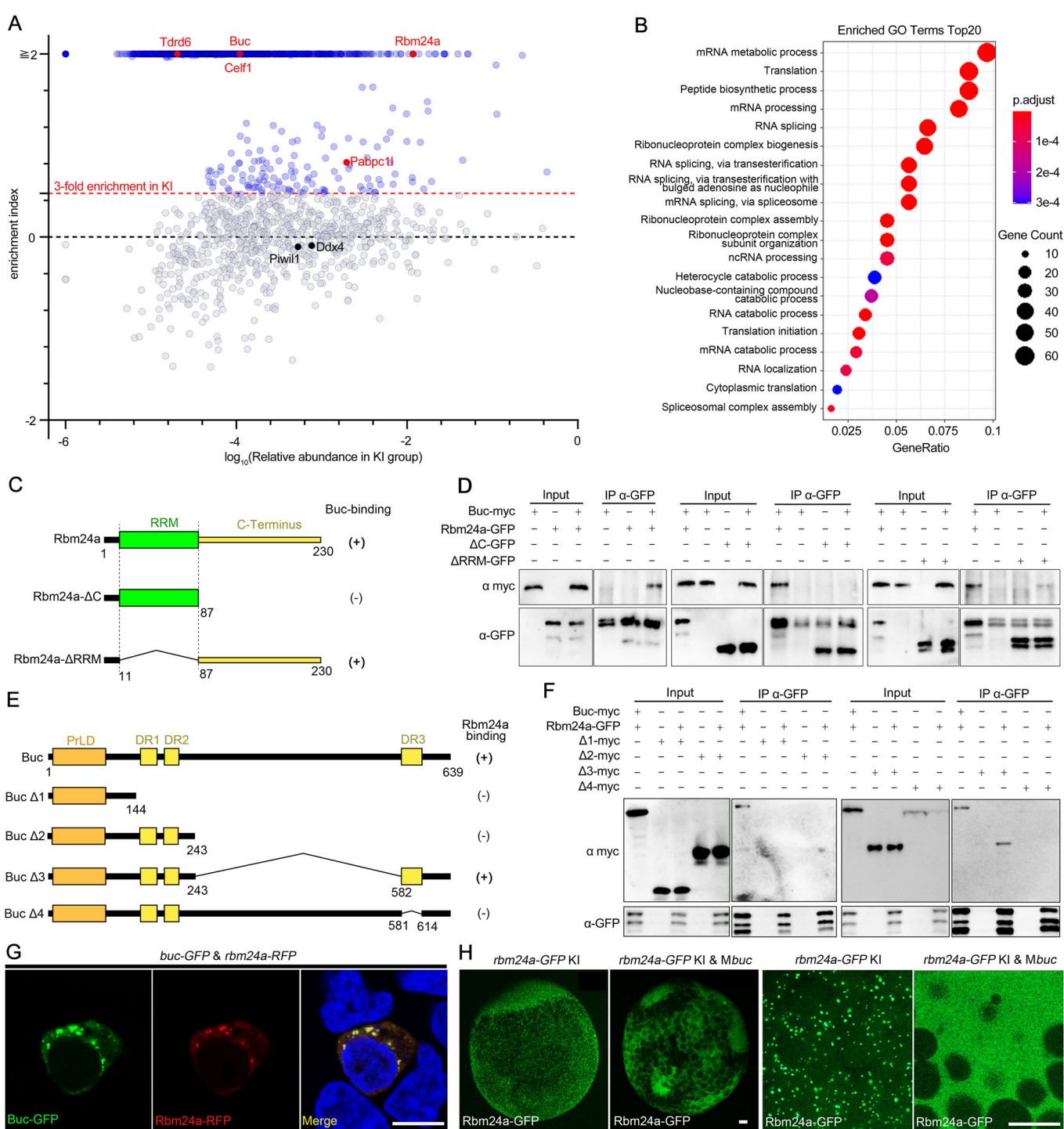

**Figure 5. Rbm24a interacts with Buc.**

(A) Scatter plot of enrichment to relative intensity (see "Methods"). Blue dots designate proteins with more than a threefold enrichment. Red dots highlight previously known Rbm24a-binding proteins and protein components of the germ plasm. (B) GO analysis of Rbm24a-interacting proteins. The hypergeometric test is employed for statistical assessment, and the Benjamini–Hochberg (BH) method is utilized for *P* value adjustment. (C) Full-length and truncations of Rbm24a. (D) Co-IP analysis shows the interaction between Buc and the C-terminal region of Rbm24a. (E) Full-length and truncated versions of Buc. (F) Co-IP shows the interaction between Rbm24a and a disordered region (DR3) of Buc. (G) Colocalization of Rbm24a with Buc in HEK293 cells. Scale bar, 10 μm. (H) Mutation of Buc eliminates the phase-separated pattern of Rbm24a-GFP in a 1-cell stage embryo. Scale bars, 50 μm. Source data are available online for this figure.

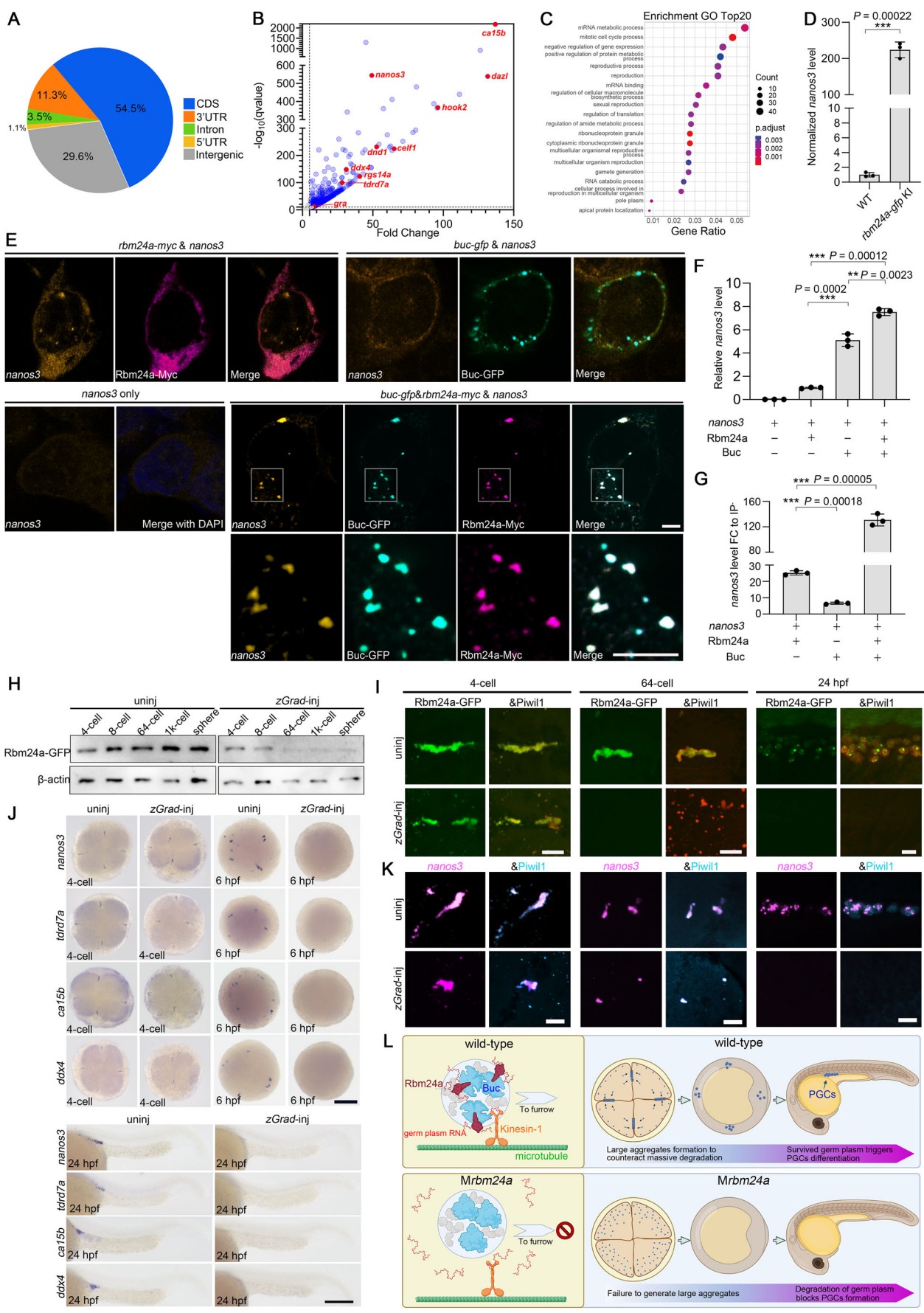

**Figure 6. Rbm24a interacts with germ plasm mRNAs and maintains their stability.**

(A) The category of reads identified by RIP-seq. (B) Scatter plot showing the interaction between Rbm24a and a panel of mRNAs, with red dots highlighting germ plasm mRNAs. (C) GO analysis of Rbm24a-interacting transcripts. The hypergeometric test is employed for statistical assessment, and the Benjamini–Hochberg (BH) method is utilized for $P$ value adjustment. (D) RIP-qPCR verification of Rbm24a binding to *nanos3* mRNA, showing a more than 200-fold enrichment ($n = 3$ independent biological samples). Data are presented as mean ± SD. ****$P < 0.0001$, unpaired Student's $t$ test. (E) Localization analysis of *nanos3* mRNA, Rbm24a-myc and Buc-GFP in HEK293 cells. Note that the simultaneous presence of Buc and Rbm24a efficiently recruits *nanos3* mRNA into phase-separated granules. Scale bars, 10 μm. (F) Coexpression of *nanos3* mRNA with Rbm24a, Buc, or both leads to an enhanced level of *nanos3* transcripts ($n = 3$ independent biological samples). Data are presented as mean ± SD. **$P < 0.01$, ***$P < 0.001$ and ****$P < 0.0001$, unpaired Student's $t$ test. (G), RIP-qPCR in HEK293 cells showing the synergistic effect of Rbm24a and Buc in binding to *nanos3* mRNA ($n = 3$ independent biological samples). Data are presented as mean ± SD. ****$P < 0.0001$, unpaired Student's $t$ test. (H) Temporally regulated degradation of maternal Rbm24a-GFP by *zGrad* mRNA injection. (I) Confocal imaging showing the progressive elimination of Rbm24a-GFP from the 64-cell stage onward. Piwil1 is also absent in zGrad-expressing embryos at 24 hpf, due to the absence of PGCs. Scale bars, 20 μm. (J) Degradation of Rbm24a in homozygous *rbm24a-gfp* KI embryos does not affect the formation of large germ plasm condensates at the 4-cell stage, but causes the absence of germ plasm mRNAs and PGCs in embryos at 6 hpf and 24 hpf. Scale bars, 200 μm. (K) FISH and IF experiments show the gradual disappearance of *nanos3* mRNA and Piwil1 protein following *zGrad* mRNA injection. Notably, the size of *nanos3*- and Piwil1-positive germ granules is remarkably decreased at the 64-cell stage, and germ granules completely disappear by 24 hpf. Scale bars, 20 μm. (L) Working model of maternal Rbm24a function in germ plasm granule assembly. Rbm24a complexed with Buc serves as a reader to recruit and maintain germ plasm mRNAs in the condensates, promoting the formation of large germ plasm aggregates and protecting the stability of germ plasm mRNAs for differentiation of PGCs. Source data are available online for this figure.

(Appendix Fig. 13). We then verified the binding of Rbm24a-GFP to *nanos3* mRNA by RIP-qPCR and impressively observed a 224-fold enrichment (Fig. 6D), suggesting a robust interaction. These results demonstrate that Rbm24a, in complex with Buc, interacts with germ plasm mRNAs.

As HEK293 cells do not express germ plasm factors, we conducted an in vitro reconstitution experiment by expressing *nanos3* mRNA, Rbm24-RFP, and Buc-GFP individually or in combination to investigate the specific recruitment of germ plasm mRNAs by Rbm24a. Interestingly, *nanos3* mRNA was expressed at a remarkably low level when transfected alone, rendering its detection using FISH and qRT-PCR techniques challenging (Fig. 6E,F). However, in the presence of Buc or Rbm24a, the expression of *nanos3* mRNA was significantly enhanced (Fig. 6E,F). Notably, Buc alone exerted limited effect in capturing *nanos3* mRNA into phase-separated condensates (Fig. 6E). In contrast, the coexpression of Buc and Rbm24a efficiently recruited *nanos3* mRNA (Fig. 6E). Remarkably, the presence of Buc facilitated Rbm24a-mediated enrichment of *nanos3* mRNA by 130 folds in HEK293 cells, indicating a strong synergistic effect (Fig. 6G). Therefore, these findings provide compelling evidence that the Rbm24-Buc complex may play a pivotal role in the stabilization of germ plasm mRNAs and their specific sequestration into germ plasm aggregates.

Given the essential function of Rbm24a in germ granule assembly, we further wanted to know if Rbm24a could serve as a target for controlling the fertility of zebrafish. Overexpression experiments were then conducted to assess its ability to increase PGC numbers. Injection of yeast-derived recombinant Rbm24a protein into M*rbm24a* embryos resulted in the emergence of rescued PGCs and the formation of large germ granules, albeit with low efficiency (Fig. EV5A–C). Co-injection of *rbm24a* and *buc* mRNAs into M*rbm24a* embryos promoted PGC formation in 1 among 30 embryos, while separate injections had no effect (Fig. EV5D). Consistent with the rescue experiments, over-expression of Rbm24a, either via mRNA or recombinant protein injection, in a wild-type background also led to the formation of ectopic and larger germ granules, increasing PGC numbers in ~10–25% of embryos (Fig. EV5E–H). The limited PGC rescue or inductive efficiency by exogenous Rbm24a supplementation post fertilization is likely due to the fact that germ granule resembles a solid-phase separation state, which is characterized by extremely slow component exchange (Eno et al, 2019; Kar et al, 2025). Similar to Buc

in zebrafish, Oskar in *Drosophila* also demonstrates the ability to form solid-phase separation (Bose et al, 2022). This suggests that Rbm24a protein may face difficulties in incorporating into the solid-phase-separated germ plasm particles in a timely and efficient manner.

Finally we tested if Rbm24a could be used as a contraceptive target in fish. To specifically target maternal Rbm24a-GFP protein, we injected *zGrad* mRNA into embryos produced by homozygous *rbm24a-GFP* KI females crossed with wild-type males. It has been reported that the zGrad protein functions as an artificial E3 ligase specifically degrading GFP fusion proteins with high efficiency in zebrafish (Yamaguchi et al, 2019). Overexpression of zGrad led to the disappearance of Rbm24a-GFP in the *rbm24a-GFP* KI line, as revealed by western blot analysis and Piwil1- immunofluorescence staining (Fig. 6H,I), suggesting a rapid and almost complete degradation of the fusion protein from the 64-cell stage onward. Piwil1 protein was maintained at the 64-cell stage but was absent at 24 hpf due to the deficiency of PGCs (Fig. 6I). The deleterious effects of this late Rbm24a-GFP degradation on PGC formation were unequivocally confirmed by examining the expression of PGC markers at 4-cell stage, 6 hpf (shield stage), and 24 hpf, which showed the presence of intact germ granules at 4-cell stage but a complete absence of PGCs at 6 hpf and 24 hpf (Fig. 6J). At 64-cell stage, however, the condensation size of both *nanos3* mRNA and Piwil1 protein was reduced following zGrad expression. By 24 hpf, both the mRNA and the protein components had disappeared (Fig. 6K). Therefore, a deficiency of maternal Rbm24a protein after the formation of large germ plasm aggregates also blocks PGC development, highlighting an essential role of Rbm24a protein in the temporal regulation of germ plasm mRNA localization and stability. Thus, targeting maternal Rbm24a protein degradation during early development may offer a promising application for contraceptive purpose in fish.

## Discussion

The preformation of PGCs in a large group of animals depends on maternally derived germline determinants, which are organized as germ plasm and accumulated in specialized subcellular compartments in the embryo. The mechanism underlying germ granule assembly is intriguing and has attracted broad interest. While the

contribution of core granule proteins, such as Oskar, Vasa, Tudor and Aubergine (Aub), to germ granule formation and PGC specification has been well documented in *Drosophila* (Chiappetta et al, 2022; Lehmann, 2016), how RBPs coordinate the aggregation of germ granules remains poorly understand in other preformation species. This study reveals that maternal Rbm24a functions as a key protein component of the germ plasm and plays an essential role for PGC differentiation. Specifically, Rbm24a complexes with Buc to serve as a nucleating organizer of germ plasm particles by recruiting germ plasm mRNAs into phase-separated granules; it is also essential for the kinesin-dependent transport of germ granules to the cleavage furrows. Both functions are critical for stabilizing germ plasm components and maintaining germ cell fate (Fig. 6L).

RNP granules function to regulate RNA metabolism and cell fate (Banani et al, 2017; Gomes and Shorter, 2019). Their formation is dependent on multivalent interactions between RNAs, proteins, and RNAs-proteins (Curnutte et al, 2023; Ripin and Parker, 2023; Tauber et al, 2020). Nevertheless, how germ granule formation is initiated and what is the driving force promoting the aggregation of RNP components remain a subject of debate. If proteins condense first, there must be a recruitment mechanism that enables specific mRNA molecules to integrate into the protein scaffolds that already undergo phase separation. Rbm24a likely functions as a reader of germ plasm mRNAs to recruit them into phase-separated condensates. Upon removal of Rbm24a, germ plasm mRNAs are unable to localize within germ granules and are released to the entire cytoplasm. Although Oskar, Rump and Aub proteins have been reported to mediate the recruitment of mRNAs into germ granules in *Drosophila* (Becalska et al, 2011; Ephrussi and Lehmann, 1992; Jain and Gavis, 2008), the present work provides compelling evidence demonstrating that vertebrate Rbm24 is a novel factor controlling the localization of germ plasm mRNAs in phase-separated structures. What is the underlying mechanism? Rbm24a possesses a single RRM, typically recognizing short RNA sequences of approximately five nucleotides (Qian et al, 2020). It is difficult to imagine how this could confer the specificity for mRNA localization. Based on our observations, it is possible that Rbm24a-containing complex may capture germ plasm mRNAs displaying a particular structure. Indeed, consistent with a previous report (Krishnakumar et al, 2018), we find that Buc binds to *nanos3* mRNA and shows strong synergistic effects with Rbm24a in this interaction. These results suggest that Buc, a scaffold protein, may act as a noncanonical RBP and form a reader complex with Rbm24a to recognize common structural features of germ plasm mRNAs and recruit them into condensates.

Unexpectedly, the formation of phase-separated protein scaffold is independent of Rbm24a. We have provided evidence that protein components, such as Celf1, Piwil1, Tdrd6, NMII-p, and Buc, are still recruited to germ plasm condensates in the absence of Rbm24a. Although the condensation can be governed by Buc, germ plasm mRNA components are clearly indispensable. Therefore, contrary to the notion that mRNAs actively facilitate condensate formation through multivalent interactions (Banani et al, 2017; Guo et al, 2021; Langdon and Gladfelter, 2018), they do not act as the primary driving force for phase separation of the germ plasm (Curnutte et al, 2023). This is supported by the evidence that despite the release of all germ plasm mRNAs due to Rbm24a deficiency, the remaining protein scaffold can still undergo phase separation, although unable to promote PGC development. The fact that Rbm24a and its associated RNAs are dispensable for phase separation of protein scaffolds in germ granules may be of important significance. It could help studying the mechanism controlling other phase-separated events such as stress granules, P-bodies, chromatoid bodies, paraspeckles, etc, in which Rbm24 might play a role (Wang et al, 2022). Another key aspect of Rbm24a function is its requirement for the directed transport of germ granules towards the cleavage furrows, a process previously shown to rely on Kif5Ba protein of the kinesin family (Campbell et al, 2015). Maternal ablation of Kif5ba disrupts the transport process, resulting in the failure of large germ plasm aggregates formation at the cleavage furrows and producing sterile animals that closely resemble M*rbm24a* mutants (Campbell et al, 2015). Injection of a function-blocking antibody against kinesin-1 also recapitulates the phenotype of M*rbm24a* embryos. Therefore, our work establishes a functional link between Rbm24a and kinesin-dependent transport, although the precise mechanism remains unknown. Rbm24a may directly bind to kinesin-1 or exert an allosteric effect on the structure of Buc, thereby promoting its binding to kinesin-1 (Campbell et al, 2015). Another possibility is that germ plasm mRNA components bound by the Buc-Rbm24a complex act as adapters, bridging the condensates with the kinesin motor.

This work also provides crucial evidence supporting the long-standing hypothesis that germ cell fate determination depends on the size of inherited germ plasm aggregates (Eno and Pelegri, 2013). Using *rbm24a-GFP* KI as a tool for live imaging, the dynamics of endogenous germ granules can be recorded at a high resolution, from the formation of initial large germ plasm condensates to the removal of excessive germ granules. It appears that massive degradation of germ granules begins at the 128-cell stage, whatever their size. Therefore, only cleavage furrow-anchored large germ plasm aggregates persist during this degradation window and will be encapsulated into future PGCs to promote germ cell development. This universal degradation process prevents excessive PGC formation, typically by preserving only the four largest aggregates formed during the first two cleavages.

In summary, we have discovered an essential protein component of the germ plasm that functions as an organizer for aggregating germ granules. Rbm24a-Buc complex sequesters germ plasm mRNAs and is essential for loading germ plasm condensate onto the kinesin motor. Thus, Rbm24a may serve as a promising contraceptive target in fish, by utilizing degron-tagging associated with protein degradation approaches. Germ plasm condensates in zebrafish embryos may also serve as an in vivo model in understanding how to aggregate complicated phase-separated granules.

## Methods

**Reagents and tools table**

| Reagent/resource | Reference or source | Catalog number |
| --- | --- | --- |
| **Experimental models** | | |
| zebrafish | AB strain | N/A |
| Human embryonic kidney 293T (HEK293T) cells | Ang Li lab, Shandong University | N/A |
| Sf9 cells | Dalei Wu lab, Shandong University | N/A |

| Reagent/resource | Reference or source | Catalog number |
|---|---|---|
| *Pichia Pastoris* GS115 | Beyotime | D0412 |
| **Recombinant DNA** | | |
| pBluescript SK-rbm24a-GFP KI | This study | N/A |
| pBluescript SK-rbm24a-RFP KIzpc:cas9 | This study | N/A |
| pGGDestEB-rbm24a-4sgRNA | This study | N/A |
| pCS2-buc-GFP | This study | N/A |
| pCS2-rbm24a-GFP | This study | N/A |
| pT2AL200R150G-buc-GFP | This study | N/A |
| pCS2-buc-myc | This study | N/A |
| pCS2-celf1-myc | This study | N/A |
| pCS2-tdrd6-myc | This study | N/A |
| pCS2-piwil1-myc | This study | N/A |
| pCS2-bucΔ1-myc | This study | N/A |
| pCS2-bucΔ2-myc | This study | N/A |
| pCS2-bucΔ3-myc | This study | N/A |
| pCS2-bucΔ4-myc | This study | N/A |
| pCS2-nanos3 | This study | N/A |
| pCS2-rbm24a-RFP | This study | N/A |
| pPIC9K-rbm24a-BFP-6xHis | This study | N/A |
| pPIC9K-buc-GFP-6xHis | This study | N/A |
| pETDuet1-*buc-RFP-6xHis* | This study | N/A |
| pFastBac-HTA2-*rbm24a-BFP-6xhis* | This study | N/A |
| pFastBac-HTA2-buc-GFP-6xHis | This study | N/A |
| pCS2-*rbm24a*ΔRRM-GFP | Shao et al, 2020 | N/A |
| pCS2-*rbm24a*ΔC-GFP | Shao et al, 2020 | N/A |
| **Antibodies** | | |
| Mouse monoclonal anti-GFP | Roche | Cat#REF11814461001 |
| Mouse monoclonal anti-Myc | Abcam | Cat#ab32 |
| Mouse anti-Kinesin,heavy chain | Sigma-Aldrich | Cat#MAB1614 |
| Rabbit monoclonal anti-Piwil1 | ABclonal | Raised against the synthetic peptide EGQLVGRGRQKPAPGC |
| Rabbit anti-Phospho-Myosin Light Chain 2 (Ser19) | Cell signaling | Cat#3671 |
| Rabbit mCherry Polyclonal antibody | Proteintech | Cat#26765-1-AP |
| Mouse Monoclonal anti-His Tag | Beyotime | Cat#AF2876 |
| Goat anti-mouse IgG HRP-linked | zenbio | Cat#550125 |
| Goat anti-rabbit IgG HRP-linked | Boster-Bio | Cat#BA1054 |
| Goat anti-mouse IgG Alexa Fluor488 | ZSGB-BIO | Cat#ZF-0512 |
| Goat anti-rabbit IgG Alexa Fluor 594 | ZSGB-BIO | Cat#ZF-0516 |
| Goat anti-mouse IgG Alexa Fluor 647 | Bioworlde | Cat#BS21846 |
| Goat anti-rabbit IgG Alexa Fluor 647 | Bioworlde | Cat#BS21849 |
| Anti-Digoxigenin-AP | Roche | Cat#11093274910 |
| Anti-Fluorescein-POD | Roche | Cat#11426346910 |
| Anti-Digoxigenin-POD | Roche | Cat#11633716001 |
| **Oligonucleotides and other sequence-based reagents** | | |
| Primers and DNA sequences | This study | Apendix table S2 |
| Fluorescent probe *nanos3* 1 | 5'-CGAGCGCGCGATCAAAACAA-3' Quasar ®570 | Biosearch Technologies |
| Fluorescent probe *nanos3* 2 | 5'-AGAAAAAGCCATGTTGCTCT-3' Quasar ®570 | Biosearch Technologies |
| Fluorescent probe *nanos3* 3 | 5'-TCCATGAGCAGAAAGGATGA-3' Quasar ®570 | Biosearch Technologies |
| Fluorescent probe *nanos3* 4 | 5'-CTGGTTTCTAGTCTCCATAG-3' Quasar ®570 | Biosearch Technologies |
| Fluorescent probe *nanos3* 5 | 5'-TAGTCCTTCCAAGGCTGAAA-3' Quasar ®570 | Biosearch Technologies |
| Fluorescent probe *nanos3* 6 | 5'-TTCATGCCTCTGATCATGTC-3' Quasar ®570 | Biosearch Technologies |
| Fluorescent probe *nanos3* 7 | 5'-CGTCAGATTGCATTTCTTGC-3' Quasar ®570 | Biosearch Technologies |
| Fluorescent probe *nanos3* 8 | 5'-AGAACTTTCTCTCTGCTGGG-3' Quasar ®570 | Biosearch Technologies |
| Fluorescent probe *nanos3* 9 | 5'-CGTTGTGTTTGCAGAAGCTG-3' Quasar ®570 | Biosearch Technologies |
| Fluorescent probe *nanos3* 10 | 5'-TAGTGAGAGGTGTACACGGC-3' Quasar ®570 | Biosearch Technologies |
| Fluorescent probe *nanos3* 11 | 5'-TCTCCATCGCGGTTTTTTAA-3' Quasar ®570 | Biosearch Technologies |
| Fluorescent probe *nanos3* 12 | 5'-CGGAGATACGGGCACATCAC-3' Quasar ®570 | Biosearch Technologies |
| Fluorescent probe *nanos3* 13 | 5'-CACACAGGGGACACTTGTAC-3' Quasar ®570 | Biosearch Technologies |
| Fluorescent probe *nanos3* 14 | 5'-CATCGGGCAGAATCTCTTGG-3' Quasar ®570 | Biosearch Technologies |
| Fluorescent probe *nanos3* 15 | 5'-TACACCGAGCAGTAGTTCTT-3' Quasar ®570 | Biosearch Technologies |
| Fluorescent probe *nanos3* 16 | 5'-CGCTTCACCATGTTGATTTG-3' Quasar ®570 | Biosearch Technologies |
| Fluorescent probe *nanos3* 17 | 5'-TCAAATCTACCGGAGCATCA-3' Quasar ®570 | Biosearch Technologies |
| Fluorescent probe *nanos3* 18 | 5'-CCTGCGGTAAAAAGTGTTTC-3' Quasar ®570 | Biosearch Technologies |
| Fluorescent probe *nanos3* 19 | 5'-CCGCGTATCAACCAAACAAA-3' Quasar ®570 | Biosearch Technologies |

| Reagent/resource | Reference or source | Catalog number |
|---|---|---|
| Fluorescent probe *nanos3* 20 | 5'-TGCATGCAAACTCGCAATCC-3' Quasar ®570 | Biosearch Technologies |
| Fluorescent probe *nanos3* 21 | 5'-ATCAAACAGTGAACGCACAC-3' Quasar ®570 | Biosearch Technologies |
| Fluorescent probe *nanos3* 22 | 5'-CACACATATACACACACACA-3' Quasar ®570 | Biosearch Technologies |
| Fluorescent probe *nanos3* 23 | 5'-AACACAACACCAGTGCACAC-3' Quasar ®570 | Biosearch Technologies |
| Fluorescent probe *nanos3* 24 | 5'-GGCTTGTGTACAAGTTTGTT-3' Quasar ®570 | Biosearch Technologies |
| Fluorescent probe *nanos3* 25 | 5'-TTTGTTTTTGAGTGCGGTTG-3' Quasar ®570 | Biosearch Technologies |
| Fluorescent probe *nanos3* 26 | 5'-GCGACATGAAAATATGGCGA-3' Quasar ®570 | Biosearch Technologies |
| Fluorescent probe *nanos3* 27 | 5'-TTCACTCCATCACAAGTCAA-3' Quasar ®570 | Biosearch Technologies |
| Fluorescent probe *vasa* 1 | 5'-TGCAAGACACAACGGGACTC-3'CAL Fluor Red 610 | Biosearch Technologies |
| Fluorescent probe *vasa* 2 | 5'-ATCAGAACCATTTGAGCCTA-3'CAL Fluor Red 610 | Biosearch Technologies |
| Fluorescent probe *vasa* 3 | 5'-ACCTCCTGTATAAAAGCTCT-3'CAL Fluor Red 610 | Biosearch Technologies |
| Fluorescent probe *vasa* 4 | 5'-CTGTTTGACTTGTCATTTCC-3'CAL Fluor Red 610 | Biosearch Technologies |
| Fluorescent probe *vasa* 5 | 5'-AATCACCAGTCATTTTCCAT-3'CAL Fluor Red 610 | Biosearch Technologies |
| Fluorescent probe *vasa* 6 | 5'-TTAAAACCGCTAAAGCCTCC-3'CAL Fluor Red 610 | Biosearch Technologies |
| Fluorescent probe *vasa* 7 | 5'-GCCATTCTCATCAATTTCTG-3'CAL Fluor Red 610 | Biosearch Technologies |
| Fluorescent probe *vasa* 8 | 5'-CCATCATCATTTCCATTTTC-3'CAL Fluor Red 610 | Biosearch Technologies |
| Fluorescent probe *vasa* 9 | 5'-TCACGGAAACCTCCACGAAA-3'CAL Fluor Red 610 | Biosearch Technologies |
| Fluorescent probe *vasa* 10 | 5'-CTCATCGTTTCCATTTTCAT-3'CAL Fluor Red 610 | Biosearch Technologies |
| Fluorescent probe *vasa* 11 | 5'-GCGGCACATAAACAACCTTG-3'CAL Fluor Red 610 | Biosearch Technologies |
| Fluorescent probe *vasa* 12 | 5'-AAGTCCTGCTTCCTCAAAAG-3'CAL Fluor Red 610 | Biosearch Technologies |
| Fluorescent probe *vasa* 13 | 5'-CATTTTTGCTCAGTGAGTCA-3'CAL Fluor Red 610 | Biosearch Technologies |
| Fluorescent probe *vasa* 14 | 5'-CTGAGCACAAGCCATTAGAT-3'CAL Fluor Red 610 | Biosearch Technologies |
| Fluorescent probe *vasa* 15 | 5'-TAAAGCGCTGTAGGATAGGC-3'CAL Fluor Red 610 | Biosearch Technologies |

| Reagent/resource | Reference or source | Catalog number |
|---|---|---|
| Fluorescent probe *vasa* 16 | 5'-TGATAAGTTCTCTGGTGGGA-3'CAL Fluor Red 610 | Biosearch Technologies |
| Fluorescent probe *vasa* 17 | 5'-CCAGTATTTATACCTCCATA-3'CAL Fluor Red 610 | Biosearch Technologies |
| Fluorescent probe *vasa* 18 | 5'-ATGCAATCTTCCAGGAGTAG-3'CAL Fluor Red 610 | Biosearch Technologies |
| Fluorescent probe *vasa* 19 | 5'-ATCTTTCCACGACCAATGAG-3'CAL Fluor Red 610 | Biosearch Technologies |
| Fluorescent probe *vasa* 20 | 5'-TCTGCTTCATCCAGAACTAG-3'CAL Fluor Red 610 | Biosearch Technologies |
| Fluorescent probe *vasa* 21 | 5'-TCTTTTGAAGGCATACCAGG-3'CAL Fluor Red 610 | Biosearch Technologies |
| Fluorescent probe *vasa* 22 | 5'-CAGCAGCCATTCTTTGAATA-3'CAL Fluor Red 610 | Biosearch Technologies |
| Fluorescent probe *vasa* 23 | 5'-CCACACCAACAGCAAGGAAA-3'CAL Fluor Red 610 | Biosearch Technologies |
| Fluorescent probe *vasa* 24 | 5'-ACCTGAACAATGGTTTGCTC-3'CAL Fluor Red 610 | Biosearch Technologies |
| Fluorescent probe *vasa* 25 | 5'-CTGAGCAATTCAAGCAGCTG-3'CAL Fluor Red 610 | Biosearch Technologies |
| Fluorescent probe *vasa* 26 | 5'-AAACCATTGTGCGCTCATTA-3'CAL Fluor Red 610 | Biosearch Technologies |
| Fluorescent probe *vasa* 27 | 5'-ATCAGCACTTCTTTTGGTTT-3'CAL Fluor Red 610 | Biosearch Technologies |
| Fluorescent probe *vasa* 28 | 5'-TGTGGTTGAGATCTTCTCTT-3'CAL Fluor Red 610 | Biosearch Technologies |
| Fluorescent probe *vasa* 29 | 5'-GCCAAGGCGAAAATCACTGA-3'CAL Fluor Red 610 | Biosearch Technologies |
| Fluorescent probe *vasa* 30 | 5'-TGCTGGACTTGCTCAATATC-3'CAL Fluor Red 610 | Biosearch Technologies |
| Fluorescent probe *vasa* 31 | 5'-GCTGGGCATGTCAAAATTCA-3'CAL Fluor Red 610 | Biosearch Technologies |
| Fluorescent probe *vasa* 32 | 5'-TGCGATGGACATACTCATCG-3'CAL Fluor Red 610 | Biosearch Technologies |
| Fluorescent probe *vasa* 33 | 5'-CGAGCTAATGGAGTGTCAGA-3'CAL Fluor Red 610 | Biosearch Technologies |
| Fluorescent probe *vasa* 34 | 5'-CATGAGCACTGAAGGCAACT-3'CAL Fluor Red 610 | Biosearch Technologies |
| Fluorescent probe *vasa* 35 | 5'-CGCGAGTCTGTAGATGCAAA-3'CAL Fluor Red 610 | Biosearch Technologies |
| Fluorescent probe *dazl* 1 | 5'-AGATGCACATAATCGACCGC-3'Quasar ®570 | Biosearch Technologies |
| Fluorescent probe *dazl* 2 | 5'-ACGAATATTTATGCTCTCCT-3'Quasar ®570 | Biosearch Technologies |
| Fluorescent probe *dazl* 3 | 5'-GCTGGGCATTTAGGGTAAAA-3'Quasar ®570 | Biosearch Technologies |

| Reagent/resource | Reference or source | Catalog number |
|---|---|---|
| Fluorescent probe dazl 4 | 5'-ATCGTGTTTCTTCATTAGGC-3'Quasar ®570 | Biosearch Technologies |
| Fluorescent probe dazl 5 | 5'-AAACCGGATAGATCCCGAAC-3'Quasar ®570 | Biosearch Technologies |
| Fluorescent probe dazl 6 | 5'-AAATGCTCAATCCACACGCG-3'Quasar ®570 | Biosearch Technologies |
| Fluorescent probe dazl 7 | 5'-AAACAAACCCCAAAGTCCGT-3'Quasar ®570 | Biosearch Technologies |
| Fluorescent probe dazl 8 | 5'-GTTTTGCTACCTAAGCTTTT-3'Quasar ®570 | Biosearch Technologies |
| Fluorescent probe dazl 9 | 5'-CCCTGAACCATTTTGAATGA-3'Quasar ®570 | Biosearch Technologies |
| Fluorescent probe dazl 10 | 5'-ATATCAGGCACACGGGTAAC-3'Quasar ®570 | Biosearch Technologies |
| Fluorescent probe dazl 11 | 5'-CTGGATATCCTGTGAATACA-3'Quasar ®570 | Biosearch Technologies |
| Fluorescent probe dazl 12 | 5'-GACGGAAAACCCTGACGATG-3'Quasar ®570 | Biosearch Technologies |
| Fluorescent probe dazl 13 | 5'-ACCGTTAGACAACTTCAGGG-3'Quasar ®570 | Biosearch Technologies |
| Fluorescent probe dazl 14 | 5'-TTTTCCCCTCAGGTAAAATG-3'Quasar ®570 | Biosearch Technologies |
| Fluorescent probe dazl 15 | 5'-GACGAACAGTGTGTTGGGCG-3'Quasar ®570 | Biosearch Technologies |
| Fluorescent probe dazl 16 | 5'-TCCACCTTCATATCAATACC-3'Quasar ®570 | Biosearch Technologies |
| Fluorescent probe dazl 17 | 5'-AAGAATTCCCTGATCTCGTT-3'Quasar ®570 | Biosearch Technologies |
| Fluorescent probe dazl 18 | 5'-TAACTTCTTTCACTGAGCCA-3'Quasar ®570 | Biosearch Technologies |
| Fluorescent probe dazl 19 | 5'-TTCCTCCTCGATAAGTGATG-3'Quasar ®570 | Biosearch Technologies |
| Fluorescent probe dazl 20 | 5'-ACGAAACCATATCCTTTGCA-3'Quasar ®570 | Biosearch Technologies |
| Fluorescent probe dazl 21 | 5'-GTCTGGATATCAACATCCTC-3'Quasar ®570 | Biosearch Technologies |
| Fluorescent probe dazl 22 | 5'-ACTGATCGGCTGATCAACGA-3'Quasar ®570 | Biosearch Technologies |
| Fluorescent probe dazl 23 | 5'-GTTTGAGTTTTTTCCCTTTA-3'Quasar ®570 | Biosearch Technologies |
| Fluorescent probe dazl 24 | 5'-CTCTTTCATGATTGCAGGTC-3'Quasar ®570 | Biosearch Technologies |
| Fluorescent probe dazl 25 | 5'-GATGACACTGACCGAGAACT-3'Quasar ®570 | Biosearch Technologies |
| Fluorescent probe dazl 26 | 5'-CACTGTGATGGACCAATCAT-3'Quasar ®570 | Biosearch Technologies |
| Fluorescent probe dazl 27 | 5'-GCAGCTGCAGTACATATATG-3'Quasar ®570 | Biosearch Technologies |
| Fluorescent probe dazl 28 | 5'-AAATACAGGTGATGGTGGGG-3'Quasar ®570 | Biosearch Technologies |
| Fluorescent probe dazl 29 | 5'-AAGGCTGCATGTACTGATTT-3'Quasar ®570 | Biosearch Technologies |
| Fluorescent probe dazl 30 | 5'-CCTGGAGGACTGGAGTAAGA-3'Quasar ®570 | Biosearch Technologies |
| Fluorescent probe dazl 31 | 5'-TGGCACCTGTGGAACCATAA-3'Quasar ®570 | Biosearch Technologies |
| Fluorescent probe dazl 32 | 5'-CATACGTGGTCTGTGCATAG-3'Quasar ®570 | Biosearch Technologies |
| Fluorescent probe dazl 33 | 5'-TGTGGCAGGGGATACTGATA-3'Quasar ®570 | Biosearch Technologies |
| Fluorescent probe dazl 34 | 5'-ATTCTGATTGACAAGCCTCG-3'Quasar ®570 | Biosearch Technologies |
| Fluorescent probe dazl 35 | 5'-AAGGGTTAGCAAAGTCTGCA-3'Quasar ®570 | Biosearch Technologies |
| **Chemicals, enzymes, and other reagents** | | |
| Ni Bestarose FF(Fast Flow) | Bestchrom | Cat#AA0051 |
| DMEM | TransGen | Cat#FI101-01 |
| Closed sheep serum (liquid) | Solarbio | Cat#SL039 |
| Coomassie Brilliant BlueG250 | Fluka | Cat#13628 |
| Fetal Bovine Serum | Gibco | Cat#A5256701 |
| Polyethylenimine Linear | Yeasen | Cat#40816ES02 |
| Cap analog | NEB | Cat#S1407S |
| SP6 RNA polymerase | NEB | Cat#M0207S |
| T7 RNA polymerase | NEB | Cat#M0251L |
| rNTP | Thermo Fisher Scientific | Cat#R0481 |
| RiboLock RNase Inhibitor | Thermo Fisher Scientific | Cat#EO0381 |
| Phusion™ High–Fidelity DNA Polymerase | Thermo Fisher Scientific | Cat#F530L |
| 2X M5 HiPer Taq PCR mix | Mei5bio | Cat#MF001-BD-100 |
| DNase I | Roche | Cat#4716728001 |
| Silicon hydroxyl magnetic beads | Sangon Biotech | Cat#B518720-0001 |
| NLS-Cas9 Nuclease | Self made | NA |
| DIG-labeling mix | Roche | Cat#11277073910 |
| Fluorescein RNA labeling mix | Roche | Cat#11685619910 |
| Low-melting agarose | Biotech | Cat#CA1351 |
| Ribonucleoside Vanadyl Complex | Beyotime | Cat#R0107 |
| Dextran sulfate | Sangon Biotech | Cat#A600160-0010 |
| GFP-Trap Magnetic Agarose beads | ChromoTek | Cat#gtma |
| Dynabeads Protein G beads | Invitrogen | Cat#10004D |

| Reagent/resource | Reference or source | Catalog number |
|---|---|---|
| Protein G agarose beads | Sigma | Cat#P7700 |
| Protease inhibitor cocktail | Roche | Cat#40609 |
| GFP-Trap Agarose beads | ChromoTek | Cat#gta-20 |
| Proteinase K | Roche | Cat#03115828001 |
| Murine RNase Inhibitor | Vazyme | Cat#R301-03-AA |
| pPIC9K Vector | Thermo Fisher Scientific | Cat#V175-20 |
| Bac-to-Bac™ Baculovirus Expression System | Thermo Fisher Scientific | Cat#CN10359016 |
| DH10Bac Super Competent Cells | Beyotime | Cat#D1029M |
| Cellfectin™ II | Thermo Fisher Scientific | Cat#CN10362100 |
| IgG | Beyotime | Cat#A7016 |
| IgG | Abcam | Cat#ab18413 |
| IPTG | Beyotime | Cat#ST098 |
| Rosetta(DE3) | Beyotime | Cat#D1065S |
| PMSF | Coolaber | Cat#CP8651 |
| DTT | Thermo Fisher Scientific | Cat#R0861 |
| Insect Cell Culture Medium | Union-biotech | Cat#UK1000 |
| D-Sorbitol | Dingguo | Cat#DH317-2 |
| Oxoid™ Tryptone | Thermo Fisher Scientific | Cat#LP0042B |
| Oxoid™ Yeast Extract Powder | Thermo Fisher Scientific | Cat#LP0021B |
| Sterile Syringe Filter 0.22μm | Biosharp | Cat#BS-PES-22 |
| YPD Broth | Sangon Biotech | Cat#A507022-0250 |
| Geneticin G-418 | Coolaber | Cat#CG5471 |
| X-gal | Beyotime | Cat#ST912 |
| Gentamicin | Shyuanye | Cat#T21508 |
| Kanamycin sulfate | Sangon | Cat#A430277-0200 |
| Tetracyclin | Solarbio | Cat#64-75-5 |
| Aprotinin | Roche | Cat#10236624001 |
| Leupeptin | Sigma | Cat#L2884 |
| **Critical commercial assays** | | |
| TSA Individual Cyanine 3 Tyramide Reagent Pack Kit | PerkinElmer | Cat#SAT704B001EA |
| AxyPrep Plasmid Miniprep Kit | Axygen | Cat#AP-MN-P-4 |
| Fast SilverStain Kit | Beyotime | Cat#P0017S |
| Bacmid Miniprep Kit | Beyotime | Cat#D0031 |
| One-Step gDNA Removal and cDNA Synthesis SuperMix Kit | TransScript | Cat#AT311-02 |
| **Software** | | |
| ImageJ | Open source | https://imagej.net/software/fiji/ |
| GraphPad Prism 9 | GraphPad | https://www.graphpad.com/ |
| Zeiss Zen Blue | Zeiss | N/A |
| LAS X | Leica | N/A |
| OlyVIA | OLYMPUS | N/A |

| Reagent/resource | Reference or source | Catalog number |
|---|---|---|
| Code for scRNA-seq and RIP-seq bioinformatic analysis | This study | https://github.com/benjaminfang/rbm24-workflow https://github.com/jiangliufengzi/rip-seq-data-analysis |
| **Other** | | |
| Illumina HiSeq 2000 | Illumina | Azenta |
| Illumina NextSeq 500 | Illumina | Gene Denovo |

## Ethical statement

Animal experiments were performed following the ARRIVE guidelines and approved by the Ethics Committee for Animal Research of Life Science of Shandong University (permit number SYDWLL-2021-15).

## Zebrafish and microinjection

Wild-type zebrafish of the AB strain and transgenic or mutant lines were raised in standard husbandry systems (Haisheng) at 28 °C. Microinjection was performed using a PLI-100A Picoliter Microinjector (Harward Apparatus), and a volume of 1–2 nL was injected into each embryo at the 1-cell stage.

## Cell culture and transfection

Human embryonic kidney 293 T (HEK293T) cells were cultured in DMEM (TransGen, FI101-01) supplemented with 10% fetal bovine serum (FBS) at 37 °C under a humidified atmosphere consisting of 5% $CO_2$. Transfection of plasmids was performed when cell coverage reached around 70%, using Polyethylenimine Linear (Yeasen, 40816ES02) according to the manufacturer's instructions. Cells cultured in a 90 mm petri dish were transfected with 10 μg of pCS2-*buc-GFP*, 250 ng of pCS2-*rbm24a-myc* or 250 ng of pCS2-*nanos3*, while the amount of each plasmid was reduced to one-fifth for transfection of cells cultured in a six-well plate.

## Plasmid construction

Most constructs were generated following a previously reported TEDA method (Xia et al, 2019). To obtain the knock-in donor plasmid for *rbm24a-GFP* KI, a 1704 bp DNA fragment preceding the *rbm24a* stop codon was fused to the GFP coding sequence, which was ligated with a 2274 bp genomic sequence after the stop codon of *rbm24a*. The whole fragment was cloned into the pBluescript SK- vector. The knock-in donor plasmid for *rbm24a-RFP* KI$^{zpc:cas9}$ was constructed by replacing the GFP coding sequence in *rbm24a-GFP* KI plasmid with *RFP*, and inserting the *zpc:cas9* cassette downstream of the *rbm24a* 3'-UTR. The plasmid for creating Tg(*ef1α:buc-GFP*) was constructed by cloning the coding sequence of zebrafish *buc* into the BamHI site of pT2AL200R150G (Urasaki et al, 2006), ensuring inframe ligation of *buc* coding sequence with *GFP*. To construct plasmids used for in vitro mRNA transcription, the coding regions of genes of interest were cloned upstream of and inframe with the 6× myc tags or the GFP sequence in the pCS2 vector. PCR amplification and Gibson Ligation were

used to create pCS2-based plasmids containing truncated variants of *rbm24a* and *buc*. The pGGDestEB-*rbm24a*-4sgRNA plasmid for maternal genome editing was constructed by Golden Gate assembly as described (Xia et al, 2019).

## Synthesis of capped mRNAs

Recombinant vectors were linearized by NotI digestion. After purification, the DNA templates were added to the mRNA synthesis mix containing 0.5 mM cap analog (NEB, S1407S), 0.5 mM ATP, 0.5 mM CTP, 0.5 mM UTP, 0.1 mM GTP, 20 U SP6 RNA polymerase (NEB, M0207S), 20 U RNA inhibitor (Thermo Scientific, EO0381), and 1× RNA polymerase buffer. The mixture was incubated at 37 °C for 1.5 h. Subsequently, the DNA templates were removed by DNase I treatment (Roche, 4716728001) at 37 °C for 15 minutes. Transcribed mRNAs were purified using silicon hydroxyl magnetic beads (B518720-0001) (Oberacker et al, 2019) and resuspended in 15–20 μL RNase-free water.

## Knock-in

The sgRNAs were designed using the online software CRISPRScan and synthesized by in vitro transcription (Moreno-Mateos et al, 2015). A mixture of 6.7 pg *rbm24a-GFP* knock-in plasmid, 100 pg sgRNA, and 600 ng Cas9 protein was injected into the blastodisc of each embryo at the 1-cell stage. At 30 hpf, embryos with GFP signals in the lens, heart, skeletal muscle, and hair cells were picked out as early integration founders for breeding. PCR and sequencing analyses were performed to confirm the successful generation of the *rbm24a-GFP* knock-in strain. The *rbm24a-RFP* KI$^{zpc:cas9}$ line was generated by following a similar approach.

## Conditional knockout of maternal genes

Four sgRNAs targeting exons 1 and 3 of *rbm24a* were used to construct the sgRNA expression vector, pGGDestEB-*rbm24a*-4sgRNA, which also contains the sgRNA expression elements and a BFP expression cassette driven by the *ef1a* promoter. The plasmid and *Tol2 transposase* mRNA were coinjected into homozygous *rbm24a-RFP* KI$^{zpc:cas9}$ embryos at the 1-cell stage to produce F0 founders carrying the transgene. Embryos spawned by the F0 founders with BFP but not Rbm24a-RFP expression are maternal *rbm24a* mutants (M*rbm24a*). Maternal *Buc* mutants were obtained by injecting four highly efficient sgRNAs with Cas9 protein into *rbm24a-GFP* KI embryos, followed by phenotypic screening in the offspring. Cas9 protein was produced through expression in *E. coli* in our lab following an established protocol (Bhoir et al, 2018).

To accurately assess the editing efficiency, the remaining wild-type mRNAs were estimated instead of analyzing mutation rate. It would be difficult to quantify the ratio of mutant mRNAs or genomic lesions, because oocyte-specific genome editing may generate mutant mRNAs undergoing nonsense-mediated decay or unintended large genomic deletions without primer binding sites (Zhang et al, 2021). In contrast, measuring the intact wild-type mRNAs by qRT-PCR is much more feasible. As such, a two-step protocol was designed. The first step was to perform a qRT-PCR analysis employing primers flanking the four sgRNA targeting sites. The amplification products should represent wild-type mRNAs and

mutant mRNAs that can be amplified during the RT-PCR reaction, and their levels relative to KI siblings were designated as ratio1. The next step was to subject PCR products to high-resolution capillary electrophoresis. The ratio of PCR products representing wild-type mRNAs (ratio2) can be obtained based on the electropherogram. Hence, the overall editing efficiency was calculated as $1 -$ ratio1 × ratio2.

## Generation of *buc-GFP* transgenic lines

A mixture of 20 pg *buc-GFP* transgenic plasmid in which *ef1α* promoter and *SV40* 3'-UTR were used to support transgene expression and 100 pg *Tol2 transposase* mRNA was microinjected into 1-cell stage embryos. After injection, F0 founders were raised to adulthood, and F1 embryos from F0 females were screened by assessing specific fluorescent signals in the PGCs.

## ISH

It was performed following previously established protocols (Shao et al, 2020). DNA templates for the synthesis of *rbm24a*, *nanos3*, *dnd1*, *ca15b*, *kop*, *tdrd7a* and *ddx4* probes were amplified by PCR using primers listed in Appendix Table S1 and cloned into the pZeroback vector. Digoxigenin-labeled probes were synthesized by in vitro transcription using the DIG-labeling mix (Roche, 11277073910) and SP6 or T7 RNA polymerase. Following DNase I treatment (Roche, 4716728001), probes were directly precipitated with 50% (v/v) isopropanol and resuspended in the hybridization buffer.

## Confocal microscopy and light sheet microscopy

Embryos were embedded in 1% low-melting agarose (Biotech, CA1351), and confocal images were acquired using Olympus SpinSR10, ZEISS LSM 900 with Airyscan 2, and Leica STELLARIS 5. For time-lapse imaging, frames were captured at a 25-s interval, and Z-stacks were acquired using a ×10, ×20, or ×40 objective lens, with step intervals of less than 2.0 μm. Light sheet microscopy images were obtained using the LUXENDO MuVi-SPIM microscope.

## FISH and smFISH

Zebrafish embryos at the cleavage stage or 24 hpf were fixed in 4% paraformaldehyde (PFA) at 4 °C overnight. They were dehydrated using a methanol gradient and stored in absolute methanol at −20 °C overnight. Subsequently, the embryos were rehydrated in PBS containing 0.1% Tween-20 and treated according to the standard ISH methods. Anti-digoxigenin-POD antibody (1:1000; Roche, 11633716001) and the TSA Individual Cyanine 3 Tyramide Reagent Pack kit (PerkinElmer, SAT704B001EA) were used for fluorescence staining to visualize mRNA distribution.

HEK293T cells grown on glass slides and transfected with plasmids were briefly washed three times with PBS and fixed in 4% PFA at 4 °C for 10 minutes. They were then washed with PBS twice and permeabilized with 0.2% Triton X-100 in PBS at 4 °C for 10 minutes. The cells were incubated with prehybridization buffer (2× SSC, 10% formamide, 0.1% Triton X-100, 2 mM Ribonucleoside Vanadyl Complex, and 0.02% bovine serum albumin) at room temperature for 5 minutes, followed by incubation in hybridization

solution (prehybridization buffer with 10% dextran sulfate) containing *nanos3* fluorescent probes (Appendix Table S1) at 30 °C overnight. After removal of the hybridization solution and washing ten times with wash buffer (0.1% Triton X-100, 2× SSC, and 10% formamide), twice with 2× SSC, and twice with PBSTr (0.1% Triton X-100 in PBS), the cells were blocked with 10% sheep serum at 30 °C for 30 minutes and incubated with anti-myc antibody (1:100; Abcam, ab32) at 30 °C for 1 h. After washing four times with PBSTr, incubation of anti-mouse IgG(H&L)-AF647 secondary antibody (1:100; Bioworlde, BS21846) was performed at 30 °C for 1 h. The glass slides were then washed three times and prepared for confocal imaging analysis.

Oocytes or embryos expressing Rbm24a-GFP were fixed in 4% PFA at 4 °C overnight, and smFISH experiments were carried out using the methods described above. The sequences of smFISH probes targeting *nanos3* and *dazl* are listed in Appendix Table S1.

## Image analysis

Maximum Intensity Projection (MIP) was performed using OlyVIA software. Size measurements, density calculations, and analyses of movement trajectories were conducted using ImageJ software, accessible at https://fiji.sc/. Frame sequences were imported into ImageJ using the "Import Image Sequence" function, and images were converted to 8-bit format using the "Image-Type-8 bit" option. Scale calibration was performed using the "Analyze-Set Scale" function. To enhance visualization, brightness and contrast adjustments were applied to the images based on regions of interest (ROIs). Background removal was achieved by adjusting the threshold via the "Image-Adjust-Threshold" function, enabling the calculation of germ granule size and density. Subsequently, measurements were conducted using the "Analyze-Analyze Particles" function. Manual tracking was performed using the "Plugins-Tracking-Manual Track" tool to track the trajectories of germ granules.

## Western blot

Protein samples were separated on SDS-PAGE and transferred to the nitrocellulose membrane. Primary antibodies are the following: anti-GFP antibody (1:5000; Roche, 11814461001), anti-myc antibody (1:5000; Abcam, ab32), anti-β-actin antibody (1:5000; Proteintech, 66009-1-Ig), and anti-Piwil antibody (1:5000; ABclonal, produced in rabbit against the synthetic peptide EGQLVGRGRQKPAPGC) (Houwing et al, 2007). Protein bands were visualized using the Western-Lightning Plus-ECL substrate (PerkinElmer) after incubation with horseradish peroxidase-conjugated secondary antibody (1:10,000; Zenbio, 550125).

## Co-IP

GFP-Trap Magnetic Agarose beads (ChromoTek, gtma) and uncross-slinked anti-GFP antibody beads (anti-GFP antibody incubated with protein G Dynabeads overnight) were used for immunoprecipitation experiments. They were washed thrice and stored at 4 °C in the blocking buffer (0.5% bovine serum albumin in PBS) before use. Synthetic mRNAs (100 pg) were injected into 1-cell stage embryos, and ~50 injected embryos were collected around 10 hpf. After adding 250 μL of lysis buffer containing 10 mM Tris-HCl, pH 7.5, 150 mM NaCl, 0.5 mM EDTA, 0.5% NP-40, and 1× protease inhibitor cocktail

(Roche, 40609), the embryos were homogenized using a grinding stick on ice for 10 minutes, followed by centrifugation at $12,000 \times g$ for 10 minutes at 4 °C. After saving 25 μL as input control, the supernatant was incubated with 10 μL of beads at 4 °C overnight. The beads were washed five times with lysis buffer, heated at 95 °C for 5 minutes, and subjected to western blot analysis.

## scRNA-seq and bioinformatic analysis

M*rbm24a* and sibling single cells were isolated, barcoded, and sequenced separately. The experimental procedure and bioinformatics analysis were largely conducted according to previously established methods (Wagner et al, 2018). Specifically, thirty-three M*rbm24a* or wild-type embryos at the shield stage were dissociated into cell suspensions, which were loaded onto a 10x Genomics GemCode Single-cell instrument to generate single-cell Gel Bead-In-EMulsion (GEMs). Construction of cDNA libraries was performed using the Chromium Next GEM Single Cell 3' Reagent Kits v3.1 by following the manufacturer's instructions.

Deep sequencing of each sample yielded two fastq data files, referred to as R1 and R2. Reads in the R2 file were filtered using Trimmomatic with the argument "AVGQUAL:30". Read mapping and UMI counting were handled by the Cellranger analysis piplines, and the Cellranger-filtered UMI count data were utilized in downstream analyses (Zheng et al, 2017).

DecontX was employed to filter ambient RNA contamination and obtain corrected count numbers. In addition, doublets were recognized and filtered by Scrublet. Subsequently, the high-quality barcodes were recovered based on the sum of the counts, the number of genes, and the percentage of counts contributed by the top 20 genes.

The MAD (Median Absolute Deviation) approach was used to determine the outlier of barcodes. The MADs and medians were used to calculate the upper and lower boundaries. The fivefold value of MAD was added to the median to obtain the upper boundary, while the fivefold of the MAD was subtracted from the median to define the lower boundary. Barcodes that fall outside the range between the boundaries were excluded. The filtered barcodes from all samples (experimental and control) were then integrated to create the count matrix table.

Scanpy was used to select two thousand highly variable genes (HVGs), excluding mitochondrial genes (*mt-rnr1*, *mt-rnr2*, *ND1*, *ND2*, *COX1*, *COX2*, *ATP8*, *ATP6*, *COX3*, *ND3*, *ND4L*, *ND4*, *ND5*, *ND6*, *CYTB*) or genes without correlation to other genes (defined as "correlation coefficient" >0.2) (Wolf et al, 2018). A set of cell cycle and housekeeping-associated genes (*cdk1*, *mcm2*, *mcm7*, *rrm2*, *cenpa*, *cdc6*, *ccnf*, *cdca4*, *ccnd1*, *kif4*, *hmgb1b*, *hmgb2a*, *hmgb3a*, *hspd1*, *hspa9*, *rplp0*, *hnrnpaba*, *rps2*, *rps12*, *rpl12*, *rps13*, *rps14*, *rps15a*, *rpl10*, *rps3a*, *rpl31*, *rpl37*, *rps6*, *rpl9*, *rpl11*, *rpl34*, *rpl13*, *rpl36a*, *rpl26*, *rps8a*, *rpl21*, *rps27.1*, *rpl27a*, *cirbpb*) was also excluded from downstream analyses as described (Wagner et al, 2018).

PCA, batch effect diminishment, UMAP analysis, and clustering were conducted using Scanpy. The clusters were manually annotated by marker genes, and the scripts are available at https://github.com/benjaminfang/rbm24-workflow.

## RNA-seq and alternative splicing analysis

Sample preparation, sequencing library construction, deep sequencing, and bioinformatic analysis were performed essentially as

described (Shao et al, 2020). For alternative splicing analysis, rMATS-Turbo (Wang et al, 2024) was employed to generate FDR values for alternative splicing events showing significant differences between siblings and M*rbm24a* at the 4-cell stage. Read coverage of germ plasm-related genes was also plotted to provide an intuitive representation of alternative splicing events.

## IP/MS

Approximately 2200 *rbm24a-GFP* KI or wild-type embryos at 4 hpf were lysed and incubated on ice for 10 minutes in 1 mL of cold lysis buffer containing 10 mM Tris-HCl, pH 7.5, 150 mM NaCl, 0.5 mM EDTA, 0.5% NP-40, and 1× protease inhibitor cocktail (Roche, 04693132001). The cell lysates were centrifuged at $12,000 \times g$ for 5 minutes at 4 °C, and the supernatants were precleared by treatment of protein G agarose beads (100 μl for each sample) at 4 °C for 1 h. The precleared lysates were incubated with 30 μL of GFP-Trap Agarose beads (ChromoTek, gta-20) under rotation at 4 °C overnight, and 10 μL supernatant was saved as the flow-through fraction. The beads were washed four times with wash buffer (10 mM Tris-HCl, pH 7.5, 150 mM NaCl, 0.05% NP-40, and 0.5 mM EDTA), and then mixed with loading buffer and boiled at 95 °C for 5 minutes. After controlling the IP quality by western blot, the remaining IP samples were subjected to SDS-PAGE, followed by silver staining and gel cutting for mass spectrometry (MS) analysis. Scatter plots were generated for enrichment index to relative intensity, where the enrichment index is calculated as log10(1/(relative intensity in WT/(relative intensity in KI + 0.01))), and relative intensity is the intensity of a specific protein divided by the total intensity of all detected proteins.

## RIP-seq and bioinformatics analysis

Three hundred embryos were lysed in 300 μL of lysis buffer (150 mM NaCl, 10 mM HEPES, pH 7.6, 2 mM EDTA, 0.5% NP-40, 0.5 mM DTT, 1% protease inhibitor cocktail, and 400 U/mL of RNase inhibitor). An aliquot (25 μL) of lysates was saved as input by mixing with 1 mL of TRIzol and 475 μL of lysates were precleared with 10 μL of protein G magnetic beads at 4 °C for 2 h. After removing the beads, the lysates were incubated with 15 μL of GFP-Trap Magnetic Agarose beads (ChromoTek, gtma) at 4 °C under rotation for 4 h. The beads were collected and washed four times with 1 mL of ice-cold NT2 buffer (200 mM NaCl, 50 mM HEPES, pH 7.6, 2 mM EDTA, 0.05% NP-40, 0.5 mM DTT, and 400 U/mL of RNase inhibitor), followed by incubation with proteinase K (5 mg/mL in 200 μL of 1× buffer) at 50 °C for 40 minutes. Precipitated RNAs were recovered by phenol-chloroform extraction. Libraries were constructed using Hieff NGS Ultima Dual-mode mRNA Library Prep Kit.

Adapter sequences were trimmed using the fastp (version 0.22.0). The remaining reads were aligned to the zebrafish genome (GRCz11) using STAR (version 2.7.11a) with the following parameters: --outFilterMismatchNoverReadLmax 0.02 --outFilterMultimapNmax 1 --alignEndsType Extend5pOfRead1 --outSJfilterReads Unique. To identify peaks in RIP-seq datasets, MACS2 software (v.2.2.9.1) was employed with the following parameters: macs2 callpeak -c control.bam -t treat.bam --nomodel -f BAMPE -q 0.05 -g 1368780147. This configuration was chosen to effectively identify RNA-binding protein interaction sites within the RIP-seq data. Reads were compared to their respective inputs

for the identification of significant peaks. Using this significant peak data, we calculated KI (IP vs input) and WT (IP vs input), respectively. The fold change was then determined as KI (IP vs input)/WT (IP vs input). Therefore, the control used to calculate fold changes in Fig. 6B is WT (IP vs input). Peaks were considered significant if they exhibited a fold change greater than 5 and a *P* value less than 1e-6. ClusterProfiler (version 4.8.3) was used for GO enrichment analysis (*P* value cutoff of 0.05 to determine significant terms).

For samples using IgG as a control, precleared lysates from KI embryos were incubated with IgG (Abcam, ab18413) and protein A/G magnetic beads (Invitrogen, 80104 G). Bioinformatic analysis was carried out similarly to the above.

## RIP-qPCR

The GFP-Trap Magnetic Agarose beads (ChromoTek, gtma) were used for IP. Three hundred 4-cell *rbm24a-GFP* KI or wild-type embryos were crosslinked in 0.9 ml 1% formaldehyde for 10 minutes on ice and performed following a previously established protocol (Shao et al, 2020).

## Isolation of the germ plasm particles

M*rbm24a* mutants and wild-type embryos at the 4-cell stage (13 embryos for each genotype) were lysed in 200 μL of P-body lysis solution (50 mM Tris-HCl, pH 7.4, 1 mM EDTA, 150 mM NaCl, and 0.2% Triton X-100) containing 40 U/mL of murine RNase Inhibitor (Vazyme, R301-03-AA) and 1× protease inhibitor cocktail (Hubstenberger et al, 2017). After 20 repeated aspirating/dispensing cycles using a 200 μL pipette, embryos were incubated on ice for 10 minutes, followed by centrifugation at $15,000 \times g$ at 4 °C for 20 minutes. The supernatant was carefully transferred to a new centrifuge tube for total RNA extraction using the standard TRIzol-phenol-chloroform method. Luciferase mRNA in proportion to the total RNA amount was added to each sample (1 μL for M*rbm24a* mutants and 0.9 μL for wild-type embryos) to serve as the loading control. Then, equal volumes of RNA samples were reverse transcribed and subjected to qPCR analysis to determine the expression levels of germ plasm mRNAs. Proteins extracted from the supernatant and the pellet were subjected to western blot analysis to examine the germ plasm protein Piwil1.

## Immunofluorescence

Embryos or cells were fixed in 4% PFA at 4 °C overnight. After three washes with PBSTr, they were blocked with 10% sheep serum at room temperature for 1 h, followed by incubation with primary antibodies against myc (1:200) or Piwil1 (1:200) at 4 °C overnight. Samples were washed four times with PBSTr and incubated with Alexa Fluor488 anti-mouse IgG (1:100; ZSGB-BIO) or Alexa Fluor 594 anti-rabbit IgG (1:100; ZSGB-BIO) at room temperature for 1 h. After washing three times with PBSTr, they were embedded in 1% low-melting agarose (Biotech, CA1351) for confocal imaging.

## Paraffin sectioning and H&E staining

Testes isolated from adult siblings and M*rbm24a* mutants were fixed in 4% PFA at 4 °C overnight. After washing with PBS,

they were dehydrated with graded ethanol and embedded in paraffin for sectioning at 5 μm thickness. After deparaffinization and rehydration, sections were subjected to standard H&E staining and examined using the Leica DM 2500 microscope.

## Expression and purification of recombinant Rbm24a protein

To express the protein in *Pichia* yeast, the pPIC9K plasmid (Invitrogen, V175-20) was linearized by NotI, and the *rbm24a-BFP-6×His* construct was subcloned and fused to the α-factor secretion signal via a modified Gibson Assembly (Xia et al, 2019). The resulting pPIC9K-*rbm24a-BFP-His* plasmid was then transformed into yeast cells (*Pichia Pastoris* GS115) through electroporation. Multicopy integration yeast colonies were selected following the instructions in the pPIC9K manual. Single clones were cultured in 5 ml of YPD medium for 18 h. The culture was then transferred into 100 ml of BMGY medium at a 1:100 dilution and incubated at 28 °C with shaking at 200 rpm in a baffled flask until the optical density (OD600) reached 2–6. The cells were harvested by centrifugation at $3000 \times g$ for 5 minutes at room temperature and resuspended in 500 ml of BMGY medium supplemented with 1% methanol to induce protein expression. Methanol (1% v/v) was added every 24 h, and the culture was incubated at 28 °C with shaking at 200 rpm for 6–8 days. The supernatant was collected by centrifugation at 8000 rpm for 15 minutes, cleared through a 0.22-μm filter, and incubated with nickel column resin (Bestchrom AA0051) at 4 °C for at least 4 h. After the supernatant passed through the column along with the resin, the protein was eluted using a gradient of imidazole concentrations (0, 20, 100, 500 mM). The eluted fractions were analyzed by SDS-PAGE and Coomassie blue staining. The fractions containing the target protein were concentrated using a 20 kDa ultrafiltration membrane. Finally, the purity and expression level of the target protein were assessed by western blot.

For the expression of recombinant protein in the Sf9 cell line, we utilized Gibson assembly to individually insert *rbm24a-BFP-6×His* and *buc-GFP-6×His* into the pFastBac plasmid. The resulting constructs were subsequently transformed into DH10Bac supercompetent cells. Recombinant Bacmid was extracted using the Bacmid Miniprep Kit (Beyotime, D0031), and the Bacmid was then transfected into Sf9 cells according to the Cellfectin™ II (Thermo Fisher Scientific, CN10362100) protocol. The cells were incubated in the dark at 27 °C and 130 rpm for 7 days. Following this, the supernatant, representing the P0 viral stock, was harvested by centrifugation at room temperature and 1500 rpm. 1 mL of the supernatant was then added to 100 mL of fresh Sf9 cells and cultured for an additional 5 days to obtain the P1 viral stock. The transfection procedure was repeated at a 1:100 dilution, followed by a further 3-day incubation. Finally, the cells were harvested by centrifugation and lysed by sonication. Following clearance, the supernatant was processed for protein purification using the standard method described above.

## Data availability

All data, constructs and reagents reported in this paper are available to the correspondence author Ming Shao, upon reasonable request. The

RNA-seq, scRNA-seq and RIP-seq data reported in this paper have been deposited in the National Center for Biotechnology Information Gene Expression Omnibus (GEO) database, https://www.ncbi.nlm.nih.gov/geo (accession no. GSE267406, GSE279756, GSE267086 and GSE288109). The image and numerical source data for this paper have been deposited in the Biostudies database under the accession number biostudies:S-BSST1826.

The source data of this paper are collected in the following database record: biostudies:S-SCDT-10_1038-S44318-025-00442-z.

## Peer review information

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

## Acknowledgements

We thank Jianlin Shen, Yiteng Xu, and Qingqing Wei from the Core Facility and Service Platform, School of Lifesciences, Haiyan Yu and Xiaomin Zhao from SKLMT, France Lam and Chloé Chaumeton from the IBPS imaging facility and professor Chengtian Zhao from the Ocean University of China for assistance with confocal and light sheet microscopic imaging. We are also grateful to Professors Jian Zhang, Guangshuo Ou, Li Yu, Feng Liu, and Anming Meng for sharing their ideas to improve this work, as well as to Professor Baocai Tan and his team for providing their *Pichia pastoris* protein expression system. This work was supported by the National Natural Science Foundation of China (grants 32170816, 32370860, 32450630 and 31871451 to Ming Shao; grant 32301237 to Ang Li), Program of Outstanding Middle-aged and Young Scholars of Shandong University (Ming Shao), the Taishan Scholars of Shandong Province (Ming Shao and Ang Li), Natural Science Foundation of Shandong Province (grant ZR2023QC174 to Ang Li) and Programs of Shandong University Qilu Young Scholars (Ang Li) and the French Muscular Dystrophy Association (AFM-Téléthon grant 23545 to De-Li Shi).

## Author contributions

**Yizhuang Zhang**: Data curation; Formal analysis; Validation; Investigation; Visualization; Methodology; Writing—review and editing. **Jiasheng Wang**: Investigation. **Hailing Fang**: Data curation; Investigation; Visualization. **Shuqi Hu**: Investigation. **Boya Yang**: Investigation. **Jiayi Zhou**: Investigation. **Raphaëlle Grifone**: Investigation. **Panfeng Li**: Data curation; Investigation; Visualization. **Tong Lu**: Investigation. **Zhengyang Wang**: Investigation. **Chong Zhang**: Investigation. **Yubin Huang**: Investigation; Methodology. **Dalei Wu**: Methodology. **Qianqian Gong**: Writing—review and editing. **De-Li Shi**: Formal analysis; Supervision; Funding acquisition; Investigation; Writing—review and editing. **Ang Li**: Data curation; Supervision; Investigation; Methodology; Writing —review and editing. **Ming Shao**: Conceptualization; Data curation; Formal analysis; Supervision; Funding acquisition; Investigation; Methodology; Writing —original draft; Project administration; Writing—review and editing.

Source data underlying figure panels in this paper may have individual authorship assigned. Where available, figure panel/source data authorship is listed in the following database record: biostudies:S-SCDT-10_1038-S44318-025-00442-z.

## Disclosure and competing interests statement

Claims protecting IP rights to the applications of strategies generating maternal mutants and targeting Rbm24a to control fish fertility are pending in China patent publications 2024103679709 and 2024103679728.

# Expanded View Figures

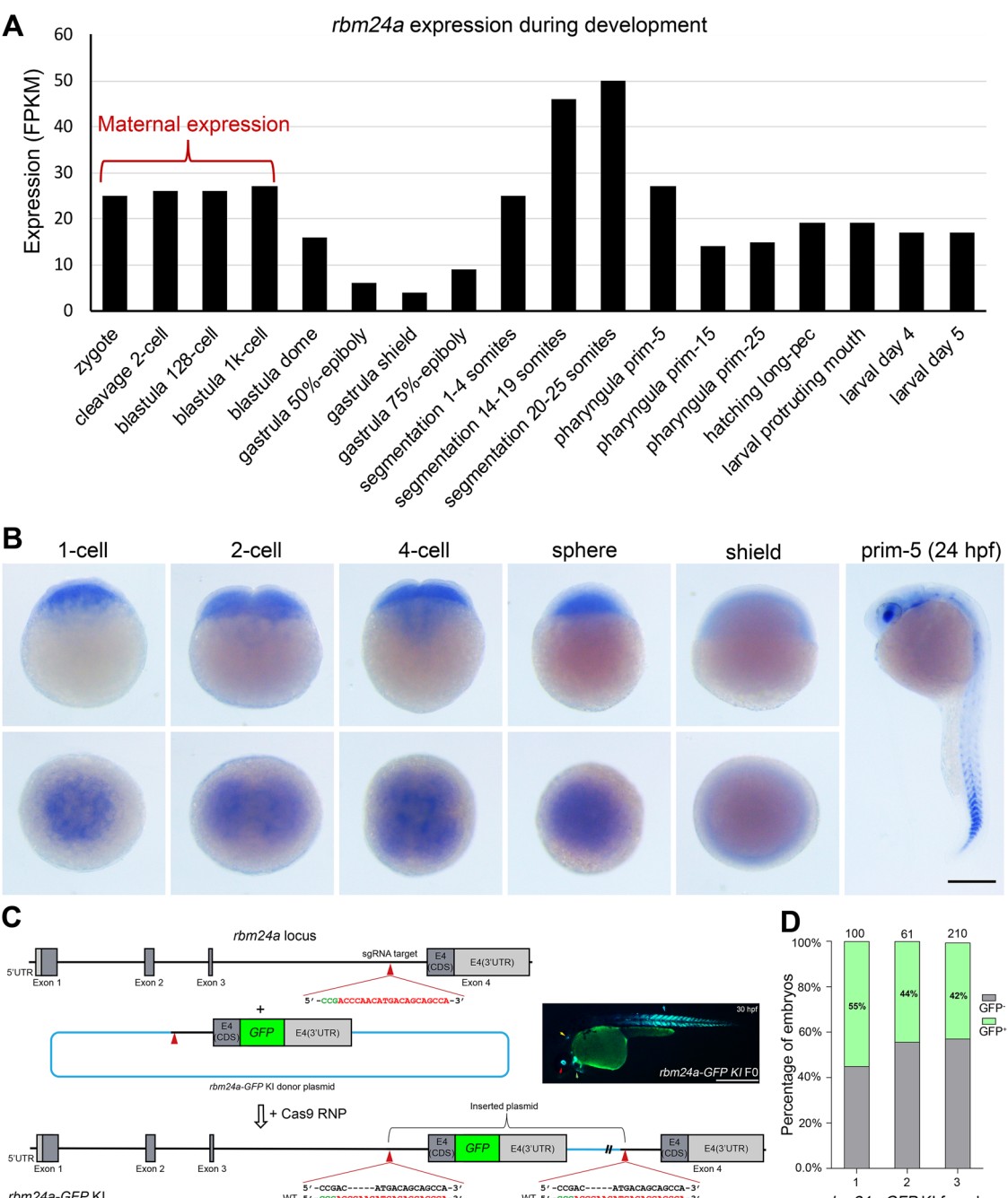

**Figure EV1. Expression of zebrafish *rbm24a* transcripts and construction of *rbm24a-GFP* KI line.**

(A) Expression levels of *rbm24a* transcripts at different developmental stages, replotted from a published resource (White et al, 2017). (B) Spatial expression of *rbm24a* transcripts. Scale bar, 250 µm. (C) Diagram illustrating the knock-in strategy to insert a GFP tag at the C-terminus of endogenous Rbm24a. The inset on the right shows an F0 embryo with early integration of the knock-in plasmid. Scale bar, 500 µm. (D) Germline transmission of three founders with early integration event. Numbers on top of each column represent embryos analyzed from each founder. Source data are available online for this figure.

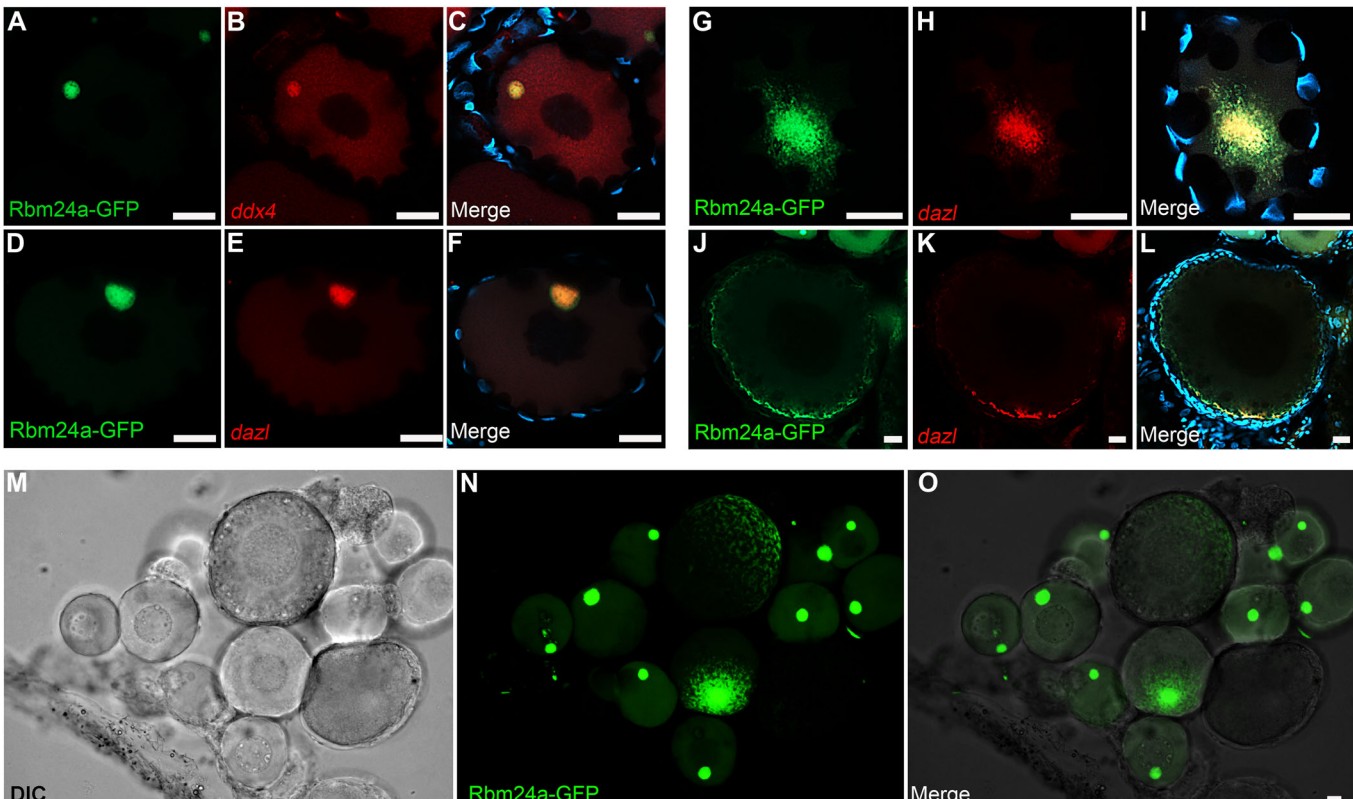

**Figure EV2.  Rbm24a localization during oogenesis.**

(A–F) Colocalization of Rbm24a-GFP with *ddx4* and *dazl* in the BB of stage Ia oocyte. Scale bars, 20 μm. (G–L) Rbm24a is released following BB breakdown in the late stage Ib and stage II oocytes. Scale bars, 20 μm. (M–O) Live *rbm24a-GFP* KI oocytes at different stages. Note the BB localization of Rbm24a in stage I oocytes and a spread of Rbm24a as patched forms in larger oocytes. Scale bar, 20 μm. Source data are available online for this figure.

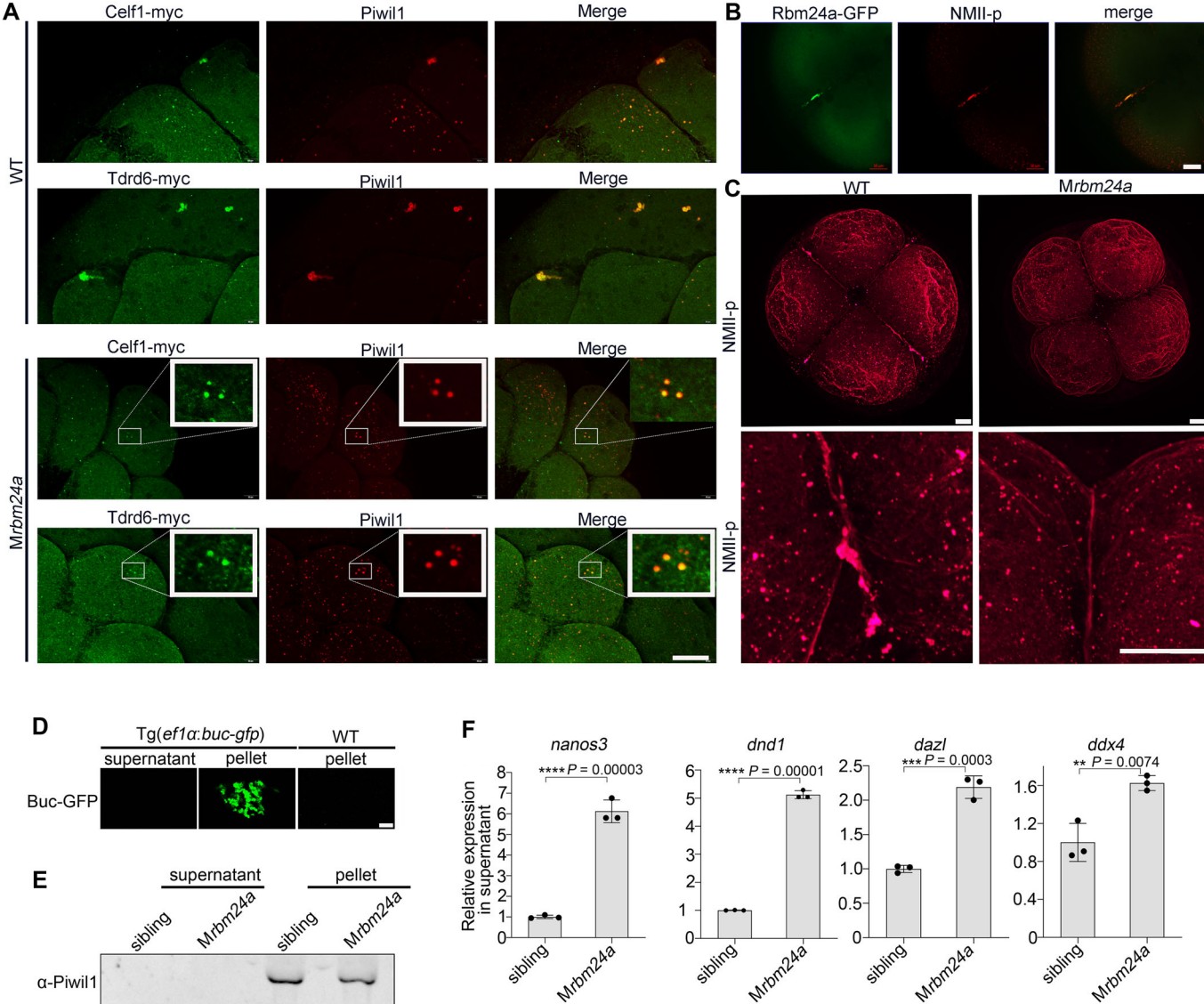

**Figure EV3. Rbm24a is required for the recruitment of germ plasm mRNAs but not proteins into germ granules.**

(A) Recruitment of Tdrd6-myc and Celf1-myc into germ plasm condensates in the presence or absence of maternal Rbm24a. Scale bar, 50 μm. (B) Colocalization of NMII-p with Rbm24a-GFP in the cleavage furrow. Scale bar, 50 μm. (C) NMII-p localizes in small germ granules in the absence of maternal Rbm24a. Scale bars, 50 μm. (D) Isolated germ plasm condensates examined by confocal microscopy. Scale bar, 10 μm. (E) Western blot analysis of Piwil1 in the supernatant and the pellet. (F) Expression levels of germ plasm mRNAs in the supernatants of siblings and M*rbm24a* embryos ($n = 3$ independent biological samples). Data are presented as mean ± SD. **$P < 0.01$, ***$P < 0.001$ and ****$P < 0.0001$, unpaired Student's *t* test. Source data are available online for this figure.

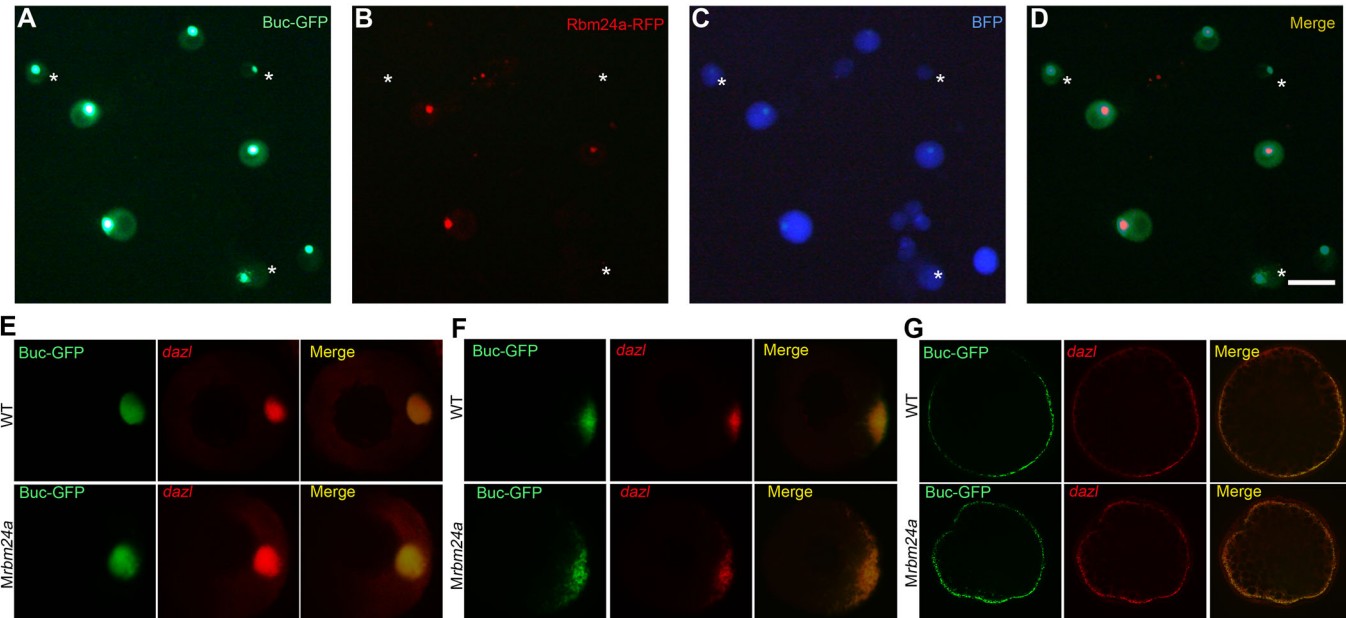

**Figure EV4. Loss of Rbm24a function does not affect the formation or dissociation of BB.**

(A–D) Identification of *rbm24a* mutant oocytes in a homozygous *rbm24a-RFP* KI^zpc:cas9^;Tg(*U6:4xsgRNA*^rbm24a^);Tg(*ef1a:buc-GFP*) background. Notably, Rbm24a-RFP signal is absent in mutant oocytes (asterisks). Scale bar, 500 μm. (E) BB formation is normal after the loss of M*rbm24a*, as revealed by the distribution of Buc-GFP and *dazl* mRNA in stage Ia oocytes. Scale bar, 20 μm. (F, G) BB dissociation is not affected by the absence of maternal Rbm24a. Oocytes in (F) are at late stage Ib, while oocytes in (G) are at stage III. Scale bars, 20 μm. Source data are available online for this figure.

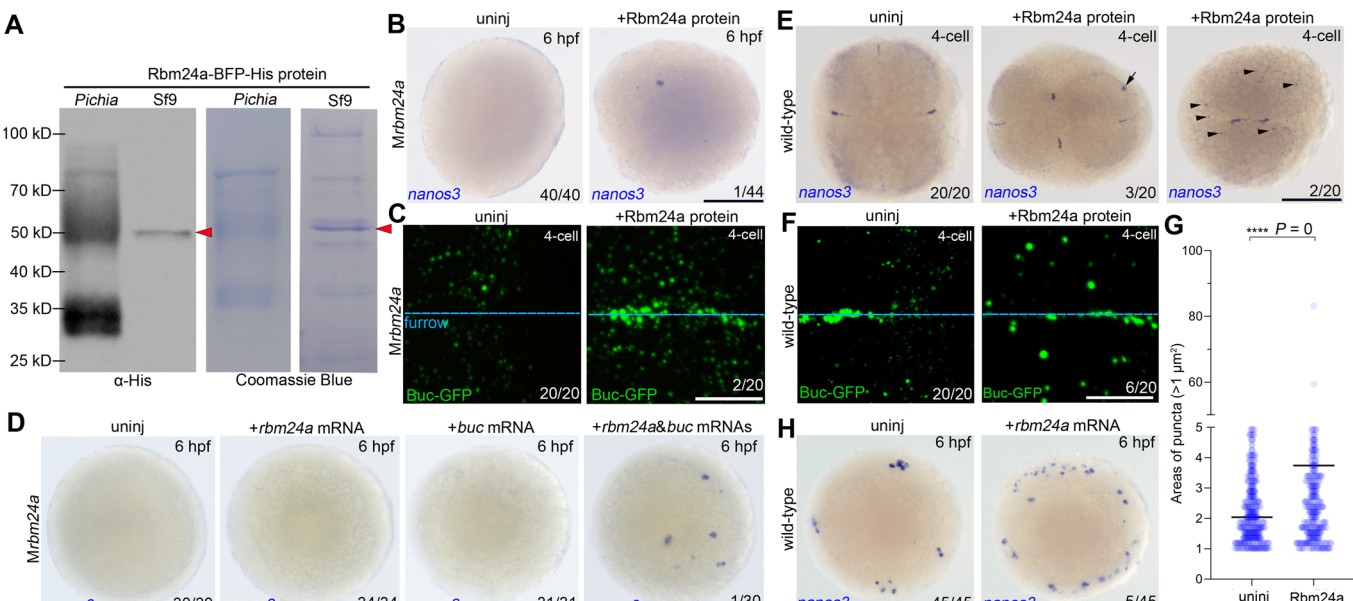

**Figure EV5. Overexpression of Rbm24a induces germ granule aggregation and PGC formation with low efficiency.**

(A) Purified recombinant Rbm24a-BFP-His protein from Pichia yeast and Sf9 cell lines. (B) Injection of recombinant Rbm24a protein rescues PGC formation in M*rbm24a* with low efficiency. Scale bar, 200 μm. (C) Supplementation of recombinant Rbm24a protein induces increased germ granule aggregation along the cleavage furrow. Scale bar, 50 μm. (D) Co-injection of *rbm24a* and *buc* mRNAs into M*rbm24a* embryos rescues PGC formation defect with low efficiency, while separate injection has no effect. Scale bar, 200 μm. (E) Injection of recombinant Rbm24a protein into wild-type embryos results in ectopic and large germ plasm aggregates. Scale bar, 200 μm. (F) Rbm24a promotes large germ granule formation outside the cleavage furrow, as monitored by transgenic Buc-GFP marker. Scale bar, 50 μm. (G) Scatter plot shows the size of germ granules outside the cleavage furrow (data are from three independent embryos for each group). Only granules with size >1 μm² were counted. The arithmetic mean was denoted by black horizontal lines. For uninjected group $n = 498$, for Rbm24a protein-injected group $n = 640$. ****$P < 0.0001$, unpaired Student's *t* test. (H) Increased PGCs in embryos injected with *rbm24a* mRNA at 6 hpf. Scale bar, 200 μm. Source data are available online for this figure.

