## [Peer Review File · The EMBO Journal]

Rbm24a dictates mRNA recruitment for germ granule assembly in zebrafish

Yizhuang Zhang, Jiasheng Wang, Hailing Fang, Shuqi Hu, Boya Yang, Jiayi Zhou, Raphaëlle Grifone, Panfeng Li, Tong Lu, Zhengyang Wang, Chong Zhang, Yubin Huang, Dalei Wu, Qianqian Gong, De-Li Shi, Ang Li, and Ming Shao

Corresponding author(s): Ming Shao (shaoming@sdu.edu.cn) , De-Li Shi (de-li.shi@upmc.fr), Ang Li (angli41@sdu.edu.cn)

Review Timeline:

Submission Date:	2nd Jul 24
Editorial Decision:	4th Sep 24
Revision Received:	1st Feb 25
Editorial Decision:	14th Mar 25
Revision Received:	15th Mar 25
Accepted:	21st Mar 25

Editor: *Cornelius Schneider*

Transaction Report:

Dr. Shao,

Thank you for submitting your manuscript for consideration by the EMBO Journal and for sharing a preliminary revision plan with me.

Based on your willingness to engage in a major revision as indicated during the pre-decision consultation, I would like to invite you to submit a revised version of the manuscript, addressing the comments of all three reviewers. I should add that it is EMBO Journal policy to allow only a single round of revision, and acceptance of your manuscript will therefore depend on the completeness of your responses in this revised version. If you have any additional questions or want to discuss the revisions further, I am happy to do so by email or video conferencing.

We generally allow three months as standard revision time, which can be extended to 6 months in case of major revisions, such as the experiments required here. As a matter of policy, competing manuscripts published during this period will not negatively impact on our assessment of the conceptual advance presented by your study. However, we request that you contact the editor as soon as possible upon publication of any related work, to discuss how to proceed. Should you foresee a problem in meeting the deadline, please let us know in advance and we may be able to grant an extension.

Thank you for the opportunity to consider your work for publication. I look forward to your revision.

Yours sincerely,

Cornelius

Cornelius Schneider, PhD
Editor
The EMBO Journal
c.schneider@embojournal.org

We realize that it is difficult to revise to a specific deadline. In the interest of protecting the conceptual advance provided by the work, we recommend a revision within 3 months (3rd Dec 2024). Please discuss the revision progress ahead of this time with the editor if you require more time to complete the revisions. Use the link below to submit your revision:

Referee #1:

In many species, including some vertebrates the germline is specified by inheritance of maternal factors, termed germ plasm. The germ plasm is produced in oocytes and accumulates in a small number of cells that will become the primordial germ cells (PGCs) of the embryo. Although several factors involved in PGC development have been identified, the factors that constitute the germ plasm and the mechanisms and regulators of its assembly remain poorly understood. In zebrafish prior work demonstrated the involvement of intrinsically disordered proteins, RNA binding proteins and kinesin motors in regulating recruitment of the germ plasm to the cleavage furrows of the cells that will become PGCs. In this work the authors investigated the role of the RNA binding protein, Rbm24 in germ plasm recruitment and PGC formation. Prior work identified zygotic requirements for Rbm24a during development that precluded studies of its later functions. In this work the authors generated a knock in line to tag endogenous Rbm24. This transgenic reporter line revealed localization of Rbm24a to the germ plasm in oocytes (within a conserved structure known as the Balbiani body and later with the germ plasm, at the cleavage furrows and in the PGCs). They conducted detailed analysis and colocalization studies with known germ plasm components to verify that Rbm24a aggregates were indeed germ plasm associated. Further, they performed timelapse analysis to visualize Rbm24a aggregation (phase separation/condensation) in the embryo. To determine if maternal Rbm24a was essential for PGC development, they developed a novel maternal out strategy to bypass the zygotic phenotypes. Specifically, they generated a tagged knock in line with Cas9 driven by an oocyte promoter downstream of the tag.

They provide multiple lines of evidence that maternal Rbm24a is required for phase-separation and germ plasm recruitment to the furrows, accumulation and PGC specification, and fertility. Single cell RNAseq data are provided to demonstrate loss of the PGCs, and no effects on somatic populations in the absence of maternal Rbm24a. Using their transgenic and mutant lines and the zGrad system, they demonstrate that maternal Rbm24a is required in the embryo - zGrad degrades the maternal GFP-tagged-Rbm24a. Using the tagged-Rbm24a transgenic line, they performed IP studies to identify interacting partners of Rbm24a. They provide evidence that Rbm24a binds to germ plasm RNAs (among many other RNAs) and also binds to the known germ plasm factor, Bucky ball (Buc). They suggest that Rbm24a and Buc form a complex that is required for recruitment and protection/stabilization of the other germ plasm factors. Using antibodies to deplete Kinesin 1, which was previously shown to be required for recruitment of germ plasm to the cleavage furrows, they show that Rbm24a recruitment is also Kinesin-1 dependent. Overall, the work is exciting, rigorous, comprehensive, well-presented, clearly written, and uses cutting edge strategies to further our knowledge of the mechanisms regulating formation of the vertebrate germline. Enthusiasm for the study is high and the paper is well done overall; however, there are a few missing details and minor points that would benefit from clarification.

Major:

1) Although data are provided regarding Page 5, Line 30. "The high genome editing efficiency of *rbm24a*-RFP-KI-Cas9 remained stable over 3 consecutive founder generations" - This point needs clarification. Although data are provided in supplemental figure 3, those data show injection of guides against *bmp* into *rbm24a*-RFP-KI-Cas9 transgenic eggs. This experiment shows that cas9 is stably expressed over multiple generations, but it does not show the efficiency or frequency of editing in the oocyte or double transgenic experiment using *rbm24a*-RFP-KI-Cas9 and Rbm24a guides expressed from a transgene. A sample clutch is shown with RFP+ and RFP- embryos, but no transmission frequencies are reported. Given the novelty of the approach, it would be beneficial to the field to include 1) the knock in frequency and number of independent lines generated and 2) the transmission frequency based on fluorescence and 3) the editing efficiency in the double transgenic context, which can only be determined by sequencing the genetic lesions. The authors likely already have this data and/or it should be readily attainable give the apparent efficiency and that the maternal mutants are viable.

Minor:

- 2) Fig. 4. Title: "Germ plasm granules fail to undergo kinesin-dependent transport in the absence of Rbm24a". Consider revising the title as the data show two things 1) Buc localization to the furrow requires Rbm24a and Rbm24a transport requires Kinesin 1. A title that more accurately reflects the data shown might be "Germ plasm transport to the furrows requires Rbm24a and is Kinesin 1 dependent".
- 3) Fig.6 panel e. The HEK293 data don't really add much to the story since somatic cells don't express germ plasm factors.
- 4) Fig. 6. A description of panel F is missing from the legend.
- 5) Fig 6 G: Is this IP from HEK293 or zebrafish embryos?
- 6) Page 3, Lines 30-33 indicate that Dnd1 and Nanos3 are required for PGC determination, but prior published work from the Moens, Draper and Raz labs (many of which are cited) show that they are required for PGC migration and maintenance of PGC/germline identity rather than PGC specification. A more recent paper Westerich et al Dev cell 2023 should be cited and discussed. Also, Dnd appears twice in this section with two conflicting statements about its functions.
- 7) Page 3, Line 36-37. Discusses why germ plasm RNAs are degraded in somatic cells but does not discuss the previously published work indicating involvement of miRNAs and Dazl-mediated repression or blocking or the miRNAs in PGCs but not somatic cells. This work should be cited.
- 8) Page 4, Line 28. "directed migration". Consider instead directed transport as migration is usually used to describe movement of cells rather than subcellular entities.
- 9) Page 10, Line 29. "This is the first evidence demonstrating that a protein component controls the localization of germ plasm mRNAs in phase separated structures". This sentence is a bit misleading as written and should be revised as several proteins have been demonstrated to be involved in localizing germ plasm mRNAs in various species. It is more accurate and fairer to others in the field to state that "Rbm24 is a novel factor that controls" or something similar.
- 10) Page 11, Line 1: "The RNA independent phenomenon of phase separation". The data don't really show that localization and aggregation of the proteins is RNA independent, rather it is independent of Rbm24a and the RNAs that it recruits.
- 11) Page 11, Line 6 "previously believed" should be "Previously shown" since there are published data cited.
- 12) Page 11, Lines 10-15 discuss the mechanism involving Buc:Rbm24a complexes and Kinesin 1. Prior work, Campbell et. al. showed that Buc binds to Kinesin 1. Several possibilities are discussed, including Rbm24a potentially binding Kinesin 1 or having allosteric effects on Buc. It would be helpful to state this and incorporate the prior work into the models.
- 13) Page 11, Line 25: "four most giant aggregates". Consider instead "four largest aggregates"
- 14) Page 11, Page 11, Line 30: Not sure what is meant by "applications in controlling reproductive traits". Do the authors mean use as a contraceptive? Or to increase fertility?

Referee #2:

In the submitted manuscript, Zhang et al. utilize a zebrafish model to investigate the role of an RNA-binding protein, Rbm24a, in regulating germ plasm assembly. They identify Rbm24a as a key factor in the specific sorting of germ plasm mRNAs during PGC formation, primarily through its interaction with Buc, which facilitates the capture of germ plasm mRNAs in phase-separated condensates. This study offers significant insights into the function of Rbm24a as an organizer of germ plasm, enriching our understanding of phase-separated protein scaffolds. While the manuscript provides valuable contributions to the field of germ cell biology, several major concerns need to be addressed before further consideration.

Major Points

1. Validation of Mrbm24a Mutants: The study extensively utilizes Mrbm24a mutant embryos. It is essential to verify that the BFP-positive embryos are indeed maternally knocked out by sequencing.
2. Rescue Experiments: Since the manuscript emphasizes the role of maternal Rbm24a in germ plasm formation, the authors should demonstrate that the injection of in vitro generated Rbm24a protein (as opposed to rbm24a mRNA) can rescue the Mrbm24a phenotype. Additionally, the effect of rbm24a MO injection should be explored.
3. Phase Separation Assay: One of the key conclusions is that Rbm24a and Buc bind to nanos3 mRNA, forming aggregates that stabilize the mRNA. A phase separation assay using purified proteins should be performed to support this claim (see <https://www.ncbi.nlm.nih.gov/pmc/articles/PMC6215329/>).
4. Further Validation of Germ Plasm Recruitment: Additional experiments are needed to substantiate the conclusion that "Maternal Rbm24a is required for the recruitment of germ plasm mRNAs." For instance, the injection of rbm24a mRNA (alone or in combination with buc mRNA) into a single blastoderm cell could be used to examine whether dispersed germ plasm in Mrbm24a mutants can be reorganized, or whether ectopic germ plasm aggregation centers can form in wild-type embryos (see <http://www.ncbi.nlm.nih.gov/pubmed/19249209>).
5. Oocyte-Specific Mutation Analysis: Given the pronounced defects in Mrbm24a mutants as early as the 1-cell stage, a more detailed analysis of germ plasm distribution in oocytes is required. What are the effects of an oocyte-specific mutation of rbm24a on oocyte development and Balbiani body formation?
6. In Fig. 2p, Mrbm24a mutants appear to lack a specific cell cluster, distinct from PGCs. What cell type is this? Its absence seems closely related to the loss of Mrbm24a.
7. In Figure 4a-d, Buc-GFP-labeled germ plasm granules in wild-type embryos rapidly move toward the cleavage furrows. In contrast, in Mrbm24a mutants, germ plasm granules still appear, which contradicts the absence of granules reported in Figure 3b. This discrepancy requires further clarification.

8. The rationale for using wild-type controls in the RIP-seq experiment should be clarified, why was an IgG control group not used. Additionally, more details are needed regarding the experimental procedures. What control was used to calculate fold changes in Figure 6b?
9. The authors suggest that maternal Rbm24a recruits germ plasm mRNAs to condensates. Given that injecting zGrad might prevent germ plasm accumulation, it would be useful to assess germ plasm distribution at earlier time points.

Minor Points

1. P5L24: There is an extra dot ".".
2. P6L2-4: In situ hybridization or immunostaining should be performed to confirm the presence of gonadal somatic cells and germ cells, verifying the germ cell-less phenotype.
3. Fig. 2o: Statistical analysis is required.
4. P6L30: The scRNAseq results should be uploaded to a public database.
5. P8L1: For the buc-GFP transgenic fish, which promoter and UTR were used?
6. P8L15: The effectiveness and specificity of the kinesin-1 antibody should be validated in this experimental context.
7. P8L40: Why was rbm24a KI not used for the co-IP experiment?
8. P9L6: It would be useful to show the reverse experiment-what happens to Buc granules in Mrbm24a mutants?
9. P9L27-28, Fig. 6e, f: The conclusion that nanos3 mRNA stability is enhanced after transfection with Buc or Rbm24a is unsupported. In situ or qRT-PCR only reflects mRNA levels, not stability.
10. The description of procedures, such as single-cell transcriptome sampling, needs more detail. Were Mrbm24a and sibling single cells mixed for transcriptome sequencing?
11. P13L6: The source of the Cas9 protein should be specified.
12. P13L16-17: The Mrbm24a mutation must be verified by sequencing.
13. P13L22: Which promoter and 3' UTR were used for the buc-GFP transgenic plasmid?
14. P15: References for the scRNAseq and bioinformatic analysis should be provided.
15. Fig. 6e: The stability of nanos3 mRNA should be assessed with different mutations in both Rbm24a and Buc.
16. Fig. 6f: Figure legends are missing.

Referee #3:

In their manuscripts entitled "Rbm24 dictates mRNA recruitment for germ plasm assembly", Zhang et al. explore the role of the Rbp24 protein in the formation of germ granules in zebrafish, particularly its involvement in mRNA recruitment to these granules. They demonstrate that the absence of Rbp24 prevents the coalescence of germ granules into larger aggregates, a phenomenon similar to the effects of disrupted kinesin-dependent motor movement. Additionally, the authors show that without Rbp24, germ granule mRNAs rapidly degrade, ultimately preventing the formation of primordial germ cells (PGCs).

The authors provide several novel insights into the mechanisms of germ granule formation in zebrafish. Notably, they identify a critical role for Rbp24 in both mRNA recruitment to germ granules and the aggregation of smaller granules into larger clusters. The complete loss of PGCs observed upon Rbp24 depletion underscores the central role of this protein in zebrafish germline development. The supplementary videos, especially Video #5, are visually striking and enhance the presentation.

However, several claims made by the authors are not fully substantiated by the experimental data provided in the manuscript. My primary and major concern is the assertion that Rbm24a loss leads to the failure of mRNA recruitment to germ granules, ultimately preventing PGC formation. While the authors suggest that this effect is specific to mRNA, as protein components of germ granules still condense, these data are not presented. Therefore, I recommend the following steps to address this gap:

- a) Examine the distribution of Buc, a protein known to nucleate germ granules in zebrafish. If WT amount of germ granules still form at the furrow, as indicated by the presence of Buc granules, the loss of Rbm24a likely impacts downstream steps in germ granule formation. Based on the images presented, it appears that Rbm24a is crucial for the recruitment of both mRNAs (e.g., nanos3, tdrd7a, ddx4) and proteins (e.g., Celf1, Piwil1, Tdrd6) to germ granules, as none of these components accumulate at the furrow in stage 4 embryos. Given that Rbm24a is a splicing factor, it is possible that it regulates the splicing of buc. In the absence of Rbm24a, improper splicing of buc could occur, preventing the formation of germ granules and subsequently hindering the recruitment of mRNAs to these structures.
- b) Show distribution of Buc in 1-cell - 4-cell embryos in WT and Rbm24a-mutant embryos, show its expression levels using qRT-PCR and western blot analysis. The authors should also verify that the splicing of Buc in Rbm24a mutant fish is similar to the one in WT fish.
- c) Since Rbm24a regulates splicing, it may regulate splicing of germ granule mRNAs, including nanos3, tdrd7a, ddx4,...Inappropriately spliced mRNAs might decay faster or lose germ granule zipcodes which would ensure their enrichment in germ granules. The authors should show expression levels of these mRNAs in WT and Rbm24a-mutant embryos to verify that lack of enrichment in germ granules might not be due to lower mRNA expression.
- d) Same goes for the splicing of Kif5ba. Lack of movement of germ granules in video 5 might be due to missplicing of this kinesin motor rather than direct effect of Rbm24a on the movement on germ granules. The authors should look at mRNA levels of Kif5ba and splice variants of this mRNA to verify that its gene expression levels and patterns are normal and like those observed in WT fish.

Additional comments: (and not in the order of importance):

- 1) Title (and abstract): The authors refer to Rbp24. But, as the author indicate, there are two isoforms in zebrafish, a and b and in this manuscript, isoform a has been studied. Please correct.

- 2) Title (and in text - introduction, line 21): in this manuscript, the authors are investigating germ granules not germ plasm. Germ plasm is the cytoplasm that contains germ granules. The two terms are not interchangeable and the authors should make sure to correct this throughout the text.
- 3) Abstract (and through the text): the authors indicate that Rbp24a is the nucleating organizer of germ granules in zebrafish where in fact Buckyball (Buc) has been shown to play this essential role
- 4) Introduction, line 25-26: "In particular, it is still debated whether proteins or RNAs coordinate the aggregation of RNP components^{18,19}." This might be the case for stress granules and P bodies but in germ granules in *Drosophila*, this role has been well characterized (please cite: <https://doi.org/10.1016/j.celrep.2023.112723>).
- 5) Introduction, line 26-28: "In *Drosophila*, it is well established that several RNA-binding proteins (RBPs), such as Oskar, Vasa, Nanos, Tudor and Piwi, are critically involved in the biogenesis of the pole plasm by regulating mRNA localization and translation." Of these, only Osk, vasa and Tudor affect germ granule assembly (note the term germ granule assembly, not germ plasm assembly). Nanos protein does not and Piwi does not either. Also, Tudor is not an RBP. Please correct these errors.
- 6) Introduction, line 30-34: The authors appear to refer to *ddx4* and *Ddx4* as two distinct regulators. Please correct.
- 7) Extended Figure 1a: please specifically state in the figure legend that the data has been re-plotted and provide specific citation.
- 8) Text, page 4, Line 24: Fig. 1c). In the F0 embryos, zygotic Rbm24a-GFP protein was expressed explicitly in the lens, heart, skeletal muscle, and inner ear hair cells, highly matching the *rbm24a* mRNA expression pattern in different tissues: Please state where specifically are these data shown or provide these data.
- 9) Page 5, line 2: The authors write that the homozygous *rbm24a*-GFP KI line is viable and fertile, yet no data is shown. Provide the survivability/hatching data of zebrafish with the *rbm24a*-GFP. Is the viability of these fish the same as the viability of the WT fish? What about its fecundity? This is important to establish that the GFP tagging and the CRISPR KI did not affect the normal function of this gene.
- 10) Figure 1c,d: label on the figure whether you are staining for mRNA or protein (for example: *ddx4* mRNA and *Ddx4* protein).
- 11) Page 5, line 3: "Moreover, the distribution pattern of Rbm24a-GFP unequivocally recapitulates the endogenous status of Rbm24a localization." This data is not shown. Also, the authors write that the GFP KI was generated because no antibody against Rbm24a exist. So, the authors likely refer to ISH of *rbm24a*. But these two stainings are not comparable as the protein expression does not necessarily mirror mRNA expression. Please, correct this statement. Also, provide ISH staining of *rbm24a*-GFP mRNA and compare it to *rbm24a* to show that the two display the same tissue distribution.
- 12) Page 5: "time until the sphere stage. At the 32-cell stage, the minimum size to successfully pass through the massive degradation window was 43.3 μm^2 (projected area)." What do authors refer to as degradation? Protein turnover? Degradation is assumed as no protein turnover has been measured. Also, the granules might have dissolved without degrading the protein. Please correct and clarify this statement.
- 13) Page 5, line 26: Like the *rbm24a*-GFPKI line, the homozygous *rbm24a*-RFPKI^{zpc:cas9} line was healthy and fertile, and Rbm24a. These data is not shown. Please provide these data (see comment # 9 above).
- 14) Page 5, line 35 - 45: Please provide the frequency (number of fish) the authors used in their analysis. Also, please provide data showing that *rbm24a* has been knocked down: provide qRT-PCR analysis and WT analysis (they have Rbm24a-RFP so Wb using anti RFP antibody is possible).
- 15) Page 7: Is Rbm24a essential for the formation of phase-separated germ plasm protein scaffold? To address this question, we checked the status of germ plasm granules in *Mrbm24a* embryos at the 1-cell stage by examining Piwil1 protein and *nanos3* mRNA. Unexpectedly, germ plasm granules." Buc is the scaffold of germ granules in zebrafish, which was shown previously - please see comments under "major comments" above.
- 16) Figure 5a and text on page 8: Another 158 proteins showed at least a 3-fold enrichment in *rbm24a*-GFPKI embryos compared to wildtype controls (Fig 5a). What is the WT control?
- 17) Page 9, Line 27: However, in the presence of Buc or Rbm24a, the stability of *nanos3* mRNA was significantly enhanced (Fig. 6e and f). Gene expression levels appear different from images but data has not been quantified. Please provide qRT-PCR analysis. Also, no data showing that the stability of *nanos3* mRNA has been affected is shown. A time course is required to determine changes in mRNA stability. Please correct these issues.
- 18) Page 9, Line 30: Closeup of Buc, Rbm24a and *nanos3* in Figure 6e would be useful.
- 19) Discussion, line 15 and 43: please cite: <https://doi.org/10.1016/j.celrep.2023.112723> as this paper characterizes the contribution of RBPs and mRNA to the formation of germ granules in *Drosophila*.
- 20) Discussion: "This is the first evidence demonstrating that a protein component controls the specific localization of germ plasm mRNAs in phase-separated structures." The role of Oskar, Rump and Aub protein in their ability to recruit mRNAs to germ granules in *Drosophila* has been demonstrated (look at Ephrussi and Gavis labs) - please include these works in your manuscript.
- 21) Discussion: Unexpectedly, the formation of phase-separated protein scaffold is independent of Rbm24a. These data are not shown. Please see my major comments.

Referee #1 (Report for Author)

In many species, including some vertebrates the germline is specified by inheritance of maternal factors, termed germ plasm. The germ plasm is produced in oocytes and accumulates in a small number of cells that will become the primordial germ cells (PGCs) of the embryo. Although several factors involved in PGC development have been identified, the factors that constitute the germ plasm and the mechanisms and regulators of its assembly remain poorly understood. In zebrafish prior work demonstrated the involvement of intrinsically disordered proteins, RNA binding proteins and kinesin motors in regulating recruitment of the germ plasm to the cleavage furrows of the cells that will become PGCs. In this work the authors investigated the role of the RNA binding protein, Rbm24 in germ plasm recruitment and PGC formation. Prior work identified zygotic requirements for Rbm24a during development that precluded studies of its later functions. In this work the authors generated a knock in line to tag endogenous Rbm24. This transgenic reporter line revealed localization of Rbm24a to the germ plasm in oocytes (within a conserved structure known as the Balbiani body and later with the germ plasm, at the cleavage furrows and in the PGCs. They conducted detailed analysis and colocalization studies with known germ plasm components to verify that Rbm24a aggregates were indeed germ plasm associated. Further, they performed timelapse analysis to visualize Rbm24a aggregation (phase separation/condensation) in the embryo. To determine if maternal Rbm24a was essential for PGC development, they developed a novel maternal out strategy to bypass the zygotic phenotypes. Specifically, they generated a tagged knock in line with Cas9 driven by an oocyte promoter downstream of the tag.

They provide multiple lines of evidence that maternal Rbm24a is required for phase-separation and germ plasm recruitment to the furrows, accumulation and PGC specification, and fertility. Single cell RNAseq data are provided to demonstrate loss of the PGCs, and no effects on somatic populations in the absence of maternal Rbm24a. Using their transgenic and mutant lines and the zGrad system, they demonstrate that maternal Rbm24a is required in the embryo - zGrad degrades the maternal GFP-tagged-Rbm24a. Using the tagged-Rbm24a transgenic line, they performed IP studies to identify interacting partners of Rbm24a. They provide evidence that Rbm24a binds to germ plasm RNAs (among many other RNAs) and also binds to the known germ plasm factor, Bucky ball (Buc). They suggest that Rbm24a and Buc form a complex that is required for recruitment and protection/stabilization of the other germ plasm factors. Using antibodies to deplete Kinesin 1, which was previously shown to be required for recruitment of germ plasm to the cleavage furrows, they show that Rbm24a recruitment is also Kinesin-1 dependent. Overall, the work is exciting, rigorous, comprehensive, well-presented, clearly written, and uses cutting edge strategies to further our knowledge of the mechanisms regulating formation of the vertebrate germline. Enthusiasm for the study is high and the paper is well done overall; however, there are a few missing details and minor points that would benefit from clarification.

We are delighted by the enthusiastic and positive assessment from this reviewer, who feels that this work is exciting, rigorous, comprehensive, well-presented and clearly written. This reviewer also appreciates the use of cutting-edge strategies to investigate the mechanisms regulating vertebrate germ cell differentiation. Indeed, these strategies allow protein tagging to

follow the dynamic distribution of germ plasm components, oocyte-specific knockout to circumvent the lethal phenotype of zygotic mutations, and zGrad-mediated degradation of GFP-tagged maternal proteins to monitor their temporal function.

Major:

1) Although data are provided regarding Page 5, Line 30. "The high genome editing efficiency of *rbm24a*-RFP-KI-Cas9 remained stable over 3 consecutive founder generations" - This point needs clarification. Although data are provided in supplemental figure 3, those data show injection of guides against *bmp* into *rbm24a*-RFP-KI-Cas9 transgenic eggs. This experiment shows that *cas9* is stably expressed over multiple generations, but it does not show the efficiency or frequency of editing in the oocyte or double transgenic experiment using *rbm24a*-RFP-KI-Cas9 and *Rbm24a* guides expressed from a transgene. A sample clutch is shown with RFP+ and RFP- embryos, but no transmission frequencies are reported. Given the novelty of the approach, it would be beneficial to the field to include 1) the knock in frequency and number of independent lines generated and 2) the transmission frequency based on fluorescence and 3) the editing efficiency in the double transgenic context, which can only be determined by sequencing the genetic lesions. The authors likely already have this data and/or it should be readily attainable given the apparent efficiency and that the maternal mutants are viable.

This reviewer thinks that the protein tagging and oocyte-specific gene knockout strategies are novel and would be beneficial to researchers studying early development. As suggested by the reviewer, we included 1) the knock-in frequency and the number of independent lines generated, 2) the transmission frequency based on fluorescence labelling, and 3) the editing efficiency in the double transgenic context. These data are incorporated in Fig. EV1, Appendix Fig. S2 and 3, and the main text. They are also collectively presented in the Fig. R1 for the ease of review.

For verification of knock-in efficiency, we selected embryos with wide-spread GFP/RFP-tagged *Rbm24a* expression as valid knock-in founders because the insertion should occur during cleavage stages and high germline transmission rate is expected. We thus term these events as "early integration" and a 0.45%~0.56% rate was observed (Fig. R1a). Three independent F0 founders with early integration events were generated for *rbm24a*-RFP^{zpc:cas9} and *rbm24a*-GFP knock-in lines, respectively. As expected, all F0 founders exhibited germline transmission, and each had 41% to 55% offspring showing fluorescent protein expression (Fig. R1b-c). Thus, the selection of founders with early integration allows raising even one F0 to obtain knock-in offspring.

Instead of analyzing mutation rate, we estimated the remaining wild-type mRNAs to accurately assess the editing efficiency. It would be difficult to quantify the ratio of mutant mRNAs or genomic lesions, because oocyte-specific genome editing may generate mutant mRNAs undergoing non-sense mediated decay or unintended large genomic deletions without primer binding sites (Zhang *et al*, 2021). In contrast, measuring the intact wild-type mRNAs by qRT-PCR is much more feasible. As such, a two-step protocol was designed. The first step was to perform a qRT-PCR analysis employing primers flanking the four sgRNA targeting sites (Fig. R1d). The amplification products should represent both wild-type and mutant mRNAs that can be amplified during the RT-PCR reaction, and their levels relative to knock-in siblings were designated as

ratio 1. The next step was to analyze PCR products by high resolution capillary electrophoresis. The ratio of PCR products representing wild-type mRNAs (ratio 2) can be obtained based on the electropherogram. Hence, the overall editing efficiency can be calculated as $1 - \text{ratio 1} \times \text{ratio 2}$ (Fig. R1e,f).

By following this strategy, we estimated the genome editing efficiencies of double transgenic embryos ($n > 20$ for each test) from three different founders (Fig. R1g). All these embryos showed high editing efficiency ranging from 71% to 81% (Fig. R1h).

Fig. R1

Minor:

2) Fig. 4. Title: "Germ plasm granules fail to undergo kinesin-dependent transport in the absence

of Rbm24a". Consider revising the title as the data show two things 1) Buc localization to the furrow requires Rbm24a and Rbm24a transport requires Kinesin 1. A title that more accurately reflects the data shown might be "Germ plasm transport to the furrows requires Rbm24a and is Kinesin 1 dependent".

We changed the title of Fig. 4 as suggested by the reviewer, with the following sentence: "*Transport of germ granules to the cleavage furrows requires Rbm24a and kinesin-1*".

3) Fig.6 panel e. The HEK293 data don't really add much to the story since somatic cells don't express germ plasm factors.

We agree with this reviewer's view that somatic cells do not express germ plasm factors. Hence, the purpose of using HEK293 data was to determine whether germ plasm mRNA recruitment by Rbm24a and Buc can be reconstituted in a clean system devoid of germ plasm components. In the revised version, we provided further interpretation in the main text (page 10, line 33).

4) Fig. 6. A description of panel F is missing from the legend.

We added the description of panel F in the revised manuscript. "*(F) Coexpression of nanos3 mRNA with Rbm24a, Buc, or both leads to an enhanced level of nanos3 transcripts.*"

5) Fig 6 G: Is this IP from HEK293 or zebrafish embryos?

This IP data was obtained using HEK293. It is mentioned in the revised main text and in the legend of Fig. 6G.

6) Page 3, Lines 30-33 indicate that Dnd1 and Nanos3 are required for PGC determination, but prior published work from the Moens, Draper and Raz labs (many of which are cited) show that they are required for PGC migration and maintenance of PGC/germline identity rather than PGC specification. A more recent paper Westerich et al Dev cell 2023 should be cited and discussed. Also, Dnd appears twice in this section with two conflicting statements about its functions.

We clarified the statement regarding the function of Dnd1 and Nanos3, and also discussed the work from Westerich et al (Dev Cell, 2023) in the revised version, with the following sentences: "*Bucky ball (Buc), Dead end1 (Dnd1), and Nanos3 are important for PGC formation, with Dnd1 and Nanos3 playing an essential role in PGC migration and maintenance (Bontems et al., 2009; Draper et al, 2007; Gross-Thebing et al, 2017; Kopranner et al, 2001; Marlow & Mullins, 2008; Weidinger et al, 2003). In addition, Dnd1 protein facilitates the localization of nanos3 mRNAs to the periphery of germ plasm particles and regulates their accessibility to the translation machinery (Westerich et al, 2023). Besides the three essential factors, maternal Tdrd6a, Tdrd7a, Rgs14a, mir202-5p and Hook2 function in the organization of germ granules and/or migration of PGCs, but they are not essential for the differentiation of PGCs (Hartwig et al, 2014; Jin et al, 2020; Roovers et al, 2018; Strasser et al, 2008).*"

Reference

Westerich KJ, Tarbashevich K, Schick J, Gupta A, Zhu M, Hull K, Romo D, Zeuschner D, Goudarzi M, Gross-Thebing T et al (2023) Spatial organization and function of RNA molecules within phase-separated condensates in zebrafish are controlled by Dnd1. *Developmental Cell* 58: 1578-1592.e1575

7) Page 3, Line 36-37. Discusses why germ plasm RNAs are degraded in somatic cells but does not discuss the previously published work indicating involvement of miRNAs and Dazl-mediated repression or blocking of the miRNAs in PGCs but not somatic cells. This work should be cited.

We followed this suggestion to discuss the role of Dazl-mediated repression of miRNAs in PGCs but not somatic cells by citing relevant works, with the following sentence: "*Dazl overexpression inhibits miR430 and promotes the stability and polyadenylation of germ plasm mRNAs (Maegawa et al, 2002; Takeda et al, 2009). However, the precise function of Dazl and other germ plasm components, such as Piwil1 and Ddx4, in germ granule formation and PGC specification remains unclear due to obstacles in obtaining their maternal mutants (Bertho et al, 2021; Hartung et al, 2014; Houwing et al, 2007)*".

References

Maegawa S, Yamashita M, Yasuda K, Inoue K (2002) Zebrafish DAZ-like protein controls translation via the sequence 'GUUC'. *Genes to Cells* 7: 971-984

Takeda Y, Mishima Y, Fujiwara T, Sakamoto H, Inoue K (2009) DAZL Relieves miRNA-Mediated Repression of Germline mRNAs by Controlling Poly(A) Tail Length in Zebrafish. *PLOS ONE* 4: e7513

8) Page 4, Line 28. "directed migration". Consider instead directed transport as migration is usually used to describe movement of cells rather than subcellular entities.

We changed the word "migration" to "transport" in the revised manuscript.

9) Page 10, Line 29. "This is the first evidence demonstrating that a protein component controls the localization of germ plasm mRNAs in phase separated structures". This sentence is a bit misleading as written and should be revised as several proteins have been demonstrated to be involved in localizing germ plasm mRNAs in various species. It is more accurate and fairer to others in the field to state that "Rbm24 is a novel factor that controls" or something similar.

We followed this suggestion by changing the sentence as follow: "*the present work provides compelling evidence demonstrating that vertebrate Rbm24 is a novel factor controlling the localization of germ plasm mRNAs in phase-separated structures.*"

10) Page 11, Line 1: "The RNA independent phenomenon of phase separation". The data don't really show that localization and aggregation of the proteins is RNA independent, rather it is independent of Rbm24a and the RNAs that it recruits.

We revised the statement as follow: "*The fact that Rbm24a and its associated RNAs are dispensable for phase separation of protein scaffolds in germ granules may be of important significance*".

11) Page 11, Line 6 "previously believed" should be "Previously shown" since there are published data cited.

"Previously believed" was changed to "Previously shown".

12) Page 11, Lines 10-15 discuss the mechanism involving Buc:Rbm24a complexes and Kinesin 1. Prior work, Campbell et. al. showed that Buc binds to Kinesin 1. Several possibilities are

discussed, including Rbm24a potentially binding Kinesin 1 or having allosteric effects on Buc. It would be helpful to state this and incorporate the prior work into the models.

We followed this recommendation and discussed the possibility that the binding of Rbm24a to Buc may allosterically change the conformation of this protein to promote its binding to kinesin-1. We changed the statement in the revised discussion as follow: "*Rbm24a may directly bind to kinesin-1 or exert an allosteric effect on the structure of Buc, thereby promoting its binding to kinesin-1 (Campbell et al., 2015).*"

13) Page 11, Line 25: "four most giant aggregates". Consider instead "four largest aggregates"

We replaced "four most giant aggregates" by "four largest aggregates".

14) Page 11, Page 11, Line 30: Not sure what is meant by "applications in controlling reproductive traits". Do the authors mean use as a contraceptive? Or to increase fertility?

We modified the original last two sentence as "*Thus, Rbm24a may serve as a promising contraceptive target in fish, by utilizing degron-tagging associated with protein degradation approaches. Germ plasm condensates in zebrafish embryos may also serve as an in vivo model in understanding how to aggregate complicated phase-separated granules.*"

Referee #2 (Report for Author)

In the submitted manuscript, Zhang et al. utilize a zebrafish model to investigate the role of an RNA-binding protein, Rbm24a, in regulating germ plasm assembly. They identify Rbm24a as a key factor in the specific sorting of germ plasm mRNAs during PGC formation, primarily through its interaction with Buc, which facilitates the capture of germ plasm mRNAs in phase-separated condensates. This study offers significant insights into the function of Rbm24a as an organizer of germ plasm, enriching our understanding of phase-separated protein scaffolds. While the manuscript provides valuable contributions to the field of germ cell biology, several major concerns need to be addressed before further consideration.

This reviewer feels that this study offers significant insights into the function of Rbm24a as an organizer of germ plasm, enriching our understanding of phase-separated protein scaffolds. We are grateful for this positive feedback.

Major Points

1. Validation of Mrbm24a Mutants: The study extensively utilizes Mrbm24a mutant embryos. It is essential to verify that the BFP-positive embryos are indeed maternally knocked out by sequencing.

To address this question, we first conducted a western blot analysis, revealing that the Rbm24a-RFP protein was absent in RFP⁻&BFP⁺ embryos ($n = 40$, Fig. R2a), as observed under a fluorescence microscope, in contrast to their RFP⁺&BFP⁺ siblings. Additionally, we genotyped these embryos using RT-PCR by amplifying the open reading frame region of *rbm24a* transcripts (Fig. R2b), followed by Sanger sequencing. The agarose gel analysis revealed that almost all PCR products amplified from BFP⁺ embryo siblings displayed smaller bands or high molecular

weight smears compared to the wild-type (Fig. R2c). Notably, the #1 RFP⁺&BFP⁺ embryo lacked a PCR signal, suggesting the presence of strong nonsense-mediated decay of the mutant mRNAs or an unintended large deletion disrupting primer binding sites (Fig. R2c). The Sanger sequencing results showed that wild-type reads were present in all cDNA libraries from four RFP⁺&BFP⁺ siblings. In contrast, the wild-type reads were entirely absent in the RFP⁻&BFP⁺ counterparts (Fig. R2d). Notably, we identified large unintended deletions in these embryos, which closely resemble those reported in a previous study (Zhang *et al.*, 2021). These results thus fully validated that RFP⁻&BFP⁺ embryos are indeed maternal mutants of *rbm24a*.

The data from Fig. R2 are incorporated into Appendix Fig. S3.

References

Zhang C, Lu T, Zhang Y, Li J, Tarique I, Wen F, Chen A, Wang J, Zhang Z, Zhang Y et al (2021) Rapid generation of maternal mutants via oocyte transgenic expression of CRISPR-Cas9 and sgRNAs in zebrafish. *Sci Adv* 7: abg4243

Fig. R2

2. Rescue Experiments: Since the manuscript emphasizes the role of maternal Rbm24a in germ plasm formation, the authors should demonstrate that the injection of in vitro generated Rbm24a protein (as opposed to *rbm24a* mRNA) can rescue the *Mrbm24a* phenotype. Additionally, the effect of *rbm24a* MO injection should be explored.

Thank you very much for your suggestion, which has allowed us to further investigate the nature of germ granules. We attempted to rescue PGC formation by injecting *rbm24a* mRNA and recombinant Rbm24a protein. As the reviewer anticipated, supplementing with *rbm24a* mRNA alone failed to rescue PGC formation in *Mrbm24a* embryos (Fig. R3a), likely because sufficient Rbm24a protein requires time for translation. Injection of *buc* mRNA into *Mrbm24a* embryos also had no effect (Fig. R3a). Interestingly, co-injecting *rbm24a* and *buc* mRNAs promoted PGC formation in 1/30 of *Mrbm24a* embryos (Fig. R3a), suggesting that Rbm24a and Buc proteins may have synergistic effects in organizing additional large germ plasm particles after the cleavage stage.

We also produced recombinant Rbm24a proteins in *Pichia* yeast and Sf9 cells (Fig. R3b). Contrary to our expectation, injection of Rbm24a protein failed to efficiently rescue PGC defects in *Mrbm24a* embryos. Recombinant Rbm24a protein derived from yeast could only rescue PGC formation in rare instances (1/44, Fig. R3c). Consistent with this result, addition of Rbm24a partially rescued the formation of large aggregates at the cleavage furrows at the 4-cell stage, albeit with low efficiency (2/20, Fig. R3d).

The rescue efficiency of mRNA or protein supplementation after fertilization appears limited, likely due to the fact that the germ granule resembles a solid-phase separation state, characterized by an extremely slow components exchange (Eno *et al*, 2019; Kar *et al*, 2025). Like Buc in zebrafish, Oskar in *Drosophila* also shows the ability to form solid-phase separation (Bose *et al*, 2022). This suggests that Rbm24a protein may struggle to be incorporated into the solid phase-separated germ granules in a timely and efficient manner.

Regarding the use of *rbm24a* morpholino, we feel that there will be no PGC phenotype because maternal Rbm24a protein has already accumulated in the early oocytes and early embryos, as we showed in the manuscript. Therefore, morpholino inhibiting the translation of *rbm24a* mRNA in the embryo will have no effect on the function of maternally derived Rbm24a protein.

The data from Fig. R3 are incorporated into Fig. EV5 a-d.

References

Bose M, Lampe M, Mahamid J, Ephrussi A (2022) Liquid-to-solid phase transition of oskar ribonucleoprotein granules is essential for their function in *Drosophila* embryonic development. *Cell* 185: 1308-1324.e1323

Eno C, Hansen CL, Pelegri F (2019) Aggregation, segregation, and dispersal of homotypic germ plasm RNPs in the early zebrafish embryo. *Dev Dyn* 248: 306-318

Kar S, Deis R, Ahmad A, Bogoch Y, Dominitz A, Shvaizer G, Sasson E, Mytlis A, Ben-Zvi A, Elkouby YM (2025) The Balbiani body is formed by microtubule-controlled molecular condensation of Buc in early oogenesis. *Curr Biol* 35: 315-332.e317

Fig. R3

3. Phase Separation Assay: One of the key conclusions is that Rbm24a and Buc bind to *nanos3* mRNA, forming aggregates that stabilize the mRNA. A phase separation assay using purified proteins should be performed to support this claim (see <https://www.ncbi.nlm.nih.gov/pmc/articles/PMC6215329/>).

We greatly appreciate the reviewer's suggestion, which has allowed us to further clarify the binding of the Rbm24a-Buc complex to *nanos3* mRNA. In our efforts, we attempted to express and purify Buc and Rbm24a proteins in *E. coli*, insect cell line Sf9, mammalian cell HEK293FT, and yeast *Pichia pastoris* (Fig. R4a-b). Unfortunately, we were unable to obtain active Buc protein, leading to an unsuccessful experiment. Buc Protein expressed in *E. coli* likely lacked activity due to folding or modification issues, as we observed no condensation formation upon injection into zebrafish embryos (Fig. R4c) — an outcome significantly different from the endogenous state (Fig. R4d). In mammalian cells, Buc expression was too low for purification of sufficient recombinant protein (Fig. R4b). In insect cells and yeast, while Buc-GFP was highly expressed, it underwent self-hydrolysis during purification (Fig. R4a), preventing the recovery of active protein. Since the discovery of Buc as a germ plasm organizer in 2009, no studies have successfully used purified Buc protein for *in vitro* phase separation experiments. This challenge appears to be due to the difficulty in expressing and purifying Buc, and recent work published in *Current Biology* also did not use purified Buc protein for phase separation analysis, opting instead for an endogenous approach (Kar *et al.*, 2025). Resolving this issue may require

identifying the hydrolysis sites and introducing appropriate mutations. However, given the likely presence of multiple hydrolysis sites, determining these sites and overcoming self-hydrolysis presents substantial challenges and uncertainties, making it infeasible in our current study.

Although our attempts to reconstruct the Rbm24a-Buc complex for *nanos3* binding using purified proteins were unsuccessful, similar efforts in HEK293 cells yielded clear results. HEK293 cells lack germ plasm factors, making these attempts comparable to reassembling germ plasm *in vitro*. We observed that the entry of *nanos3* mRNA into phase-separated particles requires the concurrent presence of Rbm24a and Buc (Fig. 6E-G). Combined with our endogenous genetic and molecular data, we have elucidated that Rbm24a and Buc bind to *nanos3* mRNA, forming aggregates that stabilize the mRNA. Based on these findings and difficulties, we plan to further optimize Buc expression, purification, and *in vitro* phase separation experiments in future work and hope for the reviewers' understanding.

References

Kar S, Deis R, Ahmad A, Bogoch Y, Dominitz A, Shvaizer G, Sasson E, Mytlis A, Ben-Zvi A, Elkouby YM (2025) The Balbiani body is formed by microtubule-controlled molecular condensation of Buc in early oogenesis. *Curr Biol* 35: 315-332.e317

Fig. R4

4. Further Validation of Germ Plasm Recruitment: Additional experiments are needed to substantiate the conclusion that "Maternal Rbm24a is required for the recruitment of germ plasm mRNAs." For instance, the injection of *rbm24a* mRNA (alone or in combination with *buc* mRNA)

into a single blastoderm cell could be used to examine whether dispersed germ plasm in *Mrbm24a* mutants can be reorganized, or whether ectopic germ plasm aggregation centers can form in wild-type embryos (see <http://www.ncbi.nlm.nih.gov/pubmed/19249209>).

By following this recommendation, we injected *Rbm24a* protein into wild-type embryos and observed that approximately 1/4 of the embryos exhibited ectopic germ plasm aggregation at the 4-cell stage (Fig. R5a-c). Some embryos displayed a single large aggregate located outside the cleavage furrow (Fig. R5b, arrow), while others showed multiple aggregates accumulating around the marginal zone, with sizes significantly larger than those observed in the uninjected controls (Fig. R5c, arrow heads). This observation is consistent with the distribution of *Buc-GFP* following *Rbm24a* protein injection, where germ granules exhibited a significantly larger volume upon *Rbm24a* overexpression (Fig. R5d-f). Additionally, when we injected *rbm24a* mRNA at the 1-cell stage, 5 out of 45 embryos displayed ectopic PGC formation (Fig. R5g-h). Together with the *Mrbm24a* data presented in Fig. R3, these results further support the conclusion that *Rbm24a* is capable of organizing ectopic germ plasm formation.

The data from Fig. R5 are incorporated into Fig. EV5e-g.

Fig. R5

5. Oocyte-Specific Mutation Analysis: Given the pronounced defects in *Mrbm24a* mutants as early as the 1-cell stage, a more detailed analysis of germ plasm distribution in oocytes is required. What are the effects of an oocyte-specific mutation of *rbm24a* on oocyte development and Balbiani body formation?

We used the *dazl* probe and Buc-GFP transgene to visualize germ plasm in oocytes carrying the *rbm24a* mutation and examine the formation of the Balbiani body at various stages of oocyte development. The first step involved isolating oocytes lacking Rbm24a-RFP under a fluorescence microscope. We identified some stage I oocytes that were devoid of Rbm24a-RFP but still exhibited Buc-GFP signals in the Balbiani body (Fig. R6a-d), suggesting that the mutation occurred before stage I and that Balbiani body formation was not affected by *rbm24a* mutation. Confocal microscopy further revealed that the absence of maternal Rbm24a protein did not disrupt the dynamic localization of Buc-GFP and *dazl* mRNA between stage I and stage III (Fig. Re-g). These results indicate that Rbm24a does not play a role in Balbiani body formation and dissociation, underscoring its specific function in germ granule assembly.

The data from Fig. R6 form a new Fig. EV4.

Fig. R6

6. In Fig. 2p, Mrbm24a mutants appear to lack a specific cell cluster, distinct from PGCs. What cell type is this? Its absence seems closely related to the loss of Mrbm24a.

Thank you for raising this important point. The clustering of scRNA-seq data included three groups of cells with unknown identities, which were combined and labelled as "not determined" in our analysis (Fig. 2P, Fig. R7a). These cell clusters do not have known marker genes, and their top differentially expressed genes (DEGs) predominantly consist of housekeeping genes with relatively mild up- or down-regulation (Fig. R7b-c). We propose that these cells may represent pluripotent cells undergoing differentiation, characterized by unstable gene expression.

As the reviewer noted, one cluster, "not determined-3", located near the EVL group, appears to be markedly reduced in the mutant. In contrast, another cluster, "not determined-2", was increased in *Mrbm24a* embryos. Since two of the top DEGs of "not determined-3" are *krt4* and *lye*, which are markers of the EVL (Fig. R7c), it is likely that "not determined-3" represents cells transitioning to the EVL, while "not determined-2" may correspond to an earlier stage of this transition. The subtle gene expression differences between these two clusters could be related

to the function of Rbm24a; however, they may also reflect artifacts due to slight developmental stage variations between siblings and mutants. Nonetheless, this observation suggests that the loss of maternal Rbm24a may impact somatic cells to some extent. Therefore, we conclude that “the loss of Rbm24a predominantly impairs germ cell differentiation, with a relatively limited influence on the development of somatic cells.”

Fig. R7 corresponds to the new Appendix Fig. S5 in the revised manuscript.

Fig. R7

7. In Figure 4a-d, Buc-GFP-labeled germ plasm granules in wild-type embryos rapidly move toward the cleavage furrows. In contrast, in Mrbm24a mutants, germ plasm granules still appear, which contradicts the absence of granules reported in Figure 3b. This discrepancy requires sibling clarification.

Actually, there is no discrepancy between these figures, because maternal Rbm24a is only essential for the incorporation of germ plasm mRNA components into the phase-separated

protein condensates. The absence of Rbm24a does not affect the phase separation of protein components. In Figure 3B, we demonstrated the failure to recruit germ plasm RNAs in the absence of Rbm24a, while in Figure 4A-D, we illustrated the defective transport of germ plasm protein condensates without Rbm24a and its recruited mRNAs. To facilitate understanding, we further explained this before describing the defective transport of germ plasm by adding the following sentence: “*In Mrbm24a mutants, the initial condensation of Buc-GFP was normal (Appendix Fig. S7a,b), consistent with the observation that the absence of Rbm24a does not disrupt phase separation of protein components (Appendix Fig. S7a)*”.

8. The rationale for using wild-type controls in the RIP-seq experiment should be clarified, why was an IgG control group not used. Additionally, more details are needed regarding the experimental procedures. What control was used to calculate fold changes in Figure 6b?

We used wild-type embryos as the control for two reasons. First, there is no GFP antigen in wild-type (WT) embryos, so the mRNAs precipitated by the GFP antibody in wild-type embryos should represent background levels. Second, the *rbm24a-gfp-KI* fish exhibit normal development and fertility, indicating that the function of Rbm24a-GFP, as well as the transcriptome in the knock-in line, should be comparable to the wild-type.

Details for the analysis of RIP-seq data were included in the experimental procedure. First, IP reads were compared to their respective inputs to identify significant peaks. Using this significant peak data, we calculated KI (IP vs input) and WT (IP vs input), respectively. The fold change was then determined as KI (IP vs input)/WT (IP vs input). Therefore, the control used to calculate fold changes in Fig. 6B is WT (IP vs input).

In response to the reviewer’s concern, we also conducted an additional RIP-seq experiment using an IgG control group and obtained similar results (Fig. R8). These data further support the conclusion that Rbm24a interacts with germ plasm mRNAs.

Fig. R8 corresponds to the new Appendix Fig. S13 in the revised manuscript.

Fig. R8

9. The authors suggest that maternal Rbm24a recruits germ plasm mRNAs to condensates. Given that injecting zGrad might prevent germ plasm accumulation, it would be useful to assess germ plasm distribution at earlier time points.

In the revised manuscript, we first examined the distribution of germ plasm mRNAs at the 4-cell stage following zGrad mRNA injection. At this early stage, the distribution of germ plasm mRNAs was unaffected or slightly affected (Fig. R9a). Using FISH and IF, we further analyzed the distribution of the mRNA marker *nanos3* and the protein marker Piwil1 at high resolution. The results showed that the distribution of both components at the 4-cell stage remained almost unchanged. At the 64-cell stage, however, the condensation size of both *nanos3* mRNA and Piwil1 was reduced in response to zGrad expression. By 24 hpf, both the mRNA and protein components had disappeared (Fig. R9b). These data further support the conclusion that Rbm24a plays a crucial role in maintaining the germ plasm during later developmental stages.

The data from Fig. R9 are incorporated into Fig. 6J and K.

Fig. R9

Minor Points

1. P5L24: There is an extra dot ".".

We removed this dot.

2. P6L2-4: In situ hybridization or immunostaining should be performed to confirm the presence of gonadal somatic cells and germ cells, verifying the germ cell-less phenotype.

In the revised manuscript, we included additional FISH results to confirm the presence of gonadal somatic cells and germ cells. Specifically, a *gsdf* probe was used to label somatic cells, while a *ddx4* probe was employed to label germ cells. Consistent with HE staining, we observed

that the germ cell marker *ddx4* was absent in *Mrbm24a* mutants, whereas expression of the somatic marker *gsdf* was still present in cells that formed the empty testicular tubule structures. These observations thus strengthened our conclusion that loss of *rbm24a* specifically blocks the differentiation of germ cells.

The data from Fig. R10 are incorporated as Appendix Fig. S4b.

Fig. R10

3. Fig. 2o: Statistical analysis is required.

We added the numbers of embryos tested for this figure, as shown here as Fig. R11.

Fig. R11

4. P6L30: The scRNAseq results should be uploaded to a public database.

The scRNAseq data has been uploaded to GEO data base with accession number: GSE267406.

5. P8L1: For the buc-GFP transgenic fish, which promoter and UTR were used?

We used *ef1a* promoter and *SV40* 3'-UTR to generate the *buc-GFP* transgenic fish. A brief description was added in the Materials and Methods section.

6. P8L15: The effectiveness and specificity of the kinesin-1 antibody should be validated in this experimental context.

To validate the specificity of the kinesin-1 antibody, we included a scramble IgG (Abcam, ab18413) injection as a control in our experiments. The results indicated that injection of this IgG at an equal dose did not impact the formation of large germ plasm aggregates (Fig. R12). Regarding the effectiveness of this antibody, we cited an article (Brady et al., 1990), in which the effectiveness of this monoclonal antibody was verified. Consistently, injection of this kinesin-1 antibody blocked transportation of germ granules towards the cleavage furrow, thus further validating its blocking activity.

Fig. R12 is incorporated into Fig. 4 to replace the original Fig. 4O.

Reference

Brady ST, Pfister KK, Bloom GS (1990) A monoclonal antibody against kinesin inhibits both anterograde and retrograde fast axonal transport in squid axoplasm. *Proc Natl Acad Sci U S A* 87: 1061-1065

Fig. R12

7. P8L40: Why was *rbm24a* KI not used for the co-IP experiment?

At the time of conducting the co-IP experiment, we did not possess the *rbm24a-RFP* KI line expressing the *buc-GFP* transgene. We now have this line and performed Co-IP using the endogenous Rbm24a-GFP. The results demonstrated that they interact under endogenous conditions (Fig. R13).

Data in Fig. R13 are incorporated to Appendix Fig. S12.

Fig. R13

8. P9L6: It would be useful to show the reverse experiment-what happens to Buc granules in *Mrbm24a* mutants?

We have examined the status of transgenic *buc-GFP* in *Mrbm24a* embryos (Fig. 4A-D, Movie EV4 and 5). Our findings indicate that while Buc-GFP is able to form condensates in *Mrbm24a*, it is unable to be transported to the cleavage furrows where aggregation of germ granules occurs.

9. P9L27-28, Fig. 6e, f: The conclusion that *nanos3* mRNA stability is enhanced after transfection with Buc or Rbm24a is unsupported. In situ or qRT-PCR only reflects mRNA levels, not stability.

We changed “mRNA stability” to “mRNA expression” wherever appropriate.

10. The description of procedures, such as single-cell transcriptome sampling, needs more detail. Were *Mrbm24a* and sibling single cells mixed for transcriptome sequencing?

Single cells from *Mrbm24a* mutants and siblings were isolated, barcoded and sequenced separately. We provided this detail of the procedure in the revised version.

11. P13L6: The source of the Cas9 protein should be specified.

Cas9 protein was produced through expression in *E. coli* in our lab following a published protocol (Bhoir *et al*, 2018). This was described in the material and methods section of the revised version.

Reference

Bhoir S, Shaik A, Thiruvankatam V, Kirubakaran S (2018) High yield bacterial expression, purification and characterisation of bioactive Human Toubled-like Kinase 1B involved in cancer. *Sci Rep* 8: 4796

12. P13L16-17: The Mrbm24a mutation must be verified by sequencing.

We verified *Mrbm24a* mutation by sequencing in the revised paper, as shown in Fig. R2.

13. P13L22: Which promoter and 3' UTR were used for the buc-GFP transgenic plasmid?

The question is identical to minor point 5. We used *ef1a* promoter and *SV40* 3'-UTR for the *buc-GFP* transgenic plasmid.

14. P15: References for the scRNAseq and bioinformatic analysis should be provided.

We provided references for the scRNAseq and bioinformatic analysis in the revised manuscript.

References

Wagner DE, Weinreb C, Collins ZM, Briggs JA, Megason SG, Klein AM (2018) Single-cell mapping of gene expression landscapes and lineage in the zebrafish embryo. *Science* 360: 981-987

Zheng GX, Terry JM, Belgrader P, Ryvkin P, Bent ZW, Wilson R, Ziraldo SB, Wheeler TD, McDermott GP, Zhu J et al (2017) Massively parallel digital transcriptional profiling of single cells. *Nat Commun* 8: 14049

Wolf FA, Angerer P, Theis FJ (2018) SCANPY: large-scale single-cell gene expression data analysis. *Genome Biology* 19: 15

15. Fig. 6e: The stability of nanos3 mRNA should be assessed with different mutations in both Rbm24a and Buc.

As suggested by the reviewer, we utilized truncated versions of Rbm24a and Buc to examine the stability of *nanos3* mRNA in HEK293 co-transfection experiments. The presence of both wild-type Rbm24a-myc and Buc-GFP significantly enhanced *nanos3* expression (Fig. R14). Intriguingly, coexpression of Buc Δ 4 with wild-type Rbm24a, as well as full-length Buc and Rbm24a Δ C, also upregulated *nanos3* expression (Fig. R14). These results appear to contradict Co-IP data, which suggest that these protein pairs do not interact. A plausible explanation is that Buc may independently bind to *nanos3* mRNA (Krishnakumar *et al*, 2018). Co-IP detects only strong interactions between Buc and Rbm24a, and truncating the disordered regions of Buc or Rbm24a may not entirely abolish the binding between the two proteins. This could explain the persistence of their synergistic effects on *nanos3* stability in the context of the HEK293 cell line.

Reference

Krishnakumar P, Riemer S, Perera R, Lingner T, Goloborodko A, Khalifa H, Bontems F, Kaufholz F, El-Brolosy MA,

Fig. R14

16. Fig. 6f: Figure legends are missing.

We added the legend for Figure 6F in the revised version. “(F) Coexpression of nanos3 mRNA with Rbm24a, Buc, or both leads to an enhanced level of nanos3 transcripts.”

Referee #3 (Report for Author)

In their manuscripts entitled "Rbm24 dictates mRNA recruitment for germ plasm assembly", Zhang et al. explore the role of the Rbp24 protein in the formation of germ granules in zebrafish, particularly its involvement in mRNA recruitment to these granules. They demonstrate that the absence of Rbp24 prevents the coalescence of germ granules into larger aggregates, a phenomenon similar to the effects of disrupted kinesin-dependent motor movement. Additionally, the authors show that without Rbp24, germ granule mRNAs rapidly degrade, ultimately preventing the formation of primordial germ cells (PGCs).

The authors provide several novel insights into the mechanisms of germ granule formation in zebrafish. Notably, they identify a critical role for Rbp24 in both mRNA recruitment to germ granules and the aggregation of smaller granules into larger clusters. The complete loss of PGCs observed upon Rbp24 depletion underscores the central role of this protein in zebrafish germline development. The supplementary videos, especially Video #5, are visually striking and enhance the presentation.

This reviewer acknowledges that our work offers several novel insights into the mechanisms underlying germ granule formation in zebrafish and emphasizes the crucial role of Rbm24a in both mRNA recruitment to germ granules and the formation of large germ plasm clusters. Notably, the reviewer highlighted that Video #5 is visually compelling and plays a significant role in supporting our conclusions. We sincerely appreciate this positive feedback on our manuscript.

However, several claims made by the authors are not fully substantiated by the experimental data provided in the manuscript. My primary and major concern is the assertion that Rbm24a loss leads to the failure of mRNA recruitment to germ granules, ultimately preventing PGC formation. While the authors suggest that this effect is specific to mRNA, as protein components of germ granules still condense, these data are not presented. Therefore, I recommend the following steps to address this gap:

a) Examine the distribution of Buc, a protein known to nucleate germ granules in zebrafish. If WT amount of germ granules still form at the furrow, as indicated by the presence of Buc granules, the loss of Rbm24a likely impacts downstream steps in germ granule formation. Based on the images presented, it appears that Rbm24a is crucial for the recruitment of both mRNAs (e.g., *nanos3*, *tldr7a*, *ddx4*) and proteins (e.g., *Celf1*, *Piwil1*, *Tldr6*) to germ granules, as none of these components accumulate at the furrow in stage 4 embryos. Given that Rbm24a is a splicing factor, it is possible that it regulates the splicing of *buc*. In the absence of Rbm24a, improper splicing of *buc* could occur, preventing the formation of germ granules and subsequently hindering the recruitment of mRNAs to these structures.

We appreciate this insightful comment. Concerning the distribution of Buc, since no commercial Buc antibody is available, we are unable to examine the distribution of endogenous Buc directly. Instead, we monitored the status of transgenic Buc-GFP in *Mrbm24a* embryos (Fig. 4A-D, Fig. R15a). Our findings indicate that while Buc-GFP expresses normally and is able to form condensates in *Mrbm24a* embryos, it is unable to be transported to the cleavage furrows where fusion of germ granules occurs (Fig. R15a-b). If the overall defects of germ plasm assembly were truly attributed to *buc* splicing, introducing a *buc* transgene lacking introns into the *Mrbm24a* embryos should rescue the splicing defect of *buc* and correct the aggregation problem of germ plasm granules. However, this was not observed.

To fully address the reviewer's concern, we have included a qRT-PCR and transcriptome analysis of *Mrbm24a* mutants and sibling embryos in the revised manuscript to verify whether the splicing of *buc* is affected. *buc* mRNA expression was not altered by the loss of *rbm24a* (Fig. R15c). Using rMATS-Turbo software (Wang *et al*, 2024), we thoroughly analyzed alternative splicing (AS) events in the transcriptome of sibling and mutant embryos at 4-cell stage. No significant differences in AS were observed for *buc* (Table R1). Mapping the RNA-seq reads onto the genome further confirmed that *buc* splicing was unaffected in *Mrbm24a* embryos at the 4-cell stage (Fig. R15d).

The data from Fig. R15 are incorporated into Appendix Fig. S7 and S10.

Reference

Wang Y, Xie Z, Kutschera E, Adams JI, Kadash-Edmondson KE, Xing Y (2024) rMATS-turbo: an efficient and flexible computational tool for alternative splicing analysis of large-scale RNA-seq data. *Nat Protoc* 19: 1083-1104

Fig. R15

In response to the comment of this reviewer, we would also like clarify here that protein components such as Celf1, Piwil1, Tdrd6, non-muscle myosin II (NMII-p), and Buc can still be recruited to germ plasm condensates in the absence of Rbm24a (as shown in Fig. EV3 and Movie EV4). However, all tested RNA components failed to do so (Fig. 3A,B). We emphasized this difference in the revised version.

b) Show distribution of Buc in 1-cell - 4-cell embryos in WT and Rbm24a-mutant embryos, show its expression levels using qRT-PCR and western blot analysis. The authors should also verify that the splicing of Buc in Rbm24a mutant fish is similar to the one in WT fish.

We examined the expression level of *buc* mRNA using qRT-PCR and assessed the splicing status of *buc* mRNA through transcriptome analysis. The conclusion is that *buc* splicing is not affected in *rbm24a* mutant embryos, as shown in Fig. R15c-d. As there is no commercially available Buc antibody, we are unable to demonstrate the distribution of endogenous Buc protein in early cleavage stage embryos.

c) Since Rbm24a regulates splicing, it may regulate splicing of germ granule mRNAs, including *nanos3*, *tldr7a*, *ddx4*,...Inappropriately spliced mRNAs might decay faster or lose germ granule zipcodes which would ensure their enrichment in germ granules. The authors should show expression levels of these mRNAs in WT and Rbm24a-mutant embryos to verify that lack of enrichment in germ granules might not be due to lower mRNA expression.

We conducted a comprehensive transcriptome and qRT-PCR analysis comparing *Mrbm24a* mutants with wild-type embryos to address the splicing status and expression levels of *nanos3*, *tldr7a*, *ddx4* and other germ plasm mRNAs. No splicing defects were identified in the germ plasm mRNAs, except for *celf1* (Table R1).

JCEC						
gene_sym	SE	A5SS	A3SS	MXE	RI	
buc	NA	NA	1.00000 0.36235	NA	NA	NA
nanos3	NA	NA	NA	NA	NA	NA
tdrd7a	NA	NA	NA	NA	NA	NA
ddx4	1.00000 1.0000	1.00000 1.00000 1.	1.00000 1.00000 1.000	NA	NA	1
kif5ba	NA	1	1	NA	NA	NA
dnd1	NA	NA	NA	NA	NA	NA
ca15b	1.00000 1.0000	1	1.00000 1.00000 1.000	NA	NA	1
celf1	1.00000 0.2169	1.00000 1.00000 1.	1.00000 1.00000 0.001	NA	NA	1
rgs14a	NA	1	NA	NA	NA	NA
dazl	1.00000 1.0000	1.00000 1.00000 1.	1.00000 1.00000 1.000	1	NA	NA
hook2	1	1.00000 1.00000 1.	1.00000 1.00000	NA	NA	NA
tdrd6	1.00000 1.0000	1.00000 1.00000 0.	1.00000 0.83337 1.000	1.00000 1.00000	0.7742	
gra	1	1	NA	NA	NA	NA
h1m	1.00000 1.0000	1.00000 1.00000 1.	0.38400 0.33760 1.000	NA	NA	1.0000

Table R1. Splicing detected in RNA-seq. Numbers indicate *P*-value for splicing differences between sibling and *Mrbm24a* embryos at 4-cell stage.

When visualized by mapping the RNA-seq reads onto the genome, no discernible alternative splicing changes were observed for any transcripts between siblings and *Mrbm24a* mutants at 4-cell stage (Fig. R16). The changes in the splicing of *Celf1* in *Mrbm24a* mutants are also not readily apparent. Since *nanos3* only has a single exon, it is not shown in this figure. Fig. R16 corresponds to the Appendix Fig. S8 in the revised manuscript.

Fig. R16

Additionally, germ plasm mRNAs were expressed at comparable levels between sibling and *Mrbm24a* embryos at the 4-cell stage, with the exception of *nanos3*, which lacks an intron but showed a significant reduction in expression (Fig. R17a-c). These results suggest that the loss of *rbm24a* does not lead to widespread splicing defects during oogenesis. Fig. R17 is now Appendix Fig. S9 in this revision.

Fig. R17

d) Same goes for the splicing of *Kif5ba*. Lack of movement of germ granules in video 5 might be due to missplicing of this kinesin motor rather than direct effect of *Rbm24a* on the movement on germ granules. The authors should look at mRNA levels of *Kif5ba* and splice variants of this mRNA to verify that its gene expression levels and patterns are normal and like those observed in WT fish.

We also did not observe any significant splicing changes of *kif5Ba* in *Mrbm24a* mutants following RNA-seq analysis (Table R1, Fig. R18a). Additionally, the expression pattern and level of *kif5Ba* were assessed by in situ hybridization and qRT-PCR, with no differences detected between siblings and *rbm24a* mutants (Fig. R18b-d). Data in Fig. R18 are incorporated to Appendix Fig. S10 in the revised version.

Fig. R18

Additional comments: (and not in the order of importance):

1) Title (and abstract): The authors refer to Rbp24. But, as the author indicate, there are two isoforms in zebrafish, a and b and in this manuscript, isoform a has been studied. Please correct.

We changed Rbm24 to Rbm24a in the title.

2) Title (and in text - introduction, line 21): in this manuscript, the authors are investigating germ granules not germ plasm. Germ plasm is the cytoplasm that contains germ granules. The two terms are not interchangeable and the authors should make sure to correct this throughout the text.

We changed “germ plasm” to “germ granules”, “germ plasm particles” or “germ plasm aggregates” in the revised manuscript wherever appropriate.

3) Abstract (and through the text): the authors indicate that Rbp24a is the nucleating organizer of germ granules in zebrafish where in fact Buckyball (Buc) has been show to play this essential role

We revised this statement as follow: "*Rbm24a/Buc complex is the nucleating organizer*".

4) Introduction, line 25-26: "In particular, it is still debated whether proteins or RNAs coordinate the aggregation of RNP components^{18,19}." This might be the case for stress granules and P bodies but in germ granules in *Drosophila*, this role has been well characterized (please cite: <https://doi.org/10.1016/j.celrep.2023.112723>).

We cited this paper in the revised manuscript and change the sentence to "*In particular, it is still debated whether proteins or RNAs coordinate the aggregation of RNP components such as stress granules and P-bodies (Ripin & Parker, 2023; Tauber et al., 2020). In Drosophila, proteins play a key role in pole plasm assembly (Curnutte et al, 2023)*".

5) Introduction, line 26-28: "In *Drosophila*, it is well established that several RNA-binding proteins (RBPs), such as Oskar, Vasa, Nanos, Tudor and Piwi, are critically involved in the biogenesis of the pole plasm by regulating mRNA localization and translation." Of these, only Osk, vasa and Tudor affect germ granule assembly (note the term germ granule assembly, not germ plasm assembly). Nanos protein does not and Piwi does not either. Also, Tudor is not an RBP. Please correct these errors.

We modified the sentence as follow: "*It is well established that several proteins, such as Oskar, Vasa and Tudor, are critically involved in the biogenesis of the pole plasm by regulating mRNA localization and translation*".

We agree that in *Drosophila*, Tudor may not function as an RNA-binding protein. Therefore, we corrected this in the revised manuscript.

6) Introduction, line 30-34: The authors appear to refer to *ddx4* and *Ddx4* as two distinct regulators. Please correct.

We changed *Ddx4* to *ddx4* in the text.

7) Extended Figure 1a: please specifically state in the figure legend that the data has been re-plotted and provide specific citation.

The reference below was cited in the figure legend, and we have clarified that the data has been re-plotted.

White, R. J. et al. A high-resolution mRNA expression time course of embryonic development in zebrafish. *Elife* 6 (2017). <https://doi.org:10.7554/eLife.30860>

8) Text, page 4, Line 24: Fig. 1c). In the F0 embryos, zygotic Rbm24a-GFP protein was expressed explicitly in the lens, heart, skeletal muscle, and inner ear hair cells, highly matching the *rbm24a* mRNA expression pattern in different tissues: Please state where specifically are these data shown or provide these data.

We thank this reviewer for bringing this to our attention and added appropriate references (Fetka *et al*, 2000; Grifone *et al*, 2014; Poon *et al*, 2012) to support these statements. In addition, the right most panel in Fig. EV1b also showed the specific expression pattern of zygotic *rbm24a* transcripts.

References

Fetka I, Radeghieri A, Bouwmeester T (2000) Expression of the RNA recognition motif-containing protein SEB-4 during *Xenopus* embryonic development. *Mech Dev* 94: 283-286

Grifone R, Xie X, Bourgeois A, Saquet A, Duprez D, Shi D-L (2014) The RNA-binding protein Rbm24 is transiently expressed in myoblasts and is required for myogenic differentiation during vertebrate development. *Mech Dev* 134: 1-15

Poon KL, Tan KT, Wei YY, Ng CP, Colman A, Korzh V, Xu XQ (2012) RNA-binding protein RBM24 is required for sarcomere assembly and heart contractility. *Cardiovasc Res* 94: 418-427

9) Page 5, line 2: The authors write that the homozygous *rbm24a*-GFP KI line is viable and fertile, yet no data is shown. Provide the survivability/hatching data of zebrafish with the *rbm24a*-GFP. Is the viability of these fish the same as the viability of the WT fish? What about its fecundity? This

is important to establish that the GFP tagging and the CRISPR KI did not affect the normal function of this gene.

We included these data to support that GFP tagging did not affect the normal function of Rbm24a. The survival rate, the number of embryos produced per pair of fish, as well as images of both embryos and adults were included in the revised manuscript (Fig. R19). Collectively, these results suggest that GFP tagging and CRISPR/Cas9-mediated knock-in did not affect the normal function of *rbm24a*.

Fig. R19 corresponds to Appendix Fig. S1 in this revision.

Fig. R19

10) Figure 1c,d: label on the figure whether you are staining for mRNA or protein (for example: *ddx4* mRNA and Ddx4 protein).

We followed this suggestion and updated the labelling in Fig. 1.

11) Page 5, line 3: "Moreover, the distribution pattern of Rbm24a-GFP unequivocally recapitulates the endogenous status of Rbm24a localization." This data is not shown. Also, the authors write that the GFP KI was generated because no antibody against Rbm24a exist. So, the authors likely refer to ISH of *rbm24a*. But these two stainings are not comparable as the protein expression does not necessarily mirror mRNA expression. Please, correct this statement. Also, provide ISH staining of *rbm24a*-GFP mRNA and compare it to *rbm24a* to show that the two display the same tissue distribution.

We followed these suggestions by modifying the sentence as follow: "Moreover, the distribution pattern of Rbm24a-GFP is similar to the localization of endogenous *rbm24a* transcripts". We further provided in situ hybridization data comparing the expression patterns of

rbm24a in wild-type and *rbm24a-gfp* KI fish (Fig. R20). We added a sentence to describe the results as follow: "*rbm24a-GFP transcripts in the homozygous knock-in line also showed an identical expression pattern to rbm24a mRNA in the wild-type embryo at 24 hpf (Appendix Fig. S1e)*"

Data in Fig. R20 are incorporated in Appendix Fig. S1e.

Fig. R20

12) Page 5: "time until the sphere stage. At the 32-cell stage, the minimum size to successfully pass through the massive degradation window was $43.3 \mu\text{m}^2$ (projected area)." What do authors refer to as degradation? Protein turnover? Degradation is assumed as no protein turnover has been measured. Also, the granules might have dissolved without degrading the protein. Please correct and clarify this statement.

We corrected this statement as follow: "*At the 32-cell stage, the minimum projected area required for successful retention in PGCs was $43.3 \mu\text{m}^2$.*"

13) Page 5, line 26: Like the *rbm24a-GFP*KI line, the homozygous *rbm24a-RFP*KIzpc:cas9line was healthy and fertile, and Rbm24a. These data is not shown. Please provide these data (see comment # 9 above).

We included these data in the supplemental material to show that the RFP tagging and cas9 insertion did not affect the normal function of Rbm24a. The survival rate, the number of embryos produced per pair of fish, as well as images of both embryos and adults were included in the

revised manuscript (Fig. R19).

14) Page 5, line 35 - 45: Please provide the frequency (number of fish) the authors used in their analysis. Also, please provide data showing that *rbm24a* has been knocked down: provide qRT-PCR analysis and WT analysis (they have Rbm24a-RFP so Wb using anti RFP antibody is possible).

We included the number of fish used in these analyses in the revised version. Additionally, we performed genotyping and western blot analysis using an anti-mCherry antibody to confirm the knockout event generated by our system (Fig R21a). qRT-PCR data also showed a drastic decrease of *rbm24a* transcripts in *Mrbm24a* embryos at the 4-cell stage (Fig R21b).

Data in Fig. R21 are presented in Appendix Fig. S3a and b in the revised manuscript.

Fig. R21

15) Page 7: Is Rbm24a essential for the formation of phase-separated germ plasm protein scaffold? To address this question, we checked the status of germ plasm granules in *Mrbm24a* embryos at the 1-cell stage by examining Piwil1 protein and nanos3 mRNA. Unexpectedly, germ plasm granules." Buc is the scaffold of germ granules in zebrafish, which was shown previously - please see comments under "major comments" above.

We corrected this statement this in the revised version as follow: "*Is Rbm24a involved in the formation of phase-separated germ plasm protein scaffold?*".

16) Figure 5a and text on page 8: Another 158 proteins showed at least a 3-fold enrichment in *rbm24a*-GFPKI embryos compared to wildtype controls (Fig 5a). What is the WT control?

The negative control for the IP/MS was performed using lysates from wild-type embryos precipitated with anti-GFP beads. Since no GFP antigen is present in wild-type embryos, this serves as a stringent negative control to accurately estimate fold enrichment.

17) Page 9, Line 27: However, in the presence of Buc or Rbm24a, the stability of nanos3 mRNA was significantly enhanced (Fig. 6e and f). Gene expression levels appear different from images but data has not been quantified. Please provide qRT-PCR analysis. Also, no data showing that the stability of nanos3 mRNA has been affected is shown. A time course is required to determine changes in mRNA stability. Please correct these issues.

The qRT-PCR analysis for the expression of *nanos3* mRNA was shown in Figure 6f. We revised our terminology to refer to mRNA expression instead of mRNA stability, and we tempered any claim related to mRNA stability accordingly.

18) Page 9, Line 30: Closeup of Buc, Rbm24a and nanos3 in Figure 6e would be useful.

We included a high-magnification image of the Buc, Rbm24a, and *nanos3* coexpression group in the revised version for better clarity (Fig. R22).

Data in Fig. R22 replaces the corresponding panels in the original Fig 6E.

Fig. R22

19) Discussion, line 15 and 43: please cite: <https://doi.org/10.1016/j.celrep.2023.112723> as this paper characterizes the contribution of RBPs and mRNA to the formation of germ granules in *Drosophila*.

We cited this work in the manuscript.

20) Discussion: "This is the first evidence demonstrating that a protein component controls the specific localization of germ plasm mRNAs in phase-separated structures." The role of Oskar, Rump and Aub protein in their ability to recruit mRNAs to germ granules in *Drosophila* has been demonstrated (look at Ephrussi and Gavis labs) - please include these works in your manuscript.

We cited these works in the revised manuscript. Also, the statement was modified as follow: "Although Oskar, Rump and Aub proteins have been reported to mediate the recruitment of mRNAs into germ granules in *Drosophila* (Becalska et al, 2011; Ephrussi & Lehmann, 1992; Jain & Gavis, 2008), the present work provides compelling evidence demonstrating that vertebrate Rbm24 is a novel factor controlling the localization of germ plasm mRNAs in phase-separated structures."

References

Becalska AN, Kim YR, Belletier NG, Lerit DA, Sinsimer KS, Gavis ER (2011) Aubergine is a component of a nanos mRNA localization complex. *Developmental Biology* 349: 46-52

Ephrussi A, Lehmann R (1992) Induction of germ cell formation by oskar. *Nature* 358: 387-392

Jain RA, Gavis ER (2008) The *Drosophila* hnRNP M homolog Rumpelstiltskin regulates nanos mRNA localization. *Development* 135: 973-982

21) Discussion: Unexpectedly, the formation of phase-separated protein scaffold is independent of Rbm24a. These data are not shown. Please see my major comments.

Protein components such as Celf1, Piwil1, Tdrd6, non-muscle myosin II (NMII-p), and Buc

can still be recruited to germ plasm condensates in the absence of Rbm24a (as shown in Fig. EV3, and Movie EV4). However, all tested RNA components failed to do so (Fig. 3A and B). We discussed this issue in the revised version.

Dear Dr Shao,

Thank you for submitting a revised version of your manuscript. Your study has now been seen by all original referees, who find that their previous concerns have been addressed and now recommend publication of the manuscript. There remain only a couple of textual edits requested by referee #1 and a few mainly editorial points that have to be addressed before I can extend formal acceptance of the manuscript:

- Please remove the figures from the manuscript file (only figure legends should remain placed below the References (I've removed figures and turned track changes off for easier checks), no track changes
- On the abstract page of the manuscript, please include 4-5 general keyword terms to enhance searchability.
- As we are switching from a free-text author contribution statement towards a more formal statement based on Contributor Role Taxonomy (CRediT) terms, please remove the present Author Contribution section and instead specify each author's contribution(s) directly in the Author Information page of our submission system during upload of the final manuscript. See <https://casrai.org/credit/> for more information.
- Please rename Supplementary Table 1 to Appendix Table S1; Please also note that there is a missing callout for Fig. 1F
- Please convert the Appendix file into PDF format; Appendix figures and tables should be compiled in one Appendix PDF file - title page should contain "Appendix for + ms title" and ToC with the page numbers for the listed items; nomenclature should be Appendix Figure Sx and Appendix Table Sx throughout ms and Appendix PDF; legends should be removed from ms file and placed below the corresponding figures and above the tables
- Please upload the R&T table as a separate file using the template from our GTA
- Please provide suggestions for a short 'blurb' text prefacing and summing up the conceptual aspect of the study in two sentences (max. 250 characters), followed by 3-5 one-sentence 'bullet points' with brief factual statements of key results of the paper; they will form the basis of an editor-written 'Synopsis' accompanying the online version of the article. Please also provide an altered synopsis image, making sure that the aspect ratio conforms to our website's format - it should be exactly 550 pixels wide and between 300-600 pixels high.
- Figure Legends (main + EV):
 1. Please indicate the statistical test used for data analysis in the legends of figures 5B, 6C
 2. Please note that information related to n is missing in the legends of figures 1F, 3E, 4E, F, K, L; EV5 G.
- Please rename the source files to Movie EV1-EV2 with the corresponding callouts, and the legends should be removed from ms file and zipped with each movie file.

With best regards,

Cornelius Schneider

Cornelius Schneider, PhD
Editor | The EMBO Journal
c.schneider@embojournal.org

We realize that it is difficult to revise to a specific deadline. In the interest of protecting the conceptual advance provided by the work, we recommend a revision within 3 months (12th Jun 2025). Please discuss the revision progress ahead of this time with

the editor if you require more time to complete the revisions. Use the link below to submit your revision:

Referee #1:

In this revised manuscript the reviewers have addressed the concerns raised previously both by revising the text and with the addition of new data. Overall, their work provides new insights into the role of maternal RNA binding protein 24 (Rbm24) in germ plasm recruitment and PGC formation. The work is exciting, rigorous, comprehensive, well-presented, clearly written, and uses cutting edge strategies to further our knowledge of the mechanisms regulating formation of the vertebrate germline. Enthusiasm for the study is high, and the authors did an excellent job addressing the reviewer concerns; however, a few minor points would benefit from clarification.

- 1) For the newly added rescue data, the data provided support the conclusion that rescue is inefficient (1/44 rescued) with recombinant protein, and the explanation that exogenous Rbm24 may not be able to incorporate into RNPs seems plausible. However, based on the co-injection data provided for Buc and Rbm24, there does not seem to be sufficient support for the conclusion that they have synergistic effects based on 1/30 rescued compared to 1/44 for Rbm24 alone, especially as the levels of each protein will vary from embryo to embryo. Thus, it is possible that the rescued embryo in the co-injection experiment simply had enough Rbm24.
- 2) "Regarding the effectiveness of this antibody, we cited an article (Brady et al., 1990), in which the effectiveness of this monoclonal antibody was verified." This should be qualified to state that it was verified in squid.
- 3) "At the time of conducting the co-IP experiment, we did not possess the *rbm24a*-RFP KI line expressing the *buc*-GFP transgene. We now have this line and performed Co-IP using the endogenous Rbm24a-GFP. The results demonstrated that they interact under endogenous conditions (Fig. R13)." While it is fair to state this represents the endogenous condition for the *rbm24a*-KI because the tag was inserted into the locus, *buc* is expressed under the *ef1* alpha promoter and thus is not expressed under endogenous conditions. It is more accurate to state *in vivo* or *in ovary*.

Referee #2:

The reviewer is satisfied with the revision.

Referee #3:

The authors have addressed all concerns I have raised during the initial review of the manuscript - thank you.

Referee #1:

In this revised manuscript the reviewers have addressed the concerns raised previously both by revising the text and with the addition of new data. Overall, their work provides new insights into the role of maternal RNA binding protein 24 (Rbm24) in germ plasm recruitment and PGC formation. The work is exciting, rigorous, comprehensive, well-presented, clearly written, and uses cutting edge strategies to further our knowledge of the mechanisms regulating formation of the vertebrate germline. Enthusiasm for the study is high, and the authors did an excellent job addressing the reviewer concerns; however, a few minor points would benefit from clarification.

Thank you again for your enthusiasm to our work.

1) For the newly added rescue data, the data provided support the conclusion that rescue is inefficient (1/44 rescued) with recombinant protein, and the explanation that exogenous Rbm24 may not be able to incorporate into RNPs seems plausible. However, based on the co-injection data provided for Buc and Rbm24, there does not seem to be sufficient support for the conclusion that they have synergistic effects based on 1/30 rescued compared to 1/44 for Rbm24 alone, especially as the levels of each protein will vary from embryo to embryo. Thus, it is possible that the rescued embryo in the co-injection experiment simply had enough Rbm24.

We agree with this comment, and deleted the statement "*This suggests that Rbm24a and Buc proteins may work synergistically to organize additional large germ plasm particles after the cleavage stages*" from the final manuscript.

2) "Regarding the effectiveness of this antibody, we cited an article (Brady et al., 1990), in which the effectiveness of this monoclonal antibody was verified." This should be qualified to state that it was verified in squid.

We added "*in squid*" before citing (Brady et al, 1990) in line 4, page 9 of the final manuscript.

3) "At the time of conducting the co-IP experiment, we did not possess the *rbm24a*-RFP KI line expressing the *buc*-GFP transgene. We now have this line and performed Co-IP using the endogenous Rbm24a-GFP. The results demonstrated that they interact under endogenous conditions (Fig. R13)." While it is fair to state this represents the endogenous condition for the *rbm24a*-KI because the tag was inserted into the locus, *buc* is expressed under the *ef1* alpha promoter and thus is not expressed under endogenous conditions. It is more accurate to state *in vivo* or *in ovary*.

Thank you for highlighting this point. Indeed, we did not use the term "endogenous" in the manuscript; instead, we referred to "*those expressed at physiological levels in genetically modified lines (Appendix Fig. S12b)*."

Referee #2:

The reviewer is satisfied with the revision.

Referee #3:

The authors have addressed all concerns I have raised during the initial review of the manuscript - thank you.

Dear Prof. Shao,

I am pleased to inform you that your manuscript has been accepted for publication in the EMBO Journal.

Yours sincerely,

Cornelius Schneider, PhD
Editor
The EMBO Journal
c.schneider@embojournal.org
